# Outcome-Aware Spectral Feature Learning for Instrumental Variable Regression

Dimitri Meunier [* 1]  Jakub Wornbard [* 1]  Vladimir R. Kostic [* 2 3]  Antoine Moulin [4]
Alek Fröhlich [2 5]  Karim Lounici [6]  Massimiliano Pontil [2 7]  Arthur Gretton [1]

## Abstract

We address the problem of causal effect estimation in the presence of hidden confounders using nonparametric instrumental variable (IV) regression. An established approach is to use estimators based on learned *spectral features*, that is, features spanning the top singular subspaces of the operator linking treatments to instruments. While powerful, such features are agnostic to the outcome variable. Consequently, the method can fail when the true causal function is poorly represented by these dominant singular functions. To mitigate this, we introduce *Augmented Spectral Feature Learning*, a framework that makes the feature-learning process *outcome-aware*. Our method learns features by minimizing a novel contrastive loss derived from an *augmented* operator that incorporates information from the outcome. By learning these task-specific features, our approach remains effective even in the presence of spectral misalignment. We provide a theoretical analysis of this framework and validate our approach on challenging benchmarks.

## 1. Introduction

We study the nonparametric instrumental variable (NPIV) model, a cornerstone of causal inference in the presence of unobserved confounding (Newey & Powell, 2003), which assumes a relationship

$$Y = h_0(X) + U, \quad \mathbb{E}[U \mid Z] = 0, \quad Z \not\!\perp\!\!\!\perp X, \quad (1)$$

where the confounder $U$ is conditionally mean zero with respect to the instrument $Z$. The goal is to recover the

---
[*]Equal contribution [1]Gatsby Computational Neuroscience Unit, University College London [2]CSML, Istituto Italiano di Tecnologia [3]Faculty of Science, University of Novi Sad [4]Universitat Pompeu Fabra [5]DIBRIS, University of Genoa [6]CMAP-Ecole Polytechnique [7]AI Centre, University College London. Correspondence to: Dimitri Meunier <dimitri.meunier.21@ucl.ac.uk>.

*Proceedings of the 43rd International Conference on Machine Learning*, Seoul, South Korea. PMLR 306, 2026. Copyright 2026 by the author(s).

causal effect from treatment $X$ to outcome $Y$ by estimating the structural function $h_0$ from i.i.d. samples of $(Y, X, Z)$. An example is the economic problem of estimating the effect of education on earnings (Card, 1993). $X$ represents years of schooling and $Y$ an individual's wage. A direct regression is likely biased because unobserved factors like innate ability or family background ($U$) can influence both educational attainment and earning potential. To disentangle this effect, one could use an individual's proximity to a college as an instrument ($Z$), as living closer may increase years of schooling ($X$) but is unlikely to be directly correlated with innate ability ($U$).

The NPIV model can be reformulated as a linear inverse problem (Darolles et al., 2011). Taking the conditional expectation with respect to $Z$ on both sides of Eq. (1) yields the integral equation

$$\mathcal{T}h_0 = r_0, \qquad r_0 \doteq \mathbb{E}[Y \mid Z], \quad (2)$$

where $\mathcal{T}: L_2(X) \to L_2(Z)$ is a bounded linear operator that maps every function $h \in L_2(X)$ to its conditional expectation $\mathbb{E}[h(X)|Z] \in L_2(Z)$. Here both the function $r_0$ and the operator $\mathcal{T}$ are unknown, and we only have access to the set of i.i.d. observations. Throughout this work, we assume that there exists a solution to the NPIV problem, that is $r_0$ is in the range of $\mathcal{T}$.

State-of-the-art techniques for NPIV estimation rely on learning adaptive features and integrating them into classic algorithms like two-stage least squares (2SLS) (Xu et al., 2021; Petrulionyte et al., 2024) or primal-dual strategies (Dikkala et al., 2020; Liao et al., 2020; Bennett et al., 2023). One successful technique is SpecIV (Sun et al., 2025), which learns neural net features by approximating a low-rank decomposition of the operator $\mathcal{T}$. Meunier et al. (2026) showed SpecIV to be optimal when the structural function $h_0$ is well-aligned with the top singular functions of $\mathcal{T}$, but degrades otherwise. The issue is that the feature learning process is agnostic to the outcome $Y$; it only captures the main aspects of the relationship between the treatment $X$ and the instrument $Z$. If $h_0$ lies outside of this dominant subspace, the resulting features are uninformative for the final task, leading to failure. In this paper, we address this limitation by proposing a framework for outcome-aware feature learning in NPIV estimation. We introduce a new

spectral objective that incorporates information from the outcome $Y$, guiding the feature learning to identify components of $X$ that are predictable from $Z$ and predictive of $Y$. This is equivalent to learning a low-rank decomposition of a perturbed version of $\mathcal{T}$. Our approach ensures good performance even in cases of spectral misalignment, where target-agnostic methods fail. While our work focuses on IV regression for causal effect estimation, our method is also useful in settings where learning spectral decompositions of conditional operators is relevant. For instance, in off-policy evaluation in reinforcement learning, where value function estimation can be framed as an IV problem (Hu et al., 2025), or when learning evolution operators in fields like molecular dynamics and climate science (Turri et al., 2025).

**Contributions.** Our contributions are as follows:

1. We identify a fundamental limitation of existing spectral methods for NPIV: the learned features are outcome-agnostic, degrading performance in cases of spectral misalignment. To address this, we propose *Augmented Spectral Feature Learning* and introduce an augmented operator, $\mathcal{T}_\delta$, which incorporates information on the outcome into the feature learning problem. This leads to a new, principled contrastive loss function for learning task-specific spectral features.

2. We provide a comprehensive theoretical analysis of our method. This includes a full generalization error bound for the resulting 2SLS estimator, characterizing the settings where the augmented approach remains robust to the spectral misalignment issues of previous methods.

3. We validate our theory on challenging synthetic and semi-synthetic examples, including a new and more challenging version of a dSprites IV benchmark (Xu et al., 2021). The results demonstrate the practical benefits of outcome-aware feature learning in the challenging regimes where standard SpecIV fails. In addition, we include an Off-Policy Evaluation (OPE) experiment in the context of reinforcement learning (Chen et al., 2022), showing that our approach remains robust and competitive in challenging, dynamically changing environments.

**Paper Organization.** The remainder of the paper is structured as follows. We situate our contribution within the broader literature in Sec. 2. Sec. 3 introduces the notation, reviews the 2SLS estimator, and the SpecIV method (Sun et al., 2025). In Sec. 4, we introduce our outcome-aware framework. Sec. 5 presents our main theoretical results. In Sec. 6 we present numerical experiments that validate our theory and demonstrate the effectiveness of our approach. All proofs are deferred to the appendix.

## 2. Related Work

This section provides an overview of the research areas that are most relevant to our study.

**2SLS methods.** The classical approach to IV regression is the 2SLS method. In its nonparametric form, the first stage involves estimating the conditional expectation of the treatment given the instrument, and the second stage uses these predictions to estimate the structural function. Early influential works used sieve or series estimators, approximating unknown functions with basis functions like polynomials or splines (Newey & Powell, 2003; Hall & Horowitz, 2005; Blundell et al., 2007; Chen & Pouzo, 2012; Chen & Christensen, 2018). Other approaches include Tikhonov regularization to stabilize the inverse problem (Darolles et al., 2011) or frame the problem in reproducing kernel Hilbert spaces (Singh et al., 2019; Meunier et al., 2024; Shen et al., 2025). More recently, deep learning has been used to handle the nonparametric components of the 2SLS procedure. DeepIV (Hartford et al., 2017) uses a mixture density network to estimate the conditional distribution of the treatment given the instruments in the first stage, and then a second network for the structural function in the second stage. Deep Feature IV (DFIV; Xu et al., 2021; Petrulionyte et al., 2024; Kim et al., 2025) uses neural networks to learn optimal features of the instruments, which are then used as inputs for the first-stage regression.

**Saddle-point methods.** An alternative to 2SLS is to frame NPIV as a minimax optimization problem. These methods, often rooted in a generalized method of moments (GMM) framework, seek an equilibrium between a player that minimizes a loss function and an adversary that maximizes the violation of the moment conditions. This approach can bypass the direct estimation of conditional expectations. Different formulations exist, such as those based on the Lagrangian of a constrained least-norm problem (Bennett et al., 2023; Liao et al., 2020) or on maximizing the moment deviation directly (Lewis & Syrgkanis, 2018; Dikkala et al., 2020; Wang et al., 2022; Bennett et al., 2025; Shen et al., 2026). These methods are particularly well-suited for high-dimensional settings and integration with deep learning models.

**Spectral features learning.** When the instrument-treatment relationship is complex, learning good features is crucial. Spectral methods use techniques such as the singular value decomposition (SVD) to obtain a low-dimensional representation of the conditional expectation operator $\mathcal{T}$. Such an SVD can typically be estimated by minimizing a contrastive loss that has been used in various contexts (Sun et al., 2025; Hu et al., 2025; Kostic et al., 2024; Turri et al., 2025). While these methods are powerful, the learned features are agnostic to the outcome; they capture only the dominant modes of the instrument-treatment relationship, which may not be sufficient for predicting the outcome.

**Outcome-aware and adaptive methods.** Our work is part of a growing literature on adaptive methods for NPIV. While spectral methods like SpecIV (Sun et al., 2025) are powerful, their outcome-agnostic nature can be a significant drawback, as discussed above. The features are learned solely from the instrument-treatment relationship and may not be informative for predicting the outcome. Our approach addresses this by making the feature learning process outcome-aware and ensuring that the learned representations are not only predictive of the treatment but also relevant to the causal relationship of interest. Bruns-Smith (2025) proposes a different adaptive method, where in stage one, he constructs a set of $Z$-features that is predictive of $Y$, and then estimates the structural function by selecting an element of an arbitrary high-dimensional function class whose projection onto the learned $Z$-features is most predictive of $Y$. The method is guaranteed to achieve low weak error under mild or testable assumptions. This approach is similar in spirit to ours but differs in that the usefulness of the $Z$ features for the IV problem is implicit and must be tested post hoc. Moreover, the strong-norm convergence holds under more stringent restrictions on the ill-posedness of the inverse problems than we needed.

## 3. Preliminaries

**Function Spaces.** $Y$ is defined on $\mathbb{R}$, while $X$ and $Z$ take values in measurable spaces $\mathcal{X}$ and $\mathcal{Z}$, respectively. For a variable $R \in \{X, Z\}$, $L_2(R)$ is the space of square-integrable functions ($\mathbb{E}[f(R)^2] < \infty$).

**Operators on Hilbert Spaces.** Let $\mathcal{H}$ be a Hilbert space. For a bounded linear operator $A$ acting on $\mathcal{H}$, we denote by $\|A\|_{\mathrm{op}}$ its operator norm, $\|A\|_{\mathrm{HS}}$ its Hilbert–Schmidt norm, $A^\dagger$ its Moore–Penrose inverse, and $A^\star$ its adjoint. For finite-dimensional operators, the Hilbert–Schmidt norm coincides with the Frobenius norm. We denote $\mathcal{R}(A)$ and $\mathcal{N}(A)$ the range and null spaces of $A$, respectively. Given a closed subspace $M \subseteq \mathcal{H}$, we write $M^\perp$ its orthogonal complement, $\overline{M}$ its closure, and $\Pi_M$ the orthogonal projection onto $M$. Denote the orthogonal projection onto $M^\perp$ by $(\Pi_M)_\perp \doteq I_{\mathcal{H}} - \Pi_M$. For $f, h \in L_2(X)$, $g \in L_2(Z)$, the rank-one operator $g \otimes f$ is defined as $(g \otimes f)(h) = \langle h, f \rangle g$, generalizing the standard outer product. For $x \in \mathbb{R}^d$, we write $\|x\|_{\ell_2}$ for the Euclidean norm.

**Data Splitting and Empirical Expectations.** We consider two independent datasets: $\tilde{\mathcal{D}}_m = \{(\tilde{z}_i, \tilde{x}_i, \tilde{y}_i)\}_{i=1}^m$, to learn features for $X$ and $Z$, and $\mathcal{D}_n = \{(z_i, x_i, y_i)\}_{i=1}^n$, to estimate the structural function. $\widehat{\mathbb{E}}_m$ and $\widehat{\mathbb{E}}_n$ denote empirical expectations with respect to $\tilde{\mathcal{D}}_m$ and $\mathcal{D}_n$, respectively.

**Feature Maps, Covariance Operators and Projections.** Let $d \in \mathbb{N}^*$, $\varphi^{(d)} \colon \mathcal{X} \to \mathbb{R}^d$ be a feature map with linearly independent components $\varphi_1^{(d)}, \ldots, \varphi_d^{(d)} \in L_2(X)$, and

$\Phi^{(d)} \doteq [\varphi_1^{(d)}, \ldots \varphi_d^{(d)}]$ be the operator defined as

$$\Phi^{(d)} \colon \mathbb{R}^d \to L_2(X), \alpha \mapsto \sum_{i=1}^d \alpha_i \varphi_i^{(d)}.$$

Its adjoint is given by $\Phi^{(d)\star} \colon h \mapsto (\langle h, \varphi_i^{(d)} \rangle_{L_2(X)})_{i=1}^d$. Let $C_{\varphi^{(d)}} \doteq \Phi^{(d)\star} \Phi^{(d)} = \mathbb{E}[\varphi^{(d)}(X)\varphi^{(d)}(X)^\intercal]$ be the (uncentered) covariance operator, $\widehat{C}_{\varphi^{(d)}} \doteq \widehat{\mathbb{E}}_n[\varphi^{(d)}(X)\varphi^{(d)}(X)^\intercal]$ its empirical counterpart, and $\Pi_{\varphi^{(d)}} \doteq \Phi^{(d)}(C_{\varphi^{(d)}})^{-1}\Phi^{(d)\star}$ be the corresponding orthogonal projection operator. Analogous definitions apply for a feature map $\psi^{(d)} \colon \mathcal{Z} \to \mathbb{R}^d$, yielding $\Psi^{(d)}, C_{\psi^{(d)}}, \widehat{C}_{\psi^{(d)}}$ and $\Pi_{\psi^{(d)}}$. The (uncentered) cross-covariance operator between $\varphi^{(d)}$ and $\psi^{(d)}$ is

$$C_{\psi^{(d)}, \varphi^{(d)}} \doteq \Psi^{(d)\star}\Phi^{(d)} = \mathbb{E}[\psi^{(d)}(Z)\varphi^{(d)}(X)^\intercal].$$

Denote $\widehat{C}_{\psi^{(d)}, \varphi^{(d)}}$ its empirical counterpart. We drop the superscript $d$ when it is clear from context.

**2SLS in feature space.** Given feature maps $\varphi^{(d)} \colon \mathcal{X} \to \mathbb{R}^d$ and $\psi^{(p)} \colon \mathcal{Z} \to \mathbb{R}^p$, a prominent estimator for NPIV is the 2SLS estimator (see, e.g., Blundell et al., 2007), given by $\widehat{h}_{2\mathrm{SLS}}(x) = \varphi^{(d)}(x)^\intercal \widehat{\beta}_{2\mathrm{SLS}}$, with

$$\widehat{\beta}_{2\mathrm{SLS}} = \left\{ \widehat{C}_{\varphi^{(d)}, \psi^{(p)}} \widehat{C}_{\psi^{(p)}}^{-1} \widehat{C}_{\psi^{(p)}, \varphi^{(d)}} \right\}^{-1}$$
$$\cdot \widehat{C}_{\varphi^{(d)}, \psi^{(p)}} \widehat{C}_{\psi^{(p)}}^{-1} \widehat{\mathbb{E}}_n[Y \psi^{(p)}(Z)].$$

When $d = p$ and the cross-covariance matrix is invertible, it simplifies to $\widehat{\beta}_{2\mathrm{SLS}} = \widehat{C}_{\psi^{(d)}, \varphi^{(d)}}^{-1} \widehat{\mathbb{E}}_n[Y \psi^{(d)}(Z)]$.

**2SLS with spectral features.** Features plugged into the 2SLS estimator can be either fixed or learned adaptively from data (see Section 2 for a discussion on related work). In the SpecIV approach (Sun et al., 2025), features are learned to approximate the top eigenstructure of $\mathcal{T}$. Throughout the paper, we make the following mild assumption (Darolles et al., 2011; Meunier et al., 2026).

**Assumption 1.** $\mathcal{T} \colon L_2(X) \to L_2(Z)$ is a compact operator.

This allows us to write a countable singular value decomposition (SVD) for $\mathcal{T}$,

$$\mathcal{T} = \sum_{i \geq 1} \lambda_i u_i \otimes v_i, \quad u_i \in L_2(Z), \quad v_i \in L_2(X),$$

$\lambda_1 \geq \lambda_2 \geq \cdots > 0$, where $u_i$'s and $v_i$'s are orthonormal bases for $\overline{\mathcal{R}(\mathcal{T})} \subseteq L_2(Z)$ and $\mathcal{N}(\mathcal{T})^\perp \subseteq L_2(X)$, respectively. The operator $\mathcal{T}^{(d)} \doteq \sum_{i=1}^d \lambda_i u_i \otimes v_i$ is the best (in terms of operator or Hilbert–Schmidt norm) rank-$d$ approximation to $\mathcal{T}$. To avoid ambiguity, we assume that $\lambda_d > \lambda_{d+1}$. We do not assume that $\mathcal{T}$ is injective, since we can always target the minimum-norm solution (Florens et al., 2011)

$$\bar{h}_0 \doteq \mathcal{T}^\dagger r_0 = \sum_{i \geq 1} \frac{1}{\lambda_i} \langle r_0, u_i \rangle_{L_2(Z)} v_i.$$

When $\mathcal{T}$ is injective, $h_0 = \bar{h}_0$. In what follows, we denote $\bar{h}_0$ as $h_0$ and do not distinguish between the structural function and the minimal solution to the NPIV problem.

Given feature maps $\varphi_\theta^{(d)} : \mathcal{X} \to \mathbb{R}^d$ and $\psi_\theta^{(d)} : \mathcal{Z} \to \mathbb{R}^d$, parametrized by neural networks, Sun et al. (2025) proposed to learn the features by minimizing the empirical counterpart to the following loss

$$
\begin{aligned}
\mathcal{L}_0^{(d)}(\theta) \doteq{} & \mathbb{E}_X \mathbb{E}_Z \Big[ \big( \varphi_\theta^{(d)}(X)^\top \psi_\theta^{(d)}(Z) \big)^2 \Big] \\
& - 2 \mathbb{E} \Big[ \varphi_\theta^{(d)}(X)^\top \psi_\theta^{(d)}(Z) \Big].
\end{aligned}
\tag{3}
$$

where the first expectation is over the product of the marginals of $X$ and $Z$, and the second is over the joint distribution. It is shown by Meunier et al. (2026, Theorem 2) that $\mathcal{L}_0^{(d)} \geq -\|\mathcal{T}_d\|_{\mathrm{HS}}^2$, and that the minimum is achieved if and only if $\Psi_\theta^{(d)} \Phi_\theta^{(d)\star} = \mathcal{T}_d$. Therefore, by minimizing the empirical counterpart to Eq. (3), one learns features that approximate the best low-rank approximation to $\mathcal{T}$.

**Remark.** Meunier et al. (2026) introduced Eq. (3) under the stronger assumption that $\mathcal{T}$ is Hilbert–Schmidt, but neither Hilbert–Schmidt nor compactness is needed for the loss to be well-defined: boundedness of $\mathcal{T}$ suffices. Compactness is used only to justify the truncated SVD target $\mathcal{T}_d$ and attainment of the rank-$d$ optimum (Appendix B).

## 4. Outcome-Aware Spectral Feature Learning

As discussed in Section 1, standard SpecIV learns features that are agnostic to the outcome $Y$. This can lead to poor performance when the structural function $h_0$ is not well-aligned with the top singular functions of $\mathcal{T}$ (Meunier et al., 2026). To mitigate this, we augment the SpecIV loss with a regularization term that incorporates information from the outcome $Y$ by projecting it onto the orthonormal basis of the $Z$-features. The resulting loss is defined as

$$
\begin{aligned}
& \mathcal{L}_\delta^{(d)}(\theta) \doteq \mathcal{L}_0^{(d)}(\theta) + \mathcal{R}_\delta^{(d)}(\theta), \\
& \mathcal{R}_\delta^{(d)}(\theta) \doteq -\delta^2 \mathbb{E}[Y \psi_\theta^{(d)}(Z)]^\top C_{\psi_\theta^{(d)}}^{-1} \mathbb{E}[Y \psi_\theta^{(d)}(Z)].
\end{aligned}
\tag{4}
$$

We propose to learn features by minimizing the empirical counterpart of Eq. (4) over the training set $\tilde{\mathcal{D}}_m$. The regularization term, controlled by the hyperparameter $\delta$, encourages the learned instrument features to be predictive of $Y$. Indeed, $\delta^{-2} \mathcal{R}_\delta^{(d)}(\theta)$ is equal to the mean squared error of the best linear predictor of $Y$ from $\psi_\theta^{(d)}(Z)$ (up to a constant independent of $\theta$). As backpropagation through the inverse covariance matrix can be numerically unstable, we instead minimize the following equivalent loss jointly over $\theta$ and $\omega \in \mathbb{R}^d$:

$$
\mathcal{L}_\delta^{(d)}(\theta, \omega) \doteq \mathcal{L}_0^{(d)}(\theta) - 2\delta \mathbb{E}[Y \psi_\theta^{(d)}(Z)]^\top \omega + \omega^\top C_{\psi_\theta^{(d)}} \omega.
\tag{5}
$$

For any fixed $\theta$, this loss is convex in $\omega$ and its minimum is attained at $\omega_\theta^{(d)} = \delta C_{\psi_\theta^{(d)}}^{-1} \mathbb{E}[Y \psi_\theta^{(d)}(Z)]$. Substituting this solution back into Eq. (5) recovers the profile loss in Eq. (4).

**Operator learning perspective.** We now show this modification is equivalent to learning a low-rank approximation of an augmented version of the operator $\mathcal{T}$. For $\delta \in \mathbb{R}$, we define the operator

$$
\mathcal{T}_\delta : L_2(X) \times \mathbb{R} \to L_2(Z), (h, a) \mapsto \mathcal{T}h + a \cdot \delta \cdot r_0.
$$

$\mathcal{T}_\delta$ augments $\mathcal{T}$ with an additional "column" aligned with $r_0$, as captured by the compact notation $\mathcal{T}_\delta = [\mathcal{T} \mid \delta r_0] = \mathcal{T}[I_{L_2(X)} \mid \delta h_0]$. As $\mathcal{T}$ is compact, $\mathcal{T}_\delta$ is also compact and admits an SVD

$$
\mathcal{T}_\delta = \sum_{i \geq 1} \sigma_{*,i} \psi_{*,i} \otimes (\varphi_{*,i}, \omega_{*,i}), \quad \psi_{*,i} \in L_2(Z), \tag{6}
$$

where $(\varphi_{*,i}, \omega_{*,i}) \in L_2(X) \times \mathbb{R}$ with $\sigma_{*,1} \geq \sigma_{*,2} \geq \cdots > 0$. The following proposition formalizes the connection: minimizing the augmented loss $\mathcal{L}_\delta^{(d)}$ is equivalent to finding the best rank-$d$ approximation of the augmented operator $\mathcal{T}_\delta$. We denote this best approximation by

$$
\mathcal{T}_\delta^{(d)} \doteq \sum_{i=1}^{d} \sigma_{*,i} \psi_{*,i} \otimes (\varphi_{*,i}, \omega_{*,i}).
$$

To avoid ambiguity in the definition of $\mathcal{T}_\delta^{(d)}$, we assume that $\sigma_{*,d} > \sigma_{*,d+1}$.

**Proposition 4.1.** *Given $\delta \in \mathbb{R}$, for all parameters $\theta$ and $\omega$ it holds that $\mathcal{L}_\delta^{(d)}(\theta, \omega) \geq - \left\| \mathcal{T}_\delta^{(d)} \right\|_{\mathrm{HS}}^2$. The lower bound is achieved if and only if the learned operator $\Psi_\theta^{(d)}[\Phi_\theta^{(d)\star} \mid \omega]$ is equal to $\mathcal{T}_\delta^{(d)}$.*

Intuitively, augmenting the operator with outcome information amplifies the components of $h_0$ that would otherwise lie in the low singular value region of $\mathcal{T}$, thereby improving their alignment with the top spectral features of the augmented operator, as illustrated in Figure 1. This operator characterization is the bridge to our estimator guarantees: in Section 5, we show that approximating the leading singular subspaces of $\mathcal{T}_\delta$ controls the approximation term in the downstream 2SLS error bound. The same section formalizes this effect by bounding the distance between the learned subspaces and the signal subspaces of $\mathcal{T}$. One can also encourage the learned features to retain predictive information about additional aspects of the outcome by extending the augmentation to multiple functions of $Y$, such as higher conditional moments $\mathbb{E}[Y^k \mid Z]$. We discuss this higher-rank extension in Section F. The learning objective and optimality characterization via truncated SVD remain unchanged. A complete theoretical analysis of general rank $K$ perturbations requires further development of the perturbation framework and is left for future work.

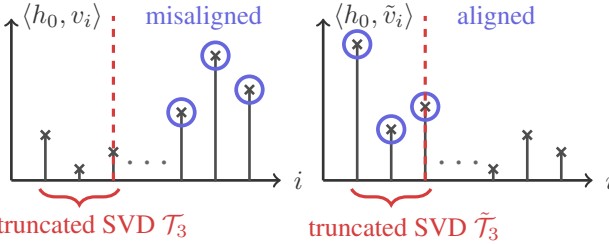

**Figure 1.** In case of severe misalignment of $h_0$ and $\mathcal{T}$ (left), an ideal solution would aim to find another operator $\tilde{\mathcal{T}}$ whose top singular functions $\tilde{v}_i$ capture the signal in $h_0$ (right).

## 5. Analysis

We now present the statistical guarantees for our estimator of $h_0$. Our first result is a non-asymptotic, high-probability error bound for the 2SLS estimator that is agnostic to the choice of representation $(\varphi_\theta^{(d)}, \psi_\theta^{(d)})$. This improves upon prior results, such as Chen & Christensen (2018), which typically provide guarantees in expectation. To this end, we introduce three standard assumptions. The first requires the whitened features to be uniformly bounded.

**Assumption 2** (Representation Boundedness). Denoting the covariances of the instrument and feature representations by $C_{Z,\theta} = \mathbb{E}[\psi_\theta^{(d)}(Z)\psi_\theta^{(d)}(Z)^\intercal]$ and $C_{X,\theta} = \mathbb{E}[\varphi_\theta^{(d)}(X)\varphi_\theta^{(d)}(X)^\intercal]$, respectively, there exists $\rho \geq 1$ such that representations satisfy

$$\operatorname*{ess\,sup}_{x\sim\mathbb{P}_X} \max_{j\in[d]} \left|\left(C_{X,\theta}^{-1/2}\varphi_\theta^{(d)}(x)\right)_j\right|$$
$$\bigvee \operatorname*{ess\,sup}_{z\sim\mathbb{P}_Z} \max_{j\in[d]} \left|\left(C_{Z,\theta}^{-1/2}\psi_\theta^{(d)}(z)\right)_j\right| \leq \rho.$$

Let $C_{ZX,\theta} = \mathbb{E}[\psi_\theta^{(d)}(Z)\varphi_\theta^{(d)}(X)^\intercal]$ be the cross-covariance between the instrument and feature representations. The measure of ill-posedness is defined as $c_{\varphi_\theta^{(d)},\psi_\theta^{(d)}} \doteq \sigma_d(C_{Z,\theta}^{-1/2}C_{ZX,\theta}C_{X,\theta}^{-1/2})$, where $\sigma_d(\cdot)$ is the $d$-th singular value.

**Assumption 3** (Measure of ill-posedness). $c_{\varphi_\theta^{(d)},\psi_\theta^{(d)}} > 0$.

The measure of ill-posedness is equal to $\sigma_d(\Pi_{\Psi_\theta^{(d)}}\mathcal{T}\Pi_{\Phi_\theta^{(d)}})$ and captures the stability of the inverse problem when restricted to the learned feature spaces. Its positivity implies non-singularity of $C_{ZX,\theta}$, which guarantees that there exists a vector $\beta_\theta \in \mathbb{R}^d$ such that $h_\theta(x) = \varphi_\theta^{(d)}(x)^\intercal\beta_\theta$ satisfies the instrumental moment condition: $C_{ZX,\theta}\beta_\theta = \mathbb{E}[Y\psi_\theta^{(d)}(Z)] = \mathbb{E}[r_0(Z)\psi_\theta^{(d)}(Z)] = \mathbb{E}[h_0(X)\psi_\theta^{(d)}(Z)]$. Assumption 3 is reasonable provided that some conditions on the eigenvalues of $\mathcal{T}$ are satisfied and our features capture an accurate representation of the singular spaces of $\mathcal{T}_d$ (see Proposition C.5 in the appendix).

Our final assumption concerns the tail behavior of the model's error terms. For the precise definition of sub-Gaussian random variables, we refer to Definition D.1 in Appendix D.2.

**Assumption 4** (Sub-Gaussian distributions). The model noise $U = Y - h_0(X)$ and the function approximation error $(h_0 - h_\theta)(X)$ are sub-Gaussian random variables.

With these conditions, we consider the following 2SLS estimator,

$$\widehat{h}_\theta(x) = \varphi_\theta^{(d)}(x)^\intercal\widehat{\mathbb{E}}_n[\psi_\theta^{(d)}(Z)\varphi_\theta^{(d)}(X)^\intercal]^{-1}\widehat{\mathbb{E}}_n[Y\,\psi_\theta^{(d)}(Z)],$$

and state our main result below.

**Theorem 5.1.** *Let Assumptions 2-4 be satisfied. Given* $\tau \in (0,1)$, *let* $n \geq 16\,d\,\rho^2\,\log^2(4d/\tau)c_{\varphi_\theta^{(d)},\psi_\theta^{(d)}}^{-2}$. *Then there exists an absolute constant* $C > 0$ *such that with probability at least* $1 - \tau$,

$$\|\widehat{h}_\theta - h_0\|_{L_2(X)} \leq C\Big(\|h_0 - h_\theta\|_{L_2(X)}$$
$$+ \frac{1}{c_{\varphi_\theta^{(d)},\psi_\theta^{(d)}}}\sqrt{\frac{d}{n}}\sqrt{\sigma_U^2 + \frac{\rho^2}{n}}\,\log\frac{4}{\tau}\Big),$$

*where* $\sigma_U^2$ *is the noise variance.*

This theorem provides a non-asymptotic excess-risk bound that separates the error into a deterministic approximation error ($\|h_0 - h_\theta\|$) and a statistical error. The latter depends on the dimension $d$, sample size $n$, and the feature-dependent ill-posedness. It improves on existing guarantees (Theorem B.1. Chen & Christensen, 2018) by holding in high probability under a sub-Gaussian assumption.

**Controlling the approximation error with learned representations.** We now specialize the general bound from Theorem 5.1 to the features learned by our outcome-aware method. In light of Proposition 4.1, recalling that the leading left and right singular subspaces of $\mathcal{T}_\delta$ are the ranges of $\Psi_*^{(d)} = [\psi_{*,1}\,|\,\dots\,|\psi_{*,d}]$ and $[\,\Phi_*^{(d)}\,|\,\omega_*\,]^\star$, where $\Phi_*^{(d)} = [\varphi_{*,1}\,|\,\dots\,\varphi_{*,d}]$, see Eq. (6), the feature learning stage of our approach produces $\psi_{\widehat{\theta}_m}^{(d)}$ and $\varphi_{\widehat{\theta}_m}^{(d)}$, where the parametrization $\widehat{\theta}_m$ is learned from dataset $\tilde{\mathcal{D}}_m$. To quantify their quality, as proposed in Kostic et al. (2024) and Meunier et al. (2026), we use the optimality gap

$$\mathcal{E}_d(\theta, \omega, \delta) \doteq \big\|\mathcal{T}_\delta^{(d)} - \Psi_\theta^{(d)}[\Phi_\theta^{(d)\star}\,|\,\omega]\big\|_{\mathrm{op}}, \tag{7}$$

With this setup, our analysis hinges on relating the singular subspaces of the augmented operator $\mathcal{T}_\delta$ back to the original singular subspaces of $\mathcal{T}$. To do this, we first partition the singular components of $\mathcal{T}$ into a $d$-dimensional *"signal"* subspace and an infinite-dimensional *"noise"* subspace. Namely, let $\overline{N} \dot{\cup} \underline{N} = \mathbb{N}$ be the partition, where we take $|\overline{N}| = d$ spectral features for the signal. We define the signal components $s_d = \overline{V}_d\overline{V}_d^*h_0 = \sum_{i\in\overline{N}}\overline{\alpha}_i v_i$ and

noise components $q_d = \underline{V}_d \underline{V}_d^* h_0 = \sum_{i \in \underline{N}} \underline{\alpha}_i v_i$, where the partition of the SVD of $\mathcal{T}$ is $\mathcal{T} = \overline{U}_d \overline{\Lambda}_d \overline{V}_d^* + \underline{U}_d \underline{\Lambda}_d \underline{V}_d^*$ with $\overline{\Lambda}_d = \mathrm{diag}(\lambda_i)_{i \in \overline{N}}$. The augmented operator $\mathcal{T}_\delta$ can be viewed as a perturbation of $\mathcal{T}$ relative to the positioning of the signal:

$$\mathcal{T}_\delta = [\,\mathcal{T} \,|\, \delta \mathcal{T} h_0\,]$$
$$= \overline{U}_d \overline{\Lambda}_d [\,\overline{V}_d^* \,|\, \delta \overline{\alpha}\,] + \underline{U}_d \underline{\Lambda}_d [\,\underline{V}_d^* \,|\, 0\,] + [\,0 \,|\, \delta \underline{U}_d \underline{\Lambda}_d \underline{\alpha}\,].$$

This decomposition shows that if the noise $\|q_d\| = \|\underline{\alpha}\|_{\ell^2}$ is small, then $\mathcal{T}_\delta$ is well-approximated by the first two terms (the noiseless part) that expose how the left and right singular subspaces of $\mathcal{T}$, relative to the partition, align with those of $\mathcal{T}_\delta$. If the singular value gap between the first and second terms

$$\gamma_d(\delta) = \left\| [\overline{\Lambda}_d (I + \delta^2 \overline{\alpha}\, \overline{\alpha}^\top)^{1/2}]^{-1} \right\|_{\mathrm{op}}^{-1} - \|\underline{\Lambda}_d\|_{\mathrm{op}} \quad (8)$$

is positive, the dominant singular subspace of the noiseless perturbation of $\mathcal{T}$ is exactly $\mathcal{R}(\overline{U}_d)$. Therefore, by carefully applying perturbation results, we are able to control the differences in orthogonal projections

$$\left\| \Pi_{\overline{U}_d} - \Pi_{\Psi_{\widehat{\theta}_m}^{(d)}} \right\|_{\mathrm{op}} \leq \left\| \Pi_{\overline{U}_d} - \Pi_{\Psi_*^{(d)}} \right\|_{\mathrm{op}}$$
$$+ \left\| \Pi_{\Psi_*^{(d)}} - \Pi_{\Psi_{\widehat{\theta}_m}^{(d)}} \right\|_{\mathrm{op}}$$

to align the appropriate left subspaces. The first term depends on the interplay between the parameter $\delta$, the positioning, and the size of the signal, and is bounded by $\delta \|q_d\|_{L_2(X)} / \gamma_d(\delta)$. The second term is the error from learning the features of $\mathcal{T}_\delta$ from dataset $\tilde{\mathcal{D}}_m$ with our representation learning method, quantified by the optimality gap $\mathcal{E}_d(\widehat{\theta}_m, \widehat{\omega}_m, \delta)$ given in Eq. (7). A similar decomposition holds for the right subspaces. By bounding these components, we can control the total approximation error. A detailed proof is provided in Theorem C.6 in Section C.3; in the following, we present the consequences for the "good" and "bad" scenarios from Meunier et al. (2026).

**Learning in the "good" scenario.** When the signal $s_d$ is spanned by the top-$d$ singular functions of $\mathcal{T}$ (i.e., $\overline{N} = \{1, \ldots, d\}$), our method works efficiently even with $\delta = 0$. In this case, the spectral gap $\gamma_d(0) = \lambda_d - \lambda_{d+1}$ is positive by assumption. A direct corollary of Theorems 5.1, and C.6 yields

$$\|h_0 - \widehat{h}_\theta\|_{L_2} \lesssim \|q_d\|_{L_2} + \frac{\mathcal{E}_d(\widehat{\theta}_m, \widehat{\omega}_m, 0)}{\lambda_d} + \frac{1}{\lambda_d}\sqrt{\frac{d}{n}} \log \tau^{-1},$$

with probability at least $1 - \tau$. This result is the high-probability version of Meunier et al. (2026, Theorem 3).

**Learning in the "bad" scenario.** Our outcome-aware method is designed to succeed even when the structural

function $h_0$ is highly aligned with a "bad" singular function $v_k$ (for large $k$). Meunier et al. (2026) showed that standard SpecIV would require learning a high-dimensional feature space of dimension $d \geq k$ (out of which $k - 1$ are spurious). In contrast, our method can isolate the relevant signal by setting $d = 1$ with the signal concentrating mostly on the space defined by $v_k$ ($\overline{N} = \{k\}$). As shown in Section C.3, choosing a $\delta$ large enough guarantees that the one-dimensional spectral gap $\gamma_1(\delta)$ becomes positive. Applying Theorem C.6 of Section C.3 in this setting shows:

$$\|h_0 - \widehat{h}_\theta\|_{L_2} \lesssim \frac{1}{\lambda_k^2} \frac{\|q_1\|_{L_2}}{\|s_1\|_{L_2}} + \frac{\mathcal{E}_1(\widehat{\theta}_m, \widehat{\omega}_m, \delta)}{\lambda_k} + \frac{\log \tau^{-1}}{\lambda_k \sqrt{n}},$$

with probability at least $1 - \tau$, indicating that whenever signal-to-noise ratio dominates the decay $\|s_1\|_{L_2(X)} / \|q_1\|_{L_2(X)} \gg \lambda_k^{-2}$, our method can recover the structural function with one feature, while for the standard SpecIV one would need to learn $k$ of them, as illustrated in Figure 1.

The guarantees above should be read as conditional on successful representation learning. In particular, the downstream 2SLS error is small when the learned features have small population optimality gap $\mathcal{E}_d(\widehat{\theta}_m, \widehat{\omega}_m, \delta)$, together with the remaining approximation and sampling terms. Proposition A.4 in the Appendix shows that this operator gap is controlled by the population excess augmented loss

$$\Delta_\delta(\theta, \omega) \doteq \mathcal{L}_\delta^{(d)}(\theta, \omega) + \left\| \mathcal{T}_\delta^{(d)} \right\|_{\mathrm{HS}}^2$$

If $\sigma_{*,d} > \sigma_{*,d+1}$, then

$$\mathcal{E}_d(\theta, \omega, \delta) \leq (1 - \rho_{\delta,d})^{-1/2} \sqrt{\Delta_\delta(\theta, \omega)},$$

where $\rho_{\delta,d} \doteq \sigma_{*,d+1}/\sigma_{*,d}$. This connection between the population contrastive excess loss and the operator optimality gap is not specific to the augmented operator: taking $\delta = 0$ yields the corresponding result for standard SpecIV (Meunier et al., 2026), and to our knowledge, this explicit loss-to-operator-gap bound is new even in that setting. Proving that empirical DNN training drives $\Delta_\delta$ to zero would require an architecture-specific analysis of the spectral contrastive objective's optimization and generalization, which we leave open.

## 6. Experiments

We first illustrate the utility of our method on a synthetic example in which all parameters of $\mathcal{T}$ are controlled. To further support our claims, we benchmark outcome-aware spectral learning on two challenging benchmarks based on the dSprites dataset (Matthey et al., 2017). Finally, we include an Off-Policy Evaluation (OPE) experiment in the context of reinforcement learning, which demonstrates that the method remains robust and competitive in demanding

environments. We find that a small positive $\delta$ always leads to improved performance. In most challenging cases, the resulting method outperforms standard SpecIV ($\delta = 0$) by a wide margin.

## 6.1. Synthetic data

Following Meunier et al. (2026), we generate a dataset of tuples $(Z, X, Y)$ from a conditional expectation operator $\mathcal{T} = \mathbb{1}_Z \otimes \mathbb{1}_X + \sum_{i=1}^{d-1} \sigma_i u_i \otimes v_i$ with explicit $\sigma_i, v_i, u_i$. We take $\sigma_i$ to decay linearly from a fixed $\sigma_1$ to $\sigma_{d-1} = c_\sigma \sigma_1$, with $c_\sigma \in [0, 1]$. The families $(u_i), (v_i)$ are random orthonormal bases of the span of $(\sin(\ell \cdot x))_{\ell=1}^{d-1}$ on $[-\pi, \pi]$. By setting the structural function $h_0 = \sum_{i=1}^{d-1} \alpha_i v_i$ with the constraint $\|\alpha\|_{\ell_2} = 1$, and having the coefficients $\alpha_i$ change linearly in $i$, we are able to control the alignment of $h_0$ with the spectrum of $\mathcal{T}$. We parametrize the rate at which $\alpha_i$ change with $c_\alpha \doteq \alpha_{d-1}/\alpha_1$.

**Synthetic data results.** Figure 2 shows the distribution $\|\hat{h}_\theta - h_0\|^2$ for different values of $\delta$, normalized so that for each $c_\alpha$, the mean loss for $\delta = 0$ equals 1. The reported figures show how the losses change relative to the baseline performance of SpecIV. In cases of very poor spectral alignment, corresponding to $c_\alpha = 5.0$, increasing $\delta$ leads to improvement in the IV regression loss even when the singular values decay quickly. Moreover, when $c_\sigma$ is sufficiently large, that is more singular functions can be learned easily, we observe improvement even on a very well-aligned case when $c_\alpha = 0.2$. A more extensive evaluation of how the benefit of using $\delta$ changes with the rate of decay of singular values can be found in Section E.3.

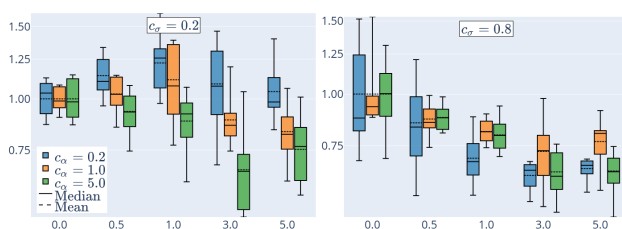

*Figure 2.* Distributions of relative IV regression MSEs ($\|\hat{h}_\theta - h_0\|^2$) for the synthetic example with $\delta \in \{0, 0.5, 1.0, 3.0, 5.0\}$ and $c_\sigma \in \{0.2, 0.8\}$

## 6.2. dSprites data

The original dSprites dataset of Matthey et al. (2017) consists of $64 \times 64$ noisy images (sprites) of hearts, squares, and ellipses with varying position, size, and orientation. In the standard IV benchmarking setting, as described in, e.g., Sun et al. (2025), only the heart images are used. The structural function takes the form

$$h_0(x) = (\|A \circ X\|^2 - 3000)/500,$$

where $A_{ij} = |32 - j|/32$ measures the distance from the central vertical bar in the image and $\circ$ denotes the pointwise (Hadamard) product. The instrumental variable is defined as $Z = $ (sprite orientation, sprite $x$-position, sprite scale), and the outcome is

$$Y = h_0(X) + 32((\text{sprite } y\text{-position}) - 32) + \varepsilon,$$

$\varepsilon \sim \mathcal{N}(0, 0.5)$. For reasons discussed below, we shall refer to this $h_0$ as $h_{\text{old}}$.

**New structural function.** As first argued in Meunier et al. (2026) and further discussed in Section E.1, the standard dSprites benchmark is an instance of the "good" case where $h_0$ is well-aligned with the leading singular functions of $\mathcal{T}$. Hence, we propose an alternative, more challenging structural function. We refer to the new function as $h_{\text{new}}$. It is based on sprite images of ellipses, as opposed to the hearts in the original case, and approximates the ellipse's orientation, which we expect to be a property that is only recovered from singular functions associated to small singular values of $\mathcal{T}$. The details of our argument and construction can be found in Section E.1.

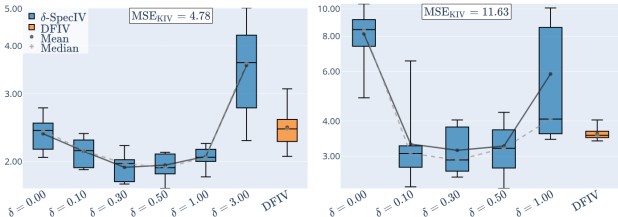

*Figure 3.* Distribution of $\|\hat{h}_\theta - h_0\|^2$ on $h_{\text{old}}$ (**left**) and $h_{\text{new}}$ (**right**) evaluated for a range of $\delta$ values, compared to those attained by DFIV and KIV (Singh et al., 2019). Our method is evaluated on 9 independently fitted models with identical hyperparameters for each $x$-axis value. 16 DFIV models were fitted in both settings.

**Experiment results.** We compare our method to DFIV (Xu et al., 2021), which is currently the most competitive method to benchmark against. In the setting of $h_{\text{old}}$ we observe an average 20% improvement from selecting a small positive $\delta$ over using standard SpecIV. The standard benchmark is well-aligned but there is still some benefit in this setting. By comparison, for $h_{\text{new}}$, we see that vanilla spectral learning ($\delta = 0$) severely underperforms DFIV. This is in line with the fact that the top of the eigenspectrum of $\mathcal{T}$ is not an optimal set of features for the downstream problem. Increasing $\delta$ allows the model to tune the learned features, increasing the projection of $h_{\text{new}}$ onto their span, and hence match or slightly exceed the performance of DFIV. Since good performance is contingent on the choice of $\delta$, we investigate strategies for $\delta$ selection in Section 6.4.

## 6.3. Off-Policy Evaluation

To further illustrate the applicability of our augmented method, we showcase its performance in Off-Policy Evalua-

tion (OPE). OPE is a fundamental problem in reinforcement learning that can be approached via NPIV (Chen et al., 2022), but poses a significant challenge for standard SpecIV because it involves a non-compact operator that cannot be learned through the usual contrastive framework. Instead, one has to learn another operator amenable to spectral feature learning while iteratively updating the outcome $Y$, which SpecIV is agnostic to. Following the experimental pipeline of Chen et al. (2022), we showcase scenarios where our proposed modification significantly improves SpecIV and performs comparably well to state-of-the-art methods for NPIV applied to OPE (Xu et al., 2021; Chen et al., 2022; Petrulionyte et al., 2024), suggesting that our method broadens the applicability of spectral feature-based NPIV for OPE.

**OPE context.** The goal of OPE is to estimate the value of a *known* target policy, $\pi(a|s)$, where $a$ denotes an action and $s$ denotes a state. The challenge is that we only have access to a fixed, "offline" dataset of transitions, $\mathcal{D} = \{(s_i, a_i, r_i, s_i')\}_i$, where $r$ denotes a reward, and $s'$ a next state transition. The dataset could have been collected by one or a mixture of potentially unknown policies of potentially unknown analytical form, denoted by $\pi_b(a|s)$. This problem is central to offline reinforcement learning, as it allows one to select the best-performing policy among a collection of candidate policies without direct interaction with the environment, which can be costly or unethical, as is often the case, e.g., in healthcare (Gottesman et al., 2018). For a detailed overview of OPE, see the work of Levine et al. (2020).

**OPE via NPIV.** The standard objective in OPE is to estimate the Q-function $Q_\pi$ of a policy $\pi$ (uniformly well or in a suitable $L_2$ norm). Xu et al. (2021) showed that $Q$-function estimation can be framed as a NPIV problem (see Section E.7.2). However, their formulation is not amenable to spectral feature learning as the conditional expectation operator they obtain is non-compact (Chen et al., 2022, Eq. (19)). To overcome this, we instead recover $Q_\pi$ through a modified NPIV problem where $X = (s', a')$ and $Z = (s, a)$ are such that $a' \sim \pi(\cdot \mid s')$ and $a \sim \pi_b(\cdot \mid s)$. Writing $\mathcal{T}q(Z) \doteq \mathbb{E}[q(X) \mid Z] = \mathbb{E}[q(s', a') \mid s, a]$, $Q_\pi$ is then identified as the solution to $\mathcal{T}Q_\pi = \mathbb{E}[Y(Q_\pi) \mid Z]$, where $Y(Q_\pi) = -\gamma^{-1}(R - Q_\pi(s, a))$, where $R \mid Z$ is the random reward associated to $Z = (s, a)$, and $\gamma$ is the discount factor. As the outcome $Y(Q_\pi)$ depends on $Q_\pi$, which is the very function we are estimating, we introduce an iterative procedure to estimate $Q_\pi$. We start with a random guess $Q_0$ (e.g., $Q_0 = 0$) and build $Y_0 = Y(Q_0)$. We then estimate $Q_1$ with Augmented Spec IV. We repeat this process for $K$ steps. We defer the details of the iterative procedure and the derivation of the new NPIV problem for OPE to Section E.7.2. The key point is that the iterative process introduces a potential for dynamic spectral misalignment,

as the target $Y_k$ changes at every iteration. If the spectral features required to estimate $Y_k$ are not the same as the dominant spectral features of $\mathcal{T}$, or if this direction shifts as $Q_k$ converges, an outcome-agnostic method will fail. This is precisely the scenario where Augmented SpecIV can help.

**Experiment results.** We follow the experimental setting of Chen et al. (2022). In particular, we evaluate the performance of DFIV, SpecIV ($\delta = 0$), and Augmented SpecIV (AugSpecIV; $\delta > 0$) on estimating the value of policies learned by Deep Q-Networks (Mnih et al., 2015) across randomized versions of the BSuite Cartpole (Barto et al., 1983), Mountain Car (Moore, 1990), and Catch environments. The OPE datasets are the pure offline versions from Chen et al. (2022), containing approximately $n = 700k/150k/20k$ (Cartpole/Mountain Car/Catch) transition tuples $(s, a, r, s')$. The results are shown in Section E.7, Figures 9 to 11. Compared with Chen et al. (2022), DFIV performed slightly worse, likely due to randomness in hyperparameter sampling. SpecIV and AugSpecIV both achieved strong performance on Catch but struggled on Mountain Car. Consistent with prior findings (Chen et al., 2022), no OPE method performed uniformly well across all tasks. In our experiments, DFIV performed poorly on Cartpole but well on Mountain Car, whereas AugSpecIV showed the opposite trend. SpecIV also underperformed on Cartpole, likely due to spectral misalignment. Since $\delta$ was automatically tuned as a hyperparameter, the selected values were $\delta = 1$ for Cartpole, $\delta = 10^{-3}$ for Mountain Car, and $\delta = 10^{-2}$ for Catch. These results suggest that our approach can adapt to the underlying spectral alignment structure.

### 6.4. Selecting $\delta$

We discuss three approaches to select $\delta$.

**Estimating alignment with $h_0$.** By Proposition 4.1, our strategy targets features $\varphi_\star^{(d)}$ from the SVD of $\mathcal{T}_\delta$ as a basis in which to learn $h_0$ (see Eq. (6)). The next result shows that we can estimate the length of the projection of $h_0$ onto the span of these features.

**Proposition 6.1.** *For any $i = 1, \ldots, d$, denote $\alpha_i = \mathbb{E}[Y\psi_{\star,i}(Z)]\sigma_{\star,i}^{-1}$. Then, the following holds*

$$\|\Pi_{\varphi_\star^{(d)}} h_0\|_{L_2(X)}^2 = \alpha^\top \left(I_d - \omega_\star \omega_\star^\top\right)^{-1} \alpha.$$

We can construct an estimator of this projection length that uses the fitted $\widehat{\mathcal{T}}_\delta$ rather than the true operator. Having learned the operator, we can approximate its SVD using a $(Z, X)$ dataset. We do so by decomposing the operator into features that are orthonormal with respect to the $L_2(\widehat{\mu}_X) \times \mathbb{R}$ and $L_2(\widehat{\mu}_Z)$ inner products, where $\widehat{\mu}$ denotes empirical distributions based on the samples. This procedure yields $\widehat{\mathcal{T}}_\delta = \sum_{i=1}^d \widehat{\sigma}_i \widehat{\psi}_i \otimes (\widehat{\varphi}_i, \widehat{\omega}_i)$ where $(\widehat{\psi}_i)_{i=1}^d$ and $(\widehat{\varphi}_i)_{i=1}^d$ are orthonormal systems with respect to the

aforementioned empirical inner products. For the plug-in estimator $\|\Pi_{\varphi_\star^{(d)}} h_0\|_{L_2(X)}^2 \approx \widehat{\alpha}^\top \left(I_d - \widehat{\omega}\widehat{\omega}^\top\right)^{-1} \widehat{\alpha}$, which shows how well our features approximate the true underlying function and can be used to select $\delta$, in Figure 4, we observe that the estimated spectral alignment increases very rapidly with small $\delta$ and stays roughly constant for all the choices of $\delta$ that lead to good results in IV regression. The exception is $\delta = 1.0$ where our IV performance has already somewhat degraded, but the estimated alignment remains high.

As $\delta$ increases, so does the norm of the optimal $\omega_*$, which may lead to unstable estimation of $(I_d - \omega_*\omega_*^\top)^{-1}$. Therefore, we suggest accepting a larger value of $\delta$ only if it leads to a substantial increase in the estimated alignment between $h_0$ and the learned features. We investigate this further in Section E.9.

**Balancing the terms in the loss.** Given that our ability to learn the actual truncated SVD of $\mathcal{T}_\delta$ hinges on the convergence of the feature neural networks to the population-level optimum, the optimal choice of $\delta$ is in part an empirical challenge. In particular, consider what happens if $\delta$ is sufficiently large that $\mathcal{R}_\delta^{(d)}$ dominates the joint loss value (Eq. (4)). The features $\psi_\theta$ minimizing $\mathcal{R}_\delta^{(d)}$ are non-unique. Any choice such that $r_0$ lies in their span is equally good. Therefore, the rest of the $Z$-features should be used to learn an approximation of the conditional expectation. However, if the gradients with respect to $\mathcal{R}_\delta^{(d)}$ are too large, they effectively drown out the signal needed to learn $\mathcal{T}$ and cause a significant decrease in $\mathcal{L}_0^{(d)}$.

A useful heuristic for selecting $\delta$ is to treat $\mathcal{R}_\delta^{(d)}$ as an additional regularization on the learned features, and tune its strength so that it influences the learned features without leading the models to neglect the original $\mathcal{L}_0^{(d)}$ term. A heuristic, which we find leads to good results, is to increase $\delta$ as long as doing so leads to substantial decreases in $\mathcal{R}_\delta^{(d)}$ with only minor changes in $\mathcal{L}_0^{(d)}$. Small values of $\delta$ are always observed to lead to improved spectral alignment. As long as increasing $\delta$ remains "free" in that our ability to approximate $\mathcal{T}$, as measured by $\mathcal{L}_0^{(d)}$, does not change by much, we continue to increase it. As seen in Figure 4, the $\mathcal{R}_\delta^{(d)}$ term eventually becomes dominant and $\mathcal{L}_0^{(d)}$ increases substantially. In our experiments, those settings correspond to parameters that yield poor results in IV regression. We also note that the method's performance is not sensitive to minor changes in $\delta$. We see in Figures 2 and 3 that all sufficiently small values of $\delta$ lead to improvement.

**Minimization of second stage loss.** A natural approach to selecting an optimal IV model is to pick one which yields the smallest 2SLS error (Xu et al., 2021; Chen et al., 2022). We find that it works relatively well in experiments on dSprites and off-policy evaluation (where it is the default model selection strategy employed in standard IV-based benchmarks).

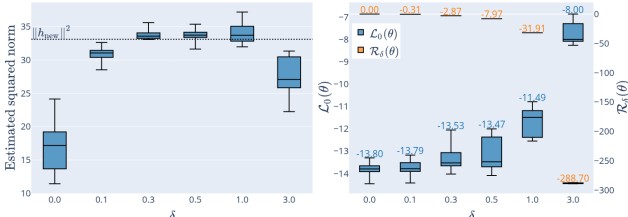

*Figure 4.* **Left:** Estimation of $\|\Pi_{\varphi_\star^{(d)}} h_{\text{new}}\|^2$ for a range of $\delta$. **Right:** Evolution of $\mathcal{L}_0^{(d)}$ and $\mathcal{R}_\delta^{(d)}$ for models learning $h_{\text{new}}$. Models with non-zero $\delta$ and a small $\mathcal{L}_0^{(d)}$ (close to the value attained at $\delta = 0$) demonstrate the best results. Each bar's mean value is noted above it.

However, as we discuss in Section E.8, the theoretical justification of this method's consistency remains elusive.

# 7. Conclusion

We have proposed ***Augmented Spectral Feature Learning***, a new framework for outcome-aware feature learning in nonparametric instrumental variable regression. By introducing an augmented operator and a contrastive loss, our method addresses the fundamental limitation of outcome-agnostic spectral features and achieves robustness in regimes with spectral misalignment to the targeted structural function. Our current approach relies on a rank-one augmentation; extending the framework to richer, higher-rank perturbations remains a promising direction for future research.

# Impact Statement

This paper contributes to the methodology of nonparametric instrumental variable regression, with the goal of improving causal effect estimation in settings with hidden confounding. There are many potential societal consequences of our work, none of which we feel must be specifically highlighted here.

# Acknowledgments and Disclosure of Funding

D.M., J.W. and A.G. are supported by the Gatsby Charitable Foundation. V.R.K. and M.P. were supported by NextGenerationEU and MUR PNRR project PE0000013 CUP J53C22003010006 "Future Artificial Intelligence Research (FAIR)". A.M. is funded via a European Research Council (ERC) project, under the European Union's Horizon 2020 research and innovation programme (Grant agreement No. 950180). This project received funding from the European Union's Horizon Europe research and innovation program under grant agreement 101120237 (ELIAS) and from ELSA - European Lighthouse on Secure and Safe AI funded by the European Union under Grant Agreement No. 101070617.

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

# A. Technical Tools

**Notation.** For a compact operator $A$, $\sigma_d(A)$ denotes the $d$-th largest singular value of $A$.

**Proposition A.1** (Weyl's inequality). *For any two compact operators $A$ and $B$, the singular values are stable under perturbation:* $|\sigma_i(A) - \sigma_i(B)| \leq \|A - B\|_{\mathrm{op}}$, $i \leq \min\{\mathrm{rank}(A), \mathrm{rank}(B)\}$.

**Theorem A.2** (Wedin sin-$\Theta$ Theorem). *Let $A$ and $B$ be compact operators. Let $P_A$ and $P_B$ be the orthogonal projections onto the subspaces spanned by the top-$d$ left singular vectors of $A$ and $B$ respectively (the same property holds with the right singular vectors). If $\gamma \doteq \sigma_d(A) - \sigma_{d+1}(B) > 0$, then,*

$$\|P_A - P_B\|_{\mathrm{op}} \leq \frac{\|A - B\|_{\mathrm{op}}}{\gamma}.$$

*Additionally, if $\gamma_A \doteq \sigma_d(A) - \sigma_{d+1}(A) > 0$ and $\|A - B\|_{\mathrm{op}} \leq \gamma_A/2$, then*

$$\|P_A - P_B\|_{\mathrm{op}} \leq 2 \cdot \frac{\|A - B\|_{\mathrm{op}}}{\gamma_A}.$$

*Proof.* The first inequality is the original theorem (Wedin, 1972). We prove the second inequality. By Weyl's inequality, Proposition A.1,

$$|\sigma_{d+1}(A) - \sigma_{d+1}(B)| \leq \|A - B\|_{\mathrm{op}}.$$

Hence,

$$\gamma \geq \sigma_d(A) - \sigma_{d+1}(A) - \|A - B\|_{\mathrm{op}} = \gamma_A - \|A - B\|_{\mathrm{op}} \geq \gamma_A/2.$$

Therefore,

$$\|P_A - P_B\|_{\mathrm{op}} \leq 2 \cdot \frac{\|A - B\|_{\mathrm{op}}}{\gamma_A}.$$

$\square$

Let $A : \mathcal{H}_1 \to \mathcal{H}_2$ be a compact operator between Hilbert spaces with Singular Value Decomposition

$$A = \sum_{i \geq 1} \sigma_i u_i \otimes v_i,$$

where $\{v_i\} \subset \mathcal{H}_1$ and $\{u_i\} \subset \mathcal{H}_2$ are orthonormal families of singular vectors, and $(\sigma_i)_i$ are the singular values of $A$, which satisfy $\sigma_1 \geq \sigma_2 \geq \cdots > 0$. For all $d \geq 1$, $A_d \doteq \sum_{i=1}^d \sigma_i u_i \otimes v_i$ denotes its rank-$d$ truncated SVD. The following theorem states that $A_d$ is a best rank-$d$ approximation of $A$, and is the unique best approximation if $\sigma_d > \sigma_{d+1}$.

**Theorem A.3** (Eckart-Young-Mirsky Theorem).

$$\|A - A_d\| = \min_{B:\mathrm{rank}(B) \leq d} \|A - B\|.$$

*The result holds for any unitarily invariant norm, in particular for the operator and Hilbert-Schmidt norms. Moreover, if $\sigma_d > \sigma_{d+1}$, $A_d$ is the unique minimizer in the Hilbert-Schmidt norm.*

The next proposition allows us to control the difference between the rank-$d$ truncated SVD and any other rank-$d$ operator, in terms of the excess Hilbert-Schmidt loss. The result is of independent interest.

**Proposition A.4** (Excess-loss to operator-gap bound). *Let $A$ be such that $\sigma_d > \sigma_{d+1}$. Define $\rho \doteq \sigma_{d+1}/\sigma_d \in [0, 1)$. Let $B$ be any operator of rank at most $d$, and define the excess Hilbert-Schmidt loss of $B$ as*

$$\Delta(B) \doteq \|A - B\|_{\mathrm{HS}}^2 - \|A - A_d\|_{\mathrm{HS}}^2 = \|A_d - B\|_{\mathrm{HS}}^2 - 2\langle B, A - A_d \rangle_{\mathrm{HS}}.$$

*Then*

$$\|A_d - B\|_{\mathrm{op}} \leq \|A_d - B\|_{\mathrm{HS}} \leq \frac{1}{\sqrt{1 - \rho}} \sqrt{\Delta(B)}.$$

*This result applied to $A = \mathcal{T}_\delta$, if $\sigma_{*,d} > \sigma_{*,d+1}$ implies*

$$\mathcal{E}_d(\theta, \omega, \delta) \leq (1 - \rho_{\delta,d})^{-1/2} \sqrt{\Delta_\delta(\theta, \omega)},$$

*with $\rho_{\delta,d} \doteq \sigma_{*,d+1}/\sigma_{*,d}$.*

*Proof.* Let $R \doteq A - A_d = \sum_{j>d} \sigma_j u_j \otimes v_j$. If $\rho = 0$, then $\sigma_{d+1} = 0$, hence $R = 0$ and $A = A_d$. Therefore

$$\Delta(B) = \|A_d - B\|_{\mathrm{HS}}^2,$$

and the claim follows from $\|A_d - B\|_{\mathrm{op}} \leq \|A_d - B\|_{\mathrm{HS}}$. Assume now that $\rho > 0$, and define the rescaled operator

$$\widetilde{A} \doteq A_d + \rho^{-1}R.$$

The singular values of $\widetilde{A}$ are

$$\sigma_1, \ldots, \sigma_d, \rho^{-1}\sigma_{d+1}, \rho^{-1}\sigma_{d+2}, \ldots.$$

For every $j > d$, $\rho^{-1}\sigma_j \leq \rho^{-1}\sigma_{d+1} = \sigma_d$. Thus $A_d$ is a best rank-$d$ approximation of $\widetilde{A}$ in Hilbert–Schmidt norm by Theorem A.3. Since $\mathrm{rank}(B) \leq d$,

$$0 \leq \left\|\widetilde{A} - B\right\|_{\mathrm{HS}}^2 - \left\|\widetilde{A} - A_d\right\|_{\mathrm{HS}}^2 = \left\|A_d - B + \rho^{-1}R\right\|_{\mathrm{HS}}^2 - \left\|\rho^{-1}R\right\|_{\mathrm{HS}}^2.$$

Expanding the square gives

$$0 \leq \|A_d - B\|_{\mathrm{HS}}^2 + 2\rho^{-1}\langle A_d - B, R\rangle_{\mathrm{HS}} = \|A_d - B\|_{\mathrm{HS}}^2 - 2\rho^{-1}\langle B, R\rangle_{\mathrm{HS}}.$$

Hence

$$2\langle B, R\rangle_{\mathrm{HS}} \leq \rho \|A_d - B\|_{\mathrm{HS}}^2.$$

On the other hand,

$$\begin{aligned}
\Delta(B) &= \|A - B\|_{\mathrm{HS}}^2 - \|A - A_d\|_{\mathrm{HS}}^2 \\
&= \|A_d - B + R\|_{\mathrm{HS}}^2 - \|R\|_{\mathrm{HS}}^2 \\
&= \|A_d - B\|_{\mathrm{HS}}^2 + 2\langle A_d - B, R\rangle_{\mathrm{HS}} \\
&= \|A_d - B\|_{\mathrm{HS}}^2 - 2\langle B, R\rangle_{\mathrm{HS}}.
\end{aligned}$$

Combining the last two displays yields

$$\Delta(B) \geq (1 - \rho)\|A_d - B\|_{\mathrm{HS}}^2.$$

Therefore

$$\|A_d - B\|_{\mathrm{op}} \leq \|A_d - B\|_{\mathrm{HS}} \leq \frac{1}{\sqrt{1-\rho}}\sqrt{\Delta(B)}.$$

For the final claim, take $A = \mathcal{T}_\delta$, $A_d = \mathcal{T}_\delta^{(d)}$, and $B = \Psi_\theta^{(d)}[\Phi_\theta^{(d)*} \,|\, \omega]$. Then $\Delta(B) = \Delta_\delta(\theta, \omega)$ and the displayed bound gives the stated inequality for $\mathcal{E}_d(\theta, \omega, \delta)$. $\qquad\square$

## B. Well-definedness of Eq. (3) and role of compactness

We review results from Kostic et al. (2024); Meunier et al. (2026). The conditional expectation operator $\mathcal{T}$ is a bounded linear operator satisfying $\|\mathcal{T}\|_{\mathrm{op}} \leq 1$. Parametrized feature maps produce a finite-rank operator $\mathcal{T}_\theta^{(d)} : L_2(X) \to L_2(Z)$ of rank at most $d$:

$$\mathcal{T}_\theta^{(d)} \doteq \sum_{i=1}^d \psi_{\theta,i}^{(d)} \otimes \varphi_{\theta,i}^{(d)},$$

where $\varphi_\theta^{(d)} : \mathcal{X} \to \mathbb{R}^d$, $\psi_\theta^{(d)} : \mathcal{Z} \to \mathbb{R}^d$, and for $u \in L_2(Z), v \in L_2(X)$ we use the rank-one notation

$$(u \otimes v)h \doteq u \langle v, h\rangle_{L_2(X)} = u \times \mathbb{E}[v(X)h(X)].$$

By construction, $\mathcal{T}_\theta^{(d)}$ is Hilbert–Schmidt (indeed finite-rank), hence $\left\|\mathcal{T}_\theta^{(d)}\right\|_{\mathrm{HS}} < \infty$. It is shown in Kostic et al. (2024); Meunier et al. (2026) that

$$\mathcal{L}_0^{(d)}(\theta) = \left\|\mathcal{T}_\theta^{(d)}\right\|_{\mathrm{HS}}^2 - 2\,\mathrm{Tr}((\mathcal{T}_\theta^{(d)})^*\mathcal{T}).$$

As $\mathrm{Tr}(A^*B)$ is well-defined whenever $A$ is finite-rank and $B$ is bounded, the loss is always well-defined. Compactness is invoked in our analysis only to guarantee the existence of a singular system $\{(\lambda_j, u_j, v_j)\}_{j \geq 1}$ and thus a well-defined truncated SVD

$$\mathcal{T}_d \doteq \sum_{j=1}^d \lambda_j \, u_j \otimes v_j.$$

# C. Omitted Proofs

We provide the proofs omitted from the main text below.

## C.1. Proof of Proposition 4.1

Given $d \geq 1$, $\delta \geq 0$, parameter $\theta$ and $\omega \in \mathbb{R}^d$, define the operator

$$\mathcal{T}_{\theta,\omega} = \Psi_\theta^{(d)}[\Phi_\theta^{(d)\star} \mid \omega] = \sum_{i=1}^d \psi_{\theta,i}^{(d)} \otimes (\varphi_{\theta,i}^{(d)}, \omega_i) : L_2(X) \times \mathbb{R} \to L_2(Z).$$

We show that

$$\mathcal{L}_\delta(\theta, \omega) = \|\mathcal{T}_\delta - \mathcal{T}_{\theta,\omega}\|_{\mathrm{HS}}^2 - \|\mathcal{T}_\delta\|_{\mathrm{HS}}^2.$$

To simplify the notations in this proof, we drop the $(d)$ subscript on the features. Let us define $\tilde{\Phi}_{\theta,\omega} : \mathbb{R}^d \to L_2(X) \times \mathbb{R}$ the operator whose adjoint is $\tilde{\Phi}_{\theta,\omega}^* = [\Phi_\theta^* \mid \omega] : L_2(X) \times \mathbb{R} \to \mathbb{R}^d$ such that we have $\mathcal{T}_{\theta,\omega} = \Psi_\theta \tilde{\Phi}_{\theta,\omega}^*$. $\tilde{\Phi}_{\theta,\omega}$ is such that

$$\tilde{\Phi}_{\theta,\omega}\beta = \sum_{i=1}^d \beta_i(\varphi_{\theta,i}, \omega_i), \qquad \tilde{\Phi}_{\theta,\omega}^*(h, a) = \Phi_\theta^* h + a \cdot \omega.$$

Therefore,

$$\tilde{\Phi}_{\theta,\omega}^\star \tilde{\Phi}_{\theta,\omega} = \mathbb{E}[\varphi_\theta(X)\varphi_\theta(X)^\intercal] + \omega\omega^\intercal = C_{\varphi_\theta} + \omega\omega^\intercal \tag{9}$$

Using the definition of the Hilbert-Schmidt norm and the cyclic property of the trace, we have

$$\|\mathcal{T}_\delta - \mathcal{T}_{\theta,\omega}\|_{\mathrm{HS}}^2 - \|\mathcal{T}_\delta\|_{\mathrm{HS}}^2 = \|\mathcal{T}_{\theta,\omega}\|_{\mathrm{HS}}^2 - 2\,\mathrm{Tr}\left(\mathcal{T}_{\theta,\omega}^\star \mathcal{T}_\delta\right) = \mathrm{Tr}\left(\Psi_\theta^* \Psi_\theta \tilde{\Phi}_{\theta,\omega}^* \tilde{\Phi}_{\theta,\omega}\right) - 2\,\mathrm{Tr}\left(\Psi_\theta^* \mathcal{T}_\delta \tilde{\Phi}_{\theta,\omega}\right).$$

For the first term, exploiting Eq. (9) and the linearity of the trace, we have

$$\mathrm{Tr}\left(\Psi_\theta^* \Psi_\theta \tilde{\Phi}_{\theta,\omega}^* \tilde{\Phi}_{\theta,\omega}\right) = \mathrm{Tr}\left(C_{\psi_\theta} C_{\varphi_\theta} + C_{\psi_\theta}\omega\omega^\intercal\right)$$
$$= \mathrm{Tr}(C_{\psi_\theta} C_{\varphi_\theta}) + \omega^\intercal C_{\psi_\theta}\omega$$
$$= \mathbb{E}_X \mathbb{E}_Z[(\varphi_\theta(X)^\intercal \psi_\theta(Z))^2] + \omega^\intercal C_{\psi_\theta}\omega,$$

where $\mathbb{E}_X \mathbb{E}_Z$ denotes the expectation where $X$ and $Z$ are treated as independent random variables drawn from their respective marginal distributions.

For the second term, recall that $\mathcal{T}_\delta = [\mathcal{T} \mid \delta r_0]$. For all $\beta \in \mathbb{R}^d$, we have

$$[\Psi_\theta^* \mathcal{T}_\delta \tilde{\Phi}_{\theta,\omega}\beta]_j = \sum_{i=1}^d \beta_i[\Psi_\theta^* \mathcal{T}_\delta(\varphi_{\theta,i}, \omega_i)]_j$$
$$= \sum_{i=1}^d \beta_i[\Psi_\theta^*(\omega_i \cdot \delta \cdot r_0 + \mathcal{T}\varphi_{\theta,i})]_j$$
$$= \sum_{i=1}^d \beta_i\left(\omega_i \cdot \delta \cdot \mathbb{E}[Y\psi_{\theta,j}(Z)] + \mathbb{E}[\varphi_{\theta,i}(X)\psi_{\theta,j}(Z)]\right).$$

Therefore $\Psi_\theta^* \mathcal{T}_\delta \tilde{\Phi}_{\theta,\omega}\beta = \delta \cdot \mathbb{E}[Y\psi_\theta(Z)]\omega^\intercal + \mathbb{E}[\psi_\theta(Z)\varphi_\theta(X)^\intercal]$, and

$$\mathrm{Tr}\left(\Psi_\theta^* \mathcal{T}_\delta \tilde{\Phi}_{\theta,\omega}\right) = \mathbb{E}[\varphi_\theta(X)^\intercal \psi_\theta(Z)] + \delta \cdot \omega^\intercal \mathbb{E}[Y\psi_\theta(Z)].$$

Putting it together, we obtain

$$\|\mathcal{T}_\delta - \mathcal{T}_{\theta,\omega}\|_{\mathrm{HS}}^2 - \|\mathcal{T}_\delta\|_{\mathrm{HS}}^2 = \mathbb{E}_X \mathbb{E}_Z[(\varphi_\theta(X)^\intercal \psi_\theta(Z))^2] + \omega^\intercal C_{\psi_\theta}\omega - 2\mathbb{E}[\varphi_\theta(X)^\intercal \psi_\theta(Z)] - 2\delta \cdot \omega^\intercal \mathbb{E}[Y\psi_\theta(Z)]$$
$$= \mathcal{L}_0(\theta) + \omega^\intercal C_{\psi_\theta}\omega - 2\delta \cdot \omega^\intercal \mathbb{E}[Y\psi_\theta(Z)]$$
$$= \mathcal{L}_\delta(\theta, \omega).$$

We conclude the proof with Theorem A.3:

$$\|\mathcal{T}_\delta - \mathcal{T}_{\theta,\omega}\|_{\mathrm{HS}}^2 - \|\mathcal{T}_\delta\|_{\mathrm{HS}}^2 \geq \left\|\mathcal{T}_\delta - \mathcal{T}_\delta^{(d)}\right\|_{\mathrm{HS}}^2 - \|\mathcal{T}_\delta\|_{\mathrm{HS}}^2 = -\left\|\mathcal{T}_\delta^{(d)}\right\|_{\mathrm{HS}}^2.$$

## C.2. Proof of Proposition 6.1

Recall the SVD of $\mathcal{T}_\delta$ (Eq. (6))

$$\mathcal{T}_\delta = \sum_{i \geq 1} \sigma_{\star,i} \psi_{\star,i} \otimes (\varphi_{\star,i}, \omega_{\star,i}).$$

The key relationships it satisfies are $\sigma_{\star,i} \varphi_{\star,i} = \mathcal{T}^* \psi_{\star,i}$, and $\sigma_{\star,i} \omega_{\star,i} = \delta \langle r_0, \psi_{\star,i} \rangle = \delta \cdot \sigma_{\star,i} \langle h_0, \varphi_{\star,i} \rangle$ (see Eq. (11)). In Eq. (14), we show that $\Phi_\star^{(d)\star} \Phi_\star^{(d)} = I_d - \omega_\star \omega_\star^\top$. Thus

$$\Pi_{\varphi_\star^{(d)}} = \Phi_\star^{(d)} (\Phi_\star^{(d)\star} \Phi_\star^{(d)})^{-1} \Phi_\star^{(d)\star} = \Phi_\star^{(d)} (I_d - \omega_\star \omega_\star^\top)^{-1} \Phi_\star^{(d)\star}.$$

Next, observe that

$$\langle \varphi_{\star,i}, h_0 \rangle = \sigma_{\star,i}^{-1} \langle \mathcal{T}^* \psi_{\star,i}, h_0 \rangle = \sigma_{\star,i}^{-1} \langle \psi_{\star,i}, \mathcal{T} h_0 \rangle = \sigma_{\star,i}^{-1} \mathbb{E}[Y \psi_{\star,i}(Z)] = \alpha_i.$$

Therefore, $\alpha$ is such that $\alpha = \Phi_\star^{(d)\star} h_0$. Alignment of $h_0$ to the true spectral features (of $\mathcal{T}_\delta$) then satisfies:

$$\|\Pi_{\varphi_\star^{(d)}} h_0\|^2 = \langle h_0, \Pi_{\varphi_\star^{(d)}} h_0 \rangle = (\Phi_\star^{(d)\star} h_0)^\top \left( I_d - \omega_\star \omega_\star^\top \right)^{-1} \Phi_\star^{(d)\star} h_0 = \alpha^\top \left( I_d - \omega_\star \omega_\star^\top \right)^{-1} \alpha.$$

## C.3. Approximation Error Analysis

Recall that for $d \geq 1$, we fixed a partition $\overline{N} \dot\cup \underline{N} = \mathbb{N}$, such that $|\overline{N}| = d$, $\mathcal{T} = \overline{U}_d \overline{\Lambda}_d \overline{V}_d^* + \underline{U}_d \underline{\Lambda}_d \underline{V}_d^*$. We introduce the projection operators

$$\Pi_{\overline{U}_d} = \overline{U}_d \overline{U}_d^* \qquad \Pi_{\overline{V}_d} = \overline{V}_d \overline{V}_d^* \qquad \Pi_{\underline{U}_d} = \underline{U}_d \underline{U}_d^* = I - \Pi_{\overline{U}_d} \qquad \Pi_{\underline{V}_d} = \underline{V}_d \underline{V}_d^* = I - \Pi_{\overline{V}_d}.$$

We then decompose $h_0$ as

$$h_0 = \overline{h}_0 + \underline{h}_0 = \Pi_{\overline{V}_d} h_0 + \Pi_{\underline{V}_d} h_0 = \sum_{i \in \overline{N}} \overline{\alpha}_i v_i + \sum_{i \in \underline{N}} \underline{\alpha}_i v_i.$$

Let us define $\overline{\lambda}_{\min}$ as the smallest positive entries of $\overline{\Lambda}_d$.

**Proposition C.1.** $\overline{\lambda}_{\min}$ *is such that*

$$\sup_{h \neq 0} \frac{\|\Pi_{\overline{V}_d} h\|_{L_2(X)}}{\|\mathcal{T} \Pi_{\overline{V}_d} h\|_{L_2(Z)}} = \overline{\lambda}_{\min}^{-1}.$$

*Proof.* Since the quantity is homogeneous in $h$, we may restrict the supremum to

$$\|\Pi_{\overline{V}_d} h\|_{L_2(X)} = 1.$$

Write $\Pi_{\overline{V}_d} h = \sum_{i \in \overline{N}} \alpha_i v_i$ so that $\sum_{i \in \overline{N}} |\alpha_i|^2 = 1$. Using $\mathcal{T} v_i = \lambda_i u_i$ for $i \in \overline{N}$ we get

$$\|\mathcal{T} \Pi_{\overline{V}_d} h\|_{L_2(Z)}^2 = \sum_{i \in \overline{N}} \lambda_i^2 |\alpha_i|^2.$$

Hence, for such $h$,

$$\frac{\|\Pi_{\overline{V}_d} h\|_{L_2(X)}}{\|\mathcal{T} \Pi_{\overline{V}_d} h\|_{L_2(Z)}} = \frac{1}{\left( \sum_{i \in \overline{N}} \lambda_i^2 |\alpha_i|^2 \right)^{1/2}} \leq \frac{1}{\overline{\lambda}_{\min}}.$$

Equality is attained by taking $h$ proportional to the singular vector $v_{i^*}$ with $\lambda_{i^*} = \overline{\lambda}_{\min}$, which concludes the proof. $\square$

The following proposition states a general approximation error bound when the span of the learned features is close to the singular spaces of $\mathcal{T}$ associated to $\overline{N}$.

**Proposition C.2.** *Let $A, B$ be constants such that $\left\| \Pi_{\overline{U}_d} - \Pi_{\psi_\theta} \right\|_{\mathrm{op}} \leq A$, $\left\| \Pi_{\overline{V}_d} - \Pi_{\varphi_\theta} \right\|_{\mathrm{op}} \leq B$, and*

$$1 - B - \frac{A}{\overline{\lambda}_{\min}} > 0.$$

*Then,*

$$\|h_\theta - h_0\|_{L_2(X)} \leq \left( 1 - B - \frac{A}{\overline{\lambda}_{\min}} \right)^{-1} \left( B \|h_0\|_{L_2(X)} + \|\Pi_{\underline{V}_d} h_0\|_{L_2(X)} \right).$$

*Proof.* Recall that under Assumption 3, $h_\theta(x) = \varphi_\theta(x)^\intercal \beta_\theta$ satisfies $C_{ZX,\theta}\beta_\theta = \mathbb{E}[r_0(Z)\psi_\theta(Z)]$. Observing that $\mathbb{E}[r_0(Z)\psi_\theta(Z)] = \Psi_\theta^* r_0 = \Psi_\theta^* \mathcal{T} h_0$ and using that $C_{ZX,\theta} = \Psi_\theta^* \mathcal{T} \Phi_\theta$, we have $C_{ZX,\theta}\beta_\theta = \Psi_\theta^* \mathcal{T} \, h_\theta$. Therefore $0 = \Psi_\theta^* \mathcal{T}(h_\theta - h_0)$, which implies

$$\Pi_{\psi_\theta} \mathcal{T}(h_\theta - h_0) = 0 \tag{10}$$

We then have the following chain of inequalities,

$$
\begin{aligned}
\|h_\theta - h_0\|_{L_2(X)} &\leq \|\Pi_{\overline{V}_d}(h_\theta - h_0)\|_{L_2(X)} + \|\Pi_{\underline{V}_d}(h_\theta - h_0)\|_{L_2(X)} \\
&\leq \overline{\lambda}_{\min}^{-1}\|\mathcal{T}\,\Pi_{\overline{V}_d}(h_\theta - h_0)\|_{L_2(Z)} + \|\Pi_{\underline{V}_d}h_\theta\|_{L_2(X)} + \|\Pi_{\underline{V}_d}h_0\|_{L_2(X)} \\
&= \overline{\lambda}_{\min}^{-1}\|\Pi_{\overline{U}_d}\mathcal{T}(h_\theta - h_0)\|_{L_2(Z)} + \|\left(\Pi_{\underline{V}_d} - \Pi_{\varphi_\theta}^\perp\right)h_\theta\|_{L_2(X)} + \|\Pi_{\underline{V}_d}h_0\|_{L_2(X)} \\
&\leq \overline{\lambda}_{\min}^{-1}\left\|\Pi_{\overline{U}_d} - \Pi_{\psi_\theta}\right\|_{\mathrm{op}}\|\mathcal{T}\|_{\mathrm{op}}\|h_\theta - h_0\|_{L_2(X)} + \left\|\Pi_{\overline{V}_d} - \Pi_{\varphi_\theta}\right\|_{\mathrm{op}}\|h_\theta\|_{L_2(X)} + \|\Pi_{\underline{V}_d}h_0\|_{L_2(X)} \\
&\leq \frac{A}{\overline{\lambda}_{\min}}\|h_\theta - h_0\|_{L_2(X)} + B\|h_\theta - h_0 + h_0\|_{L_2(X)} + \|\Pi_{\underline{V}_d}h_0\|_{L_2(X)} \\
&\leq \left(\frac{A}{\overline{\lambda}_{\min}} + B\right)\|h_\theta - h_0\|_{L_2(X)} + B\|h_0\|_{L_2(X)} + \|\Pi_{\underline{V}_d}h_0\|_{L_2(X)},
\end{aligned}
$$

where in the second inequality we used Proposition C.1 and the triangular inequality, in the equality we used $\mathcal{T}\,\Pi_{\overline{V}_d} = \Pi_{\overline{U}_d}\mathcal{T}$ and $\Pi_{\varphi_\theta}^\perp h_\theta = 0$, in the third inequality we used Eq. (10) and in the fourth inequality we used $\|\mathcal{T}\|_{\mathrm{op}} \leq 1$. Re-arranging, we obtain

$$\left(1 - \frac{A}{\overline{\lambda}_{\min}} - B\right)\|h_\theta - h_0\|_{L_2(X)} \leq B\|h_0\|_{L_2(X)} + \|\Pi_{\underline{V}_d}h_0\|_{L_2(X)},$$

and the result follows. $\qquad\square$

The next step is to obtain $A$ and $B$. Recall the SVD of $\mathcal{T}_\delta$ in Eq. (6),

$$\mathcal{T}_\delta = \sum_{i \geq 1} \sigma_{*,i}\psi_{*,i} \otimes (\varphi_{*,i}, \omega_{*,i}), \quad \psi_{*,i} \in L_2(Z), \quad (\varphi_{*,i}, \omega_{*,i}) \in L_2(X) \times \mathbb{R},$$

We denote by $\Pi_{\psi_*^{(d)}}$ and $\Pi_{\varphi_*^{(d)}}$ the orthogonal projections onto the span of $(\psi_{*,i})_{i=1}^d$ and $(\varphi_{*,i})_{i=1}^d$ respectively. It is important to note that $\varphi_{*,i}$ is not a singular function of $\mathcal{T}_\delta$ but the first component of the singular function $(\varphi_{*,i}, \omega_{*,i})$. Therefore the family $\{\varphi_{*,i}\}_i$ is not guaranteed to be orthonormal. From the SVD, we deduce the following relationships for all $i \geq 1$

$$
\begin{cases}
\sigma_{\star,i}\psi_{\star,i} = \mathcal{T}_\delta(\varphi_{\star,i}, \omega_{\star,i}), \\
\sigma_{\star,i}\varphi_{\star,i} = \mathcal{T}^* \psi_{\star,i}, \\
\sigma_{\star,i}\omega_{\star,i} = \delta\,\langle\psi_{\star,i}, r_0\rangle_{L_2(Z)},
\end{cases}
\tag{11}
$$

where we use the fact that

$$\mathcal{T}_\delta^\star r = \left(\mathcal{T}^\star r,\ \delta\,\langle r, r_0\rangle_{L_2(Z)}\right) \in L_2(X) \times \mathbb{R}, \qquad r \in L_2(Z).$$

By the triangular inequality,

$$
\left\|\Pi_{\overline{U}_d} - \Pi_{\psi_\theta}\right\|_{\mathrm{op}} \leq \left\|\Pi_{\overline{U}_d} - \Pi_{\psi_*^{(d)}}\right\|_{\mathrm{op}} + \left\|\Pi_{\psi_*^{(d)}} - \Pi_{\psi_\theta}\right\|_{\mathrm{op}} \qquad (Z-\text{features})
$$

$$
\left\|\Pi_{\overline{V}_d} - \Pi_{\varphi_\theta}\right\|_{\mathrm{op}} \leq \left\|\Pi_{\overline{V}_d} - \Pi_{\varphi_*^{(d)}}\right\|_{\mathrm{op}} + \left\|\Pi_{\varphi_*^{(d)}} - \Pi_{\varphi_\theta}\right\|_{\mathrm{op}} \qquad (X-\text{features})
$$

**Proposition C.3** (Control of the $X$-features)**.**

$$\left\|\Pi_{\varphi_*^{(d)}} - \Pi_{\overline{V}_d}\right\|_{\mathrm{op}} \leq \frac{\left\|\Pi_{\psi_*^{(d)}} - \Pi_{\overline{U}_d}\right\|_{\mathrm{op}}}{\overline{\lambda}_{\min}(1 - \left\|\Pi_{\psi_*^{(d)}} - \Pi_{\overline{U}_d}\right\|_{\mathrm{op}})}. \tag{12}$$

*Assume in addition that* $\overline{\lambda}_{\min} - \left\| \Pi_{\psi_*^{(d)}} - \Pi_{\overline{U}_d} \right\|_{\mathrm{op}} > 0$. *Then,*

$$\left\| \Pi_{\varphi_*^{(d)}} - \Pi_{\varphi_\theta} \right\|_{\mathrm{op}} \leq \frac{\left\| \mathcal{T}_\delta^{(d)} - \mathcal{T}_{\theta,\omega} \right\|_{\mathrm{op}}}{\overline{\lambda}_{\min} - \left\| \Pi_{\psi_*^{(d)}} - \Pi_{\overline{U}_d} \right\|_{\mathrm{op}}}. \tag{13}$$

*Proof.* We start with Eq. (13). Let $\pi : L_2(X) \times \mathbb{R} \to L_2(X)$ be the canonical projection such that $\pi(h, a) = h$. Recall that

$$\Phi_\star^{(d)} \beta = \sum_{i=1}^d \beta_i \varphi_{*,i}.$$

Let us define

$$\tilde{\Phi}_\star^{(d)} \beta = \sum_{i=1}^d \beta_i (\varphi_{*,i}, \omega_{*,i}),$$

that is such that $\pi \tilde{\Phi}_\star^{(d)} = \Phi_\star^{(d)}$. We then have the following decomposition

$$\Phi_\star^{(d)} \Sigma_\star^{(d)} \Psi_\star^{(d)\star} = \pi(\tilde{\Phi}_\star^{(d)} \Sigma_\star^{(d)} \Psi_\star^{(d)\star}) = \pi(\mathcal{T}_\delta^{(d)})^* = \pi(\mathcal{T}_\delta^{(d)} - \mathcal{T}_{\theta,\omega})^* + \pi \mathcal{T}_{\theta,\omega}^*$$
$$= \pi(\mathcal{T}_\delta^{(d)} - \mathcal{T}_{\theta,\omega})^* + \Phi_\theta \Psi_\theta^*$$

We will apply Wedin Sin-$\Theta$ Theorem, Theorem A.2, to $A = \Phi_\star^{(d)} \Sigma_\star^{(d)} \Psi_\star^{(d)\star}$ and $B = \Phi_\theta \Psi_\theta^*$. Note that

$$\|A - B\|_{\mathrm{op}} = \|\Phi_\star^{(d)} \Sigma_\star^{(d)} \Psi_\star^{(d)\star} - \Phi_\theta \Psi_\theta^*\|_{\mathrm{op}}$$
$$= \left\| \pi(\mathcal{T}_\delta^{(d)})^* (\Pi_{\psi_*^{(d)}} - \Pi_{\psi_\theta}) + \pi(\mathcal{T}_\delta^{(d)} - \mathcal{T}_{\theta,\omega})^* \Pi_{\psi_\theta} \right\|_{\mathrm{op}}$$
$$\leq \sigma_1(\mathcal{T}_\delta^{(d)}) \left\| \Pi_{\psi_*^{(d)}} - \Pi_{\psi_\theta} \right\|_{\mathrm{op}} + \left\| \mathcal{T}_\delta^{(d)} - \mathcal{T}_{\theta,\omega} \right\|_{\mathrm{op}}.$$

As $(\varphi_{*,i}, \omega_{*,i})_{i=1}^d$ forms an orthonormal family, we have $\tilde{\Phi}_\star^{(d)\star} \tilde{\Phi}_\star^{(d)} = I_d$. On the other hand, similarly to Eq. (9), we have

$$I_d = \tilde{\Phi}_\star^{(d)\star} \tilde{\Phi}_\star^{(d)} = \Phi_\star^{(d)\star} \Phi_\star^{(d)} + \omega_* \omega_*^\mathsf{T} \tag{14}$$

Therefore,
$$\sigma_d(A)^2 = \sigma_d(\Phi_\star^{(d)} \Sigma_\star^{(d)} \Psi_\star^{(d)\star})^2 = \sigma_d(\Psi_\star^{(d)} \Sigma_\star^{(d)} \Phi_\star^{(d)\star} \Phi_\star^{(d)} \Sigma_\star^{(d)} \Psi_\star^{(d)\star})$$
$$= \sigma_d(\Psi_\star^{(d)} \Sigma_\star^{(d)} (I_d - \omega_\star \omega_\star^\mathsf{T}) \Sigma_\star^{(d)} \Psi_\star^{(d)\star})$$
$$= \sigma_d(\mathcal{T}_\delta^{(d)} (\mathcal{T}_\delta^{(d)})^* - \delta^2 (\Psi_\star^{(d)} \Psi_\star^{(d)\star} r_0) \otimes (\Psi_\star^{(d)} \Psi_\star^{(d)\star} r_0))$$
$$= \sigma_d(\Pi_{\psi_*^{(d)}} (\mathcal{T}_\delta (\mathcal{T}_\delta)^* - \delta^2 r_0 \otimes r_0) \Pi_{\psi_*^{(d)}})$$
$$= \sigma_d(\Pi_{\psi_*^{(d)}} \mathcal{T} \mathcal{T}^* \Pi_{\psi_*^{(d)}})$$
$$= \sigma_d(\Pi_{\psi_*^{(d)}} \mathcal{T})^2,$$

where in the third equality, we used that fact that $\omega_\star = \delta(\Sigma_\star^{(d)})^{-1} \Psi_\star^{(d)\star} r_0$ by Eq. (11) and in the fifth equality we used $\mathcal{T}_\delta(\mathcal{T}_\delta)^* = \mathcal{T}\mathcal{T}^* + \delta^2 r_0 \otimes r_0$. Next, by Weyl's inequality, Proposition A.1,

$$|\sigma_d(\Pi_{\psi_*^{(d)}} \mathcal{T}) - \overline{\lambda}_{\min}| = |\sigma_d(\Pi_{\psi_*^{(d)}} \mathcal{T}) - \sigma_d(\Pi_{\overline{U}_d} \mathcal{T})| \leq \left\| (\Pi_{\psi_*^{(d)}} - \Pi_{\overline{U}_d}) \mathcal{T} \right\|_{\mathrm{op}} \leq \left\| \Pi_{\psi_*^{(d)}} - \Pi_{\overline{U}_d} \right\|_{\mathrm{op}}.$$

Hence $\sigma_d(A) \geq \overline{\lambda}_{\min} - \left\| \Pi_{\psi_*^{(d)}} - \Pi_{\overline{U}_d} \right\|_{\mathrm{op}}$ and $\sigma_{d+1}(B) = 0$. We obtain Eq. (13) applying Theorem A.2.

We now prove Eq. (12). We use

$$\Phi_\star^{(d)} \Sigma_\star^{(d)} = \mathcal{T}^* \Psi_\star^{(d)} = \overline{V}_d \overline{\Lambda}_d \overline{U}_d^* \Psi_\star^{(d)} + \underline{V}_d \underline{\Lambda}_d \underline{U}_d^* \Psi_\star^{(d)},$$

We will apply Wedin Sin-$\Theta$ Theorem, Theorem A.2, to $A' = \overline{V}_d \overline{\Lambda}_d \overline{U}_d^* \Psi_\star^{(d)}$ and $B' = \Phi_\star^{(d)} \Sigma_\star^{(d)}$. Note that

$$\|A' - B'\|_{\mathrm{op}} = \left\| \overline{V}_d \overline{\Lambda}_d \overline{U}_d^* \Psi_\star^{(d)} - \Phi_\star^{(d)} \Sigma_\star^{(d)} \right\|_{\mathrm{op}} = \|\underline{V}_d \underline{\Lambda}_d \underline{U}_d^* \Psi_\star^{(d)}\|_{\mathrm{op}} \leq \left\| \Pi_{\overline{U}_d} - \Pi_{\psi_\star^{(d)}} \right\|_{\mathrm{op}},$$

where for the last inequality, we use

$$\underline{U}_d^* \Psi_\star^{(d)} = \underline{U}_d^* \Pi_{\underline{U}_d} \Psi_\star^{(d)} = \underline{U}_d^* \Pi_{\overline{U}_d}^\perp \Psi_\star^{(d)} = \underline{U}_d^* [\Pi_{\overline{U}_d}^\perp - \Pi_{\psi_\star^{(d)}}^\perp] \Psi_\star^{(d)} = \underline{U}_d^* [\Pi_{\overline{U}_d} - \Pi_{\psi_\star^{(d)}}] \Psi_\star^{(d)}.$$

By the inequality for singular values of product of matrices, we have

$$\begin{aligned}
\sigma_d(A') = \sigma_d(\overline{V}_d \overline{\Lambda}_d \overline{U}_d^* \Psi_\star^{(d)}) &\geq \sigma_d(\overline{V}_d \overline{\Lambda}_d) \, \sigma_d(\overline{U}_d^* \Psi_\star^{(d)}) \\
&= \sigma_d(\overline{\Lambda}_d) \, \sigma_d(\overline{U}_d^* \Psi_\star^{(d)}) \\
&= \overline{\lambda}_{\min} \, \sigma_d(\overline{U}_d^* \Psi_\star^{(d)}) \\
&\geq \overline{\lambda}_{\min} (1 - \left\| \Pi_{\psi_\star^{(d)}} - \Pi_{\overline{U}_d} \right\|_{\mathrm{op}}).
\end{aligned}$$

We conclude with Theorem A.2, using that $\sigma_{d+1}(B') = 0$. $\qquad\square$

**Proposition C.4** (Control of the $Z$-features). *Assume that $\mathcal{E}_d(\theta, \omega, \delta) \leq \sigma_d(\mathcal{T}_\delta)/2$. Then,*

$$\left\| \Pi_{\psi_\star^{(d)}} - \Pi_{\psi_\theta} \right\|_{\mathrm{op}} \leq 2 \cdot \frac{\mathcal{E}_d(\theta, \omega, \delta)}{\sigma_d(\mathcal{T}_\delta)}$$

*Assume that $\gamma_d(\delta) \doteq \left\| [\overline{\Lambda}_d (I + \delta^2 \overline{\alpha}\overline{\alpha}^\mathsf{T})^{1/2}]^{-1} \right\|_{\mathrm{op}}^{-1} - \|\underline{\Lambda}_d\|_{\mathrm{op}} > 0$ and $\|\underline{\Lambda}_d\|_{\mathrm{op}} \|\underline{\alpha}_0\|_{\ell_2} \leq \gamma_d(\delta)/2$. Then,*

$$\left\| \Pi_{\overline{U}_d} - \Pi_{\psi_\star^{(d)}} \right\|_{\mathrm{op}} \leq 2 \cdot \frac{\delta \|\underline{\Lambda}_d\|_{\mathrm{op}} \|\underline{\alpha}_0\|_{\ell_2}}{\gamma_d(\delta)}.$$

*Proof.* The first inequality follows from Theorem A.2 and the definition of $\mathcal{E}_d(\theta, \omega, \delta)$.

For the second inequality, using $\mathcal{T} = \overline{U}_d \overline{\Lambda}_d \overline{V}_d^* + \underline{U}_d \underline{\Lambda}_d \underline{V}_d^*$, we get the following decomposition

$$\mathcal{T}_\delta = [\mathcal{T} \mid \delta r_0] = [\mathcal{T} \mid \delta \mathcal{T} h_0] = \underbrace{[\mathcal{T} \mid \delta \overline{U}_d \overline{\Lambda}_d \overline{V}_d^* h_0]}_{\tilde{Q}} + [0 \mid \delta \underline{U}_d \underline{\Lambda}_d \underline{V}_d^* h_0]$$

Note that

$$\left\| \mathcal{T}_\delta - \tilde{Q} \right\|_{\mathrm{op}} = \|\delta \underline{U}_d \underline{\Lambda}_d \underline{\alpha}_0\|_{L_2(Z)} \leq \delta \|\underline{\Lambda}_d\|_{\mathrm{op}} \|\underline{\alpha}_0\|_{L_2(X)}$$

To analyze the spectrum of $\tilde{Q}$, we look at

$$\tilde{Q}\tilde{Q}^* = \mathcal{T}\mathcal{T}^* + \delta^2 (\overline{U}_d \overline{\Lambda}_d \overline{V}_d^* h_0) \otimes (\overline{U}_d \overline{\Lambda}_d \overline{V}_d^* h_0) = [\overline{U}_d \mid \underline{U}_d] \begin{bmatrix} \overline{M} & 0 \\ 0 & \underline{\Lambda}_d^2 \end{bmatrix} [\overline{U}_d \mid \underline{U}_d]^*,$$

where $\overline{M} = \overline{\Lambda}_d^2 + \delta^2 \overline{\Lambda}_d \overline{V}_d^* (h_0 \otimes h_0) \overline{V}_d \overline{\Lambda}_d$. When $\lambda_{\min}(\overline{M}) > \|\underline{\Lambda}_d^2\|_{\mathrm{op}}$ then $\Pi_{\overline{U}_d}$ is associated to the top spectrum of $\tilde{Q}$ and we can apply Wedin Sin-$\Theta$ Theorem, Theorem A.2, to $A = \tilde{Q}$ and $B = \mathcal{T}_\delta$. As

$$\overline{M} = \overline{\Lambda}_d (I + \delta^2 \overline{\alpha}\overline{\alpha}^\mathsf{T}) \overline{\Lambda}_d,$$

we have

$$\lambda_{\min}(\overline{M}) = \sigma_{\min}^2(\overline{\Lambda}_d (I + \delta^2 \overline{\alpha}\overline{\alpha}^\mathsf{T})^{1/2}) = \left\| [\overline{\Lambda}_d (I + \delta^2 \overline{\alpha}\overline{\alpha}^\mathsf{T})^{1/2}]^{-1} \right\|_{\mathrm{op}}^{-2}.$$

Therefore $\lambda_{\min}(\overline{M}) > \|\underline{\Lambda}_d^2\|_{\mathrm{op}}$ is equivalent to

$$\sigma_d(A) - \sigma_{d+1}(B) = \left\| [\overline{\Lambda}_d (I + \delta^2 \overline{\alpha}\overline{\alpha}^\mathsf{T})^{1/2}]^{-1} \right\|_{\mathrm{op}}^{-1} - \|\underline{\Lambda}_d\|_{\mathrm{op}} = \gamma_d(\delta) > 0.$$

$\qquad\square$

**Proposition C.5** (Control of ill-posedness). *Under the assumptions of Proposition C.2, it holds that*

$$c_{\varphi_\theta, \psi_\theta} \geq \overline{\lambda}_{\min} \left( 1 - B - \frac{A}{\overline{\lambda}_{\min}} \right).$$

*Proof.* Start by decomposing projections as

$$\Pi_{\Psi_\theta} \mathcal{T} \Pi_{\varphi_\theta} = \Pi_{\overline{U}_d} \mathcal{T} (\Pi_{\varphi_\theta} + \Pi_{\underline{V}_d}) + (\Pi_{\Psi_\theta} - \Pi_{\overline{U}_d}) \mathcal{T} \Pi_{\varphi_\theta}.$$

Now, using inequalities of sums and products of singular values we have that

$$\sigma_d(\Pi_{\Psi_\theta} \mathcal{T} \Pi_{\varphi_\theta}) \geq \sigma_d(\Pi_{\overline{U}_d} \mathcal{T}) \sigma_{\min}(\Pi_{\varphi_\theta} + \Pi_{\underline{V}_d}) - \sigma_1((\Pi_{\Psi_\theta} - \Pi_{\overline{U}_d}) \mathcal{T} \Pi_{\varphi_\theta}).$$

But, since

$$\sigma_{\min}(\Pi_{\varphi_\theta} + \Pi_{\underline{V}_d}) = \left\| [I - (\Pi_{\overline{V}_d} - \Pi_{\varphi_\theta})]^{-1} \right\|_{\mathrm{op}}^{-1} \geq 1 - \left\| \Pi_{\overline{V}_d} - \Pi_{\varphi_\theta} \right\|_{\mathrm{op}},$$

whenever $\left\| \Pi_{\overline{V}_d} - \Pi_{\varphi_\theta} \right\|_{\mathrm{op}} < 1$, the proof is completed. $\square$

We combine the previous proposition in the following general approximation error bound.

**Theorem C.6.** *Given any $\delta \geq 0$ and partition $\overline{N} \dot\cup \underline{N} = \mathbb{N}$, denoting $\overline{\lambda}_{\min} = \min_{i \in [\overline{N}]} \lambda_i$ and $\underline{\lambda}_{\max} = \max_{i \in [\underline{N}]} \lambda_i$, if*

$$\frac{6\delta \underline{\lambda}_{\max}}{\overline{\lambda}_{\min} \gamma_d(\delta)} \|q_d\|_{L_2(X)} + \frac{2\lambda_d + \overline{\lambda}_{\min}}{\lambda_d \overline{\lambda}_{\min}} \mathcal{E}_d(\theta, \delta) \leq \kappa \leq 1/2 \tag{15}$$

*then $c_{\varphi_\theta, \psi_\theta} \geq (1 - \kappa)\overline{\lambda}_{\min}$, and*

$$\|h_0 - h_\theta\|_{L_2(X)} \leq \left( 1 + \frac{4\delta \underline{\lambda}_{\max} \|h_0\|_{L_2(X)}}{\overline{\lambda}_{\min} \gamma_d(\delta)} \right) \frac{\|q_d\|_{L_2(X)}}{1 - \kappa} + \frac{(2\lambda_d + \overline{\lambda}_{\min}) \|h_0\|_{L_2(X)}}{\overline{\lambda}_{\min} \lambda_d} \frac{\mathcal{E}_d(\theta, \delta)}{1 - \kappa}. \tag{16}$$

*Proof.* First, observe that $\sigma_d(\mathcal{T}_\delta) \geq \lambda_d$ and $\sigma_1(\mathcal{T}_\delta) \leq (1 + \delta\|r_0\|_{L_2(Z)})$. Now, recalling Proposition C.2, due to Proposition C.4 we can take

$$A = \frac{2\delta \underline{\lambda}_{\max} \|q_d\|_{L_2(X)}}{\gamma_d(\delta)} + \frac{\mathcal{E}_d(\theta, \delta)}{\lambda_d}$$

But, since Eq. (15) ensures that $A/\overline{\lambda}_{\min} < 1/2$, from Proposition C.3 we can set

$$B = \frac{4\delta \underline{\lambda}_{\max} \|q_d\|_{L_2(X)}}{\overline{\lambda}_{\min} \gamma_d(\delta)} + \mathcal{E}_d(\theta, \delta) \left( \frac{1}{\lambda_d} + \frac{2}{\overline{\lambda}_{\min}} \right)$$

and obtain that $A/\overline{\lambda}_{\min} + B \leq \kappa < 1/2$. To complete the proof we apply Proposition C.5. $\square$

**Good scenario.** Setting $\overline{N} = \{1, \ldots, d\}$ and $\delta = 0$ we obtain the control of the approximation error for the SpecIV learning method. That is, equation 16 becomes

$$\|h_0 - h_\theta\|_{L_2(X)} \leq \frac{1}{1 - \kappa} \left( \|q_d\|_{L_2(X)} + \frac{3\|h_0\|_{L_2(X)}}{\lambda_d} \mathcal{E}_d(\theta, \delta) \right),$$

whenever $\mathcal{E}_d(\theta, \delta) \leq \kappa\lambda_d/3$. Here we see that the representation learning error needs to scale as $\mathcal{E}_d(\theta, \delta) \asymp \lambda_d \|q_d\|_{L_2(X)} / \|h_0\|_{L_2(X)}$, as $d \to \infty$.

**Bad scenario.** Let $\overline{N} = \{k\}$ and assume that $\|s_1\|_{L_2(X)}=|\langle h_0, v_k\rangle| > \|h_0-\alpha_k v_k\|_{L_2(X)}=\|q_1\|_{L_2(X)}$. From Eq. (8), we have that for large enough $\delta > 0$ the gap $\gamma_1(\delta)=\lambda_k\sqrt{1+\delta^2\|s_1\|_{L_2(X)}^2}-1$ is positive. So, taking $\delta = 7(1 - \lambda_k)/(\lambda_k\|s_1\|_{L_2(X)})$, we have that $\gamma_1(\delta) > \lambda_k\delta\|s_1\|_{L_2(X)}-(1-\lambda_k)=6(1-\lambda_k)$, and, hence applying Theorem C.6 of Appendix C.3), we conclude that

$$\|h_0-h_\theta\|_{L_2(X)}\leq\frac{1}{1-\kappa}\left(\frac{5\|h_0\|_{L_2(X)} + \lambda_k^2\|s_1\|_{L_2(X)}}{\lambda_k^2\|s_1\|_{L_2(X)}}\|q_1\|_{L_2(X)}+\frac{2\|h_0\|_{L_2(X)}}{\lambda_k}\,\mathcal{E}_1(\theta, \delta)\right),$$

whenever $7\|q_1\|_{L_2(X)}/\|s_1\|_{L_2(X)}+3\,\mathcal{E}_1(\theta, \omega, \delta)\,\lambda_k \leq \kappa\lambda_k^2$. Therefore, whenever $\|s_1\|_{L_2(X)} \gg \|q_1\|_{L_2(X)}$, we are able to learn the most dominant part of the structural function with just one feature.

## D. Statistical Analysis

We recall our estimation procedure.

**Estimator:** Given i.i.d. data $(Y_i, X_i, Z_i)_{i=1}^n$, we estimate $h_0$ via the two-stage procedure:

$$\hat{C}_{ZX,\theta} = \frac{1}{n}\sum_{i=1}^n \psi_\theta(Z_i)\varphi_\theta(X_i)^\intercal \in \mathbb{R}^{d\times d}, \quad \hat{g} = \frac{1}{n}\sum_{i=1}^n Y_i\psi_\theta(Z_i) \in \mathbb{R}^d.$$

$$\hat{\beta}_\theta = \hat{C}_{ZX,\theta}^{-1}\hat{g}, \quad \hat{h}_\theta(x) = \varphi_\theta(x)^\intercal\hat{\beta}_\theta.$$

### D.1. Proof of Theorem 5.1

We decompose the excess risk as

$$\|\hat{h}_\theta - h_0\|_{L_2(X)} \leq \|\hat{h}_\theta - h_\theta\|_{L_2(X)} + \|h_\theta - h_0\|_{L_2(X)}.$$

Proposition D.2 provides the control on the estimation error, that is, w.p.a.l. $1 - \tau$

$$\|\hat{h}_\theta - h_\theta\|_{L_2(X)} \leq \frac{c}{c_{\varphi_\theta^{(d)},\psi_\theta^{(d)}}}\sqrt{\frac{d}{n}}\sqrt{\sigma_{\text{eff}}^2 + \frac{\rho^2}{n}}\,\log\frac{4}{\tau},$$

where $\sigma_{\text{eff}}^2 \doteq \|h_0 - h_\theta\|_{L_2(X)}^2 + \sigma_U^2$ and $c > 0$ is an absolute constant.

Let $a \doteq \|h_0 - h_\theta\|_{L_2(X)}$ and

$$m_n \doteq \frac{1}{c_{\varphi_\theta^{(d)},\psi_\theta^{(d)}}}\sqrt{\frac{d}{n}}\log\frac{4}{\tau}.$$

Since

$$\sqrt{\sigma_{\text{eff}}^2 + \rho^2/n} = \sqrt{a^2 + \sigma_U^2 + \rho^2/n} \leq a + \sqrt{\sigma_U^2 + \rho^2/n},$$

we obtain

$$\|\hat{h}_\theta - h_0\|_{L_2(X)} \leq (1 + cm_n)a + cm_n\sqrt{\sigma_U^2 + \rho^2/n}.$$

Moreover, the sample-size condition of Theorem 5.1 implies $m_n \leq 1/4$, using $\rho \geq 1$ and $\log(4/\tau) \leq \log(4d/\tau)$. Absorbing constants gives the stated bound.

### D.2. Proof of auxiliary results

We also recall the definition of sub-Gaussian random variables.

**Definition D.1** (Sub-Gaussian random vector)**.** We define the proxy variance $\overline{\sigma}_{\mathbf{x}}$ of a real-valued random variable $\mathbf{x}$, which controls the tail behavior of $\mathbf{x}$ via $\mathbb{P}(|\mathbf{x} - \mathbb{E}[\mathbf{x}]| > t) \leq 2\exp\left(-t^2/\overline{\sigma}_{\mathbf{x}}^2\right)$. A random vector $X \in \mathbb{R}^d$ will be called sub-Gaussian iff, there exists an absolute constant such that, for all $u \in \mathbb{R}^d$, $\overline{\sigma}_{\langle X,u\rangle} \leq c\|\langle X, u\rangle - \mathbb{E}\langle X, u\rangle\|_{L_2(\mathbb{P})}$.

**Proposition D.2** (Concentration for Sub-Gaussian Random Variables). *Let Assumptions 2-4 be satisfied. Given $\tau \in (0, 1)$, let $n \geq 16 \, d \, \rho^2 \, \log(4d/\tau) c_{\varphi_\theta^{(d)}, \psi_\theta^{(d)}}^{-2}$. Then there exists an absolute constant $c > 0$ such that, with probability at least $1 - \tau$,*

$$\|\hat{h}_\theta - h_\theta\|_{L_2(X)} \leq \frac{c}{c_{\varphi_\theta^{(d)}, \psi_\theta^{(d)}}} \sqrt{\frac{d}{n}} \sqrt{\sigma_{\text{eff}}^2 + \frac{\rho^2}{n}} \, \log \frac{4}{\tau},$$

*where $\sigma_{\text{eff}}^2 \doteq \|h_0 - h_\theta\|_{L_2(X)}^2 + \sigma_U^2$.*

*Proof of Proposition D.2.* Let us denote

$$\hat{g} \doteq \hat{\mathbb{E}}_n[Y\psi^\theta(Z)] \qquad \text{and} \qquad g \doteq \mathbb{E}[Y\psi^\theta(Z)].$$

We have

$$\|\hat{h}_\theta - h_\theta\|_{L_2(X)} = \left\| C_{X,\theta}^{1/2} \left( \hat{\beta}_\theta - \beta_\theta \right) \right\|_{\ell_2} \leq \underbrace{\left\| C_{X,\theta}^{1/2} \hat{C}_{ZX,\theta}^{-1} C_{Z,\theta}^{1/2} \right\|_{\text{op}}}_{A_1} \underbrace{\left\| C_{Z,\theta}^{-1/2} \left( \hat{g} - \hat{C}_{ZX,\theta} \beta_\theta \right) \right\|_{\ell_2}}_{A_2}.$$

Lemma D.3 below guarantees that $\mathbb{P}(A_1 \leq 2/c_{\varphi_\theta^{(d)}, \psi_\theta^{(d)}}) \geq 1 - \tau/2$. Lemma D.4 below guarantees with probability at least $1 - \tau/2$ that

$$A_2 \leq \frac{c}{\sqrt{n}} \sqrt{d \, \sigma_{\text{eff}}^2 + \frac{d \, \rho^2}{n}} \, \log \frac{4}{\tau}.$$

where $\sigma_{\text{eff}}^2 \doteq \|h_0 - h_\theta\|_{L_2(X)}^2 + \sigma_U^2$.

A union bound gives the result with an absolute constant $c > 0$ possibly different from the previous.

$\square$

**Lemma D.3** (Matrix Perturbation Control). *Let Assumptions 2 and 3 be satisfied. Assume in addition that*

$$n \geq 16 \frac{d\rho^2 \log(4d/\tau)}{c_{\varphi_\theta^{(d)}, \psi_\theta^{(d)}}^2}. \tag{17}$$

*Then with probability at least $1 - \tau/2$,*

$$\left\| C_{X,\theta}^{1/2} \hat{C}_{ZX,\theta}^{-1} C_{Z,\theta}^{1/2} \right\|_{\text{op}} \leq \frac{2}{c_{\varphi_\theta^{(d)}, \psi_\theta^{(d)}}}.$$

*Proof of Lemma D.3.* Define the centered random matrix

$$\eta_i \doteq C_{Z,\theta}^{-1/2}(\psi_\theta(Z_i)\varphi_\theta(X_i)^\intercal - C_{ZX,\theta})C_{X,\theta}^{-1/2}.$$

We have

$$C_{Z,\theta}^{-1/2}(\hat{C}_{ZX,\theta} - C_{ZX,\theta})C_{X,\theta}^{-1/2} = \frac{1}{n} \sum_{i=1}^n \eta_i.$$

**Operator norm bound:** By Assumption 2, we have $\|\eta_i\|_{\text{op}} \leq d\rho^2 + 1$ almost surely. Indeed

$$\begin{aligned}
\|\eta_i\|_{\text{op}} &= \left\| C_{Z,\theta}^{-1/2}\psi_\theta(Z_i)\varphi_\theta(X_i)^\intercal C_{X,\theta}^{-1/2} - C_{Z,\theta}^{-1/2} C_{ZX,\theta} C_{X,\theta}^{-1/2} \right\|_{\text{op}} \\
&\leq \|C_{Z,\theta}^{-1/2}\psi_\theta(Z_i)\|_{\ell_2}\|C_{X,\theta}^{-1/2}\varphi_\theta(X_i)\|_{\ell_2} + \left\| C_{Z,\theta}^{-1/2} C_{ZX,\theta} C_{X,\theta}^{-1/2} \right\|_{\text{op}} \\
&\leq \sqrt{d}\rho \cdot \sqrt{d}\rho + 1 \leq d\rho^2 + 1 \qquad a.s.
\end{aligned}$$

**Covariance bounds:** For the left covariance,

$$\mathbb{E}[\eta_i \eta_i^*] = C_{Z,\theta}^{-1/2} \mathbb{E}[(\psi_\theta(Z)\varphi_\theta(X)^\top - C_{ZX,\theta})C_{X,\theta}^{-1}(\psi_\theta(Z)\varphi_\theta(X)^\top - C_{ZX,\theta})^*]C_{Z,\theta}^{-1/2}$$

$$= C_{Z,\theta}^{-1/2} \mathbb{E}[\psi_\theta(Z)\varphi_\theta(X)^\top C_{X,\theta}^{-1}\varphi_\theta(X)\psi_\theta(Z)^\top - C_{ZX,\theta}C_{X,\theta}^{-1}C_{ZX,\theta}^*]C_{Z,\theta}^{-1/2}$$

$$\preceq C_{Z,\theta}^{-1/2} \mathbb{E}[\psi_\theta(Z)\varphi_\theta(X)^\top C_{X,\theta}^{-1}\varphi_\theta(X)\psi_\theta(Z)^\top]C_{Z,\theta}^{-1/2}$$

$$= C_{Z,\theta}^{-1/2} \mathbb{E}[\psi_\theta(Z)\psi_\theta(Z)^\top \varphi_\theta(X)^\top C_{X,\theta}^{-1}\varphi_\theta(X)]C_{Z,\theta}^{-1/2}$$

$$\preceq d\rho^2 \cdot C_{Z,\theta}^{-1/2} C_{Z,\theta} C_{Z,\theta}^{-1/2} = d\rho^2 I_d.$$

Similarly, $\mathbb{E}[\eta_i^* \eta_i] \preceq d\rho^2 I_d$.

Thus $\sigma_L^2 = \sigma_R^2 \le nd\rho^2$. Applying Theorem D.7 gives w.p.a.l. $1 - \tau/2$

$$\left\|C_{Z,\theta}^{-1/2}(\widehat{C}_{ZX,\theta} - C_{ZX,\theta})C_{X,\theta}^{-1/2}\right\|_{\mathrm{op}} \le \sqrt{\frac{2d\rho^2 \log(4d\tau^{-1})}{n}} + \frac{1 + d\rho^2}{3}\frac{\log(4d\tau^{-1})}{n} =: \Delta_n(\tau).$$

Using Weyl's inequality (Proposition A.1) and Assumption 3, we deduce that the smallest singular of $C_{Z,\theta}^{-1/2}\widehat{C}_{ZX,\theta}C_{X,\theta}^{-1/2}$ satisfies

$$\sigma_{\min}(C_{Z,\theta}^{-1/2}\widehat{C}_{ZX,\theta}C_{X,\theta}^{-1/2}) \ge c_{\varphi_\theta^{(d)}, \psi_\theta^{(d)}} - \Delta_n(\tau) \ge c_{\varphi_\theta^{(d)}, \psi_\theta^{(d)}}/2 > 0,$$

where the last inequality follows from sample complexity condition Eq. (17). Since $\|A\|_{\mathrm{op}} = \sigma_{\min}^{-1}(A^{-1})$, we get the result. $\qquad\square$

**Lemma D.4** (Vector Concentration with Sub-Gaussian variables). *Let the assumptions of Theorem 5.1 be satisfied. Define*

$$\xi \doteq (Y - h_\theta(X))C_{Z,\theta}^{-1/2}\psi_\theta(Z).$$

*Then there exists an absolute constant $c > 0$ such that, with probability at least $1 - \tau/2$,*

$$\left\|\frac{1}{n}\sum_{i=1}^n \xi_i - \mathbb{E}[\xi]\right\|_{\ell_2} \le c\sqrt{\frac{d}{n}}\sqrt{\sigma_{\mathit{eff}}^2 + \frac{\rho^2}{n}}\log\frac{4}{\tau},$$

*where $\sigma_{\mathit{eff}}^2 \doteq \|h_0 - h_\theta\|_{L_2(X)}^2 + \sigma_U^2$.*

*Proof.* We need to check the moment condition of Proposition D.6 with $A = \xi$. This condition is obviously satisfied for $p = 2$ with $\sigma^2 = \mathbb{E}\|A\|_{\ell_2}^2$. Next for any $p \ge 3$, the Cauchy-Schwarz inequality and the equivalence of moment property give

$$\mathbb{E}\|\xi\|_{\ell_2}^m \le \left(\mathbb{E}\|\xi\|_{\ell_2}^4\right)^{1/2}\left(\mathbb{E}\|\xi\|_{\ell_2}^{2(m-2)}\right)^{1/2} \lesssim \mathbb{E}\left[\|\xi\|_{\ell_2}^2\right]\left(\mathbb{E}\|\xi\|_{\ell_2}^{2(m-2)}\right)^{1/2}.$$

For the second order moment, using Cauchy-Schwarz and the equivalence of moments for sub-Gaussian random variables:

$$\mathbb{E}[\|\xi\|_{\ell_2}^2] = \mathbb{E}[(h_0(X) - h_\theta(X) + U)^2\|C_{Z,\theta}^{-1/2}\psi_\theta(Z)\|_{\ell_2}^2]$$

$$\le \mathbb{E}^{1/2}\left[(h_0(X) - h_\theta(X) + U)^4\right]\mathbb{E}^{1/2}\left[\|C_{Z,\theta}^{-1/2}\psi_\theta(Z)\|_{\ell_2}^4\right]$$

$$\le 32\,\mathbb{E}_Z\left[\|C_{Z,\theta}^{-1/2}\psi_\theta(Z)\|_{\ell_2}^2\right]\sigma_{\mathrm{eff}}^2 = 32d\,\sigma_{\mathrm{eff}}^2.$$

For higher order moments, we apply Lemma D.5 with $v = Y - h_\theta(X)$ and $V = C_{Z,\theta}^{-1/2}\psi_\theta(Z)$. Hence we get, for any $p \ge 3$,

$$\mathbb{E}^{1/2}[\|\xi\|_{\ell_2}^{2(p-2)}] \le \sqrt{2}\,\overline{\sigma}_{Y-h_\theta(X)}^{p-2}\,(\rho\sqrt{d})^{p-2}\sqrt{(p-2)!}$$

By Definition D.1 of sub-Gaussian distributions, there exists an absolute constant $c > 0$ such that

$$\overline{\sigma}^2_{Y - h_\theta(X)} \leq 2\, c \left( \text{Var}\big(h_0(X) - h_\theta(X)\big) + \text{Var}(U) \right) \leq 2\, c\, \sigma^2_{\text{eff}}.$$

We apply Proposition D.6 to get w.p.a.l. $1 - \tau/2$

$$\left\| \frac{1}{n} \sum_{i \in [n]} \xi_i - \mathbb{E}[\xi] \right\|_{\ell_2} \leq c \sqrt{\frac{d}{n}} \sqrt{\sigma^2_{\text{eff}} + \frac{\rho^2}{n}} \, \log \frac{4}{\tau}.$$

where $c > 0$ is absolute constant possibly different from the previous display.

$\square$

**Lemma D.5** (Moment Bound for Sub-Gaussian Product). *Let $Z = vV$ where $v$ is a real-valued sub-Gaussian random variable with proxy variance $\overline{\sigma}_v$, and $V$ is a $d$-dimensional random vector with $\|V\|_{\ell_2} \leq \rho\sqrt{d}$ almost surely. Then for any integer $p \geq 3$, we have*

$$\mathbb{E}^{1/2}[|v|^{2(p-2)} \|V\|^{2(p-2)}_{\ell_2}] \leq \sqrt{2}\, \overline{\sigma}_v^{p-2} \, (\rho\sqrt{d})^{p-2} \sqrt{(p-2)!}$$

*Proof.* Since $\|V\|_{\ell_2} \leq \rho\sqrt{d}$ almost surely, we have

$$\mathbb{E}[|v|^{2(p-2)} \|V\|^{2(p-2)}_{\ell_2}] \leq (\rho\sqrt{d})^{2(p-2)} \mathbb{E}[|v|^{2(p-2)}].$$

We will use the tail characterization of sub-Gaussian random variables and the integral representation of moments to bound $\mathbb{E}[|v|^{2(p-2)}]$.

Since $v$ is sub-Gaussian with proxy variance $\overline{\sigma}_v$, there exists an absolute constant $c > 0$ such that for all $t \geq 0$:

$$\mathbb{P}(|v| \geq t) \leq 2 \exp\left( -\frac{t^2}{\overline{\sigma}^2_v} \right).$$

Set $m = 2(p-2)$. Using the integral representation of moments:

$$\mathbb{E}[|v|^{2(p-2)}] = \int_0^\infty m t^{m-1} \mathbb{P}(|v| \geq t)\, dt \leq 2 \int_0^\infty m\, t^{m-1} \exp\left( -\frac{t^2}{\overline{\sigma}^2_v} \right) dt.$$

Integration by parts gives

$$\int_0^\infty m\, t^{m-1} \exp\left( -\frac{t^2}{\overline{\sigma}^2_v} \right) dt = \frac{m}{2} \overline{\sigma}_v^m \, \Gamma\left( \frac{m}{2} \right)$$

where $\Gamma$ is the Gamma function.

Since $m = 2(p-2)$, we have $\Gamma\left( \frac{m}{2} \right) = (p-3)!$ and consequently

$$\int_0^\infty m\, t^{m-1} \exp\left( -\frac{t^2}{\overline{\sigma}^2_v} \right) dt = (p-2)!$$

Thus we get:

$$\mathbb{E}^{1/2}[|v|^{2(p-2)} \|V\|^{2(p-2)}_{\ell_2}] \leq \sqrt{2}(\overline{\sigma}_v \rho\sqrt{d})^{(p-2)} \sqrt{(p-2)!}$$

$\square$

### D.3. Concentration inequalities

We present here some well-known concentration inequalities for operators that we use in our analysis.

We recall first a version of Bernstein inequality, due to Pinelis and Sakhanenko, for random variables in a separable Hilbert space, see Caponnetto & De Vito (2007, Proposition 2).

**Proposition D.6.** *Let $A_i$, $i \in [n]$ be i.i.d copies of a random variable $A$ in a separable Hilbert space with norm $\|\cdot\|$. If there exist constants $\Lambda > 0$ and $\sigma > 0$ such that for every $m \geq 2$, $\mathbb{E}\|A\|^m \leq \frac{1}{2}m!\Lambda^{m-2}\sigma^2$, then with probability at least $1 - \delta$,*

$$\left\| \frac{1}{n} \sum_{i \in [n]} A_i - \mathbb{E}A \right\| \leq \frac{4\sqrt{2}}{\sqrt{n}} \log \frac{2}{\delta} \sqrt{\sigma^2 + \frac{\Lambda^2}{n}}.$$

The following result is called the noncommutative Bernstein inequality. It was first derived by Ahlswede & Winter (2002). The following version can be found in Tropp (2015).

**Theorem D.7** (Matrix Bernstein Inequality, Tropp (2015)). *Let $\{A_k\}_{k=1}^n$ be independent random matrices of size $d_1 \times d_2$ with $\mathbb{E}[A_k] = 0$. Assume that $\|A_k\|_{\mathrm{op}} \leq R$ almost surely for all $k$. Define the matrix variance parameters*

$$\sigma_L^2 \doteq \left\| \sum_{k=1}^n \mathbb{E}[A_k A_k^*] \right\|_{\mathrm{op}}, \qquad \sigma_R^2 \doteq \left\| \sum_{k=1}^n \mathbb{E}[A_k^* A_k] \right\|_{\mathrm{op}}.$$

*Then for any $t \geq 0$,*

$$\mathbb{P}\left\{ \left\| \sum_{k=1}^n A_k \right\|_{\mathrm{op}} \geq t \right\} \leq (d_1 + d_2) \exp\left( \frac{-t^2/2}{\max\{\sigma_L^2, \sigma_R^2\} + Rt/3} \right). \tag{18}$$

A more convenient and equivalent of Eq. (18) is, for any $\tau \in (0, 1)$, w.p.a.l. $1 - \tau$

$$\left\| \frac{1}{n} \sum_{k=1}^n A_k \right\|_{\mathrm{op}} \leq \sqrt{\frac{2 \max\{\sigma_L^2, \sigma_R^2\} \log\left(\frac{d_1+d_2}{\tau}\right)}{n^2}} + \frac{R}{3} \frac{\log\left(\frac{d_1+d_2}{\tau}\right)}{n}.$$

## E. Experimental Details

The code needed to reproduce all figures in this work can be found at https://github.com/Wornbard/AugSpecIV.

### E.1. Structural functions and alignment in dSprites

The dSprites structural function $h_0 = h_{\mathrm{old}}$, as used in, e.g., Xu et al. (2024) is not a new idea. However, it differs from an alternative on which SpecIV(Sun et al., 2025) and DFIV(Xu et al., 2021) were originally evaluated. That one takes the form,

$$h_0(x) = (\|BX\|^2 - 5000)/1000$$

with $B \in \mathbb{R}^{10 \times 4096}$ consisting of i.i.d. $\mathrm{Uniform}([0, 1])$ entries. This function evaluates the squared norm of 10 random linear measurements of the image and returns a linear transformation of it. Since the entries of $B$ are i.i.d., this $h_0$ has no clear dependence on the values of $Z$. To see that it should be extremely difficult to recover $h_0$ from the relation $\mathcal{T}h_0 = r_0$, note that moving the sprite vertically, while keeping the orientation, scale and $x$-position constant (i.e. not changing the instrumental variable) can lead to different values of $h_0$. This does not quite imply that $h_0$ has a non-trivial projection onto the kernel of $X$ because the distribution of the images $X$ is not vertical-shift-invariant. But it does suggest that the instrument $Z$ is essentially uninformative about an important property of the image needed to recover $h_0$. One can reliably learn part of $h_0$ due to its monotonicity in the sprite's scale but beyond this, any dependence on $x$-position and orientation is likely to be extremely irregular and not representative of real-life applications of IV regression. For these reasons we choose not to utilize this benchmark.

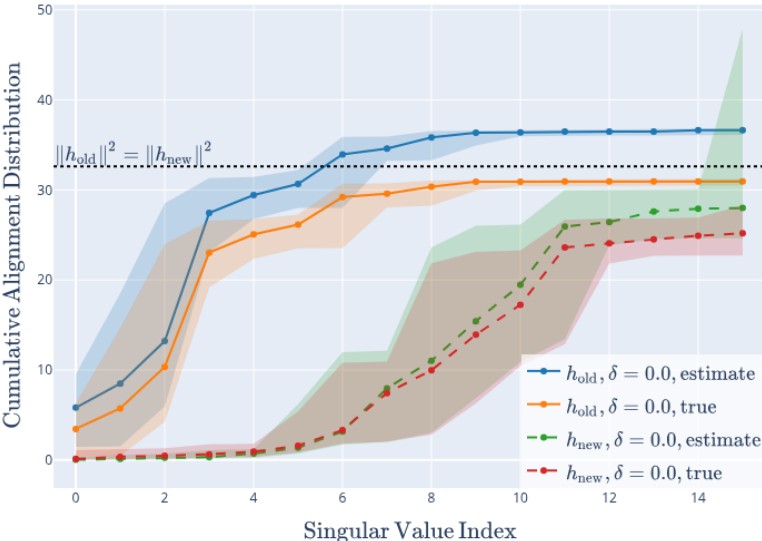

*Figure 5.* Distributions of cumulative alignment estimates and true values $\|\Pi_{\hat{v}(i)} h_0\|^2$ for increasing $i$, evaluated on separately fitted models (with identical parameters) for $h_0 = h_{\text{old}}, h_{\text{new}}$ at $\delta = 0$.

### E.2. Spectral alignment in dSprites

**Standard dSprites is well-aligned.** Intuitively, most of the variability in the image should be explained by the heart's position and scale since these quantities determine where the non-zero pixels are and how many of them there are in total. By contrast, the rotation angle only explains the location of non-zero pixels on the boundary of the broad region where the sprite is located (since most of the sprite's area is preserved by rotations around its center). Therefore, structural functions that mostly depend on scale and $x$-position should intuitively be easier to learn than those that are sensitive to rotation. Since $A$ in the dSprites structural function "measures" distance of the non-zero pixels from the vertical central bar in the image, it should be mostly sensitive to $x$-positions and the number of non-zero points to sum over.

Moreover, following Meunier et al. (2026), we can compute the empirical-distribution-based singular value decomposition of a fitted $\hat{\mathcal{T}}_d$ for $\delta = 0$ (which we just treat as an operator $L_2(X) \to L_2(Z)$). That is, we compute features $\hat{u}_i, \hat{v}_i$ and positive real numbers $\hat{\sigma}_i$ such that,

$$\hat{\mathcal{T}}_d = \sum_{i=1}^{d} \hat{\sigma}_i \hat{u}_i \otimes \hat{v}_i.$$

The feature functions $(\hat{u}_{1:d}), (\hat{v}_{1:d})$ are orthonormal systems with respect to the empirical distributions of $Z, X$ based on the samples used for the SVD estimation. If $\left\|\hat{\mathcal{T}}_d - \mathcal{T}_d\right\|_{\text{op}}$ is sufficiently small, we are guaranteed that the projections onto singular features $\hat{v}_i$ are close to the projections onto the singular features of the true truncated conditional mean operator $\mathcal{T}_d$ (Meunier et al., 2026).

We compute the individual-feature projection lengths $|\langle \hat{v}_i, h_0 \rangle|$ to conclude that most of $h_{\text{old}}$ is supported on the leading singular functions whose singular values are close to 1. Moreover, we compare these true alignment values to their estimates that are not based on $h_0$ (Meunier et al., 2026). This is to ensure that spectral (mis-)alignment in the dSprites setting can also be detected in a real-life setting where the structural function is not available. Denote the projection onto the leading $i$ singular $X$-features of $\hat{\mathcal{T}}_d$ with $\Pi_{\hat{v}(i)}$. As seen in Figure 5, both the estimates of the squared projection length onto the leading $i$ features of $\hat{\mathcal{T}}_d$ and the true values $\|\Pi_{\hat{v}(i)} h_0\|^2$ are in agreement with one another and indicate that $h_{\text{old}}$ lies in the top of the spectrum $\mathcal{T}$.

**A more difficult structural function.** Inspired by the discussion above, we would like to evaluate our methods in a more challenging setting. An intuitive method of obtaining such a benchmark is constructing a structural function which is most sensitive to the rotation of the sprite as opposed to its position or scale. In order to work with shapes whose orientation is easier to estimate from the image, we replace hearts with ellipses in our benchmark. For an image $X$ of an ellipse, let $\tilde{X}$ be

the result of convolving it with a smoothing gaussian kernel with a small bandwidth of around 4 pixels. This is done in order to decrease our method's sensitivity to noise. Then let $x_{\mathrm{max},i} \doteq \max_{j \in \{1,\ldots,64\}} \tilde{X}_{ji}$ be the maximum of $\tilde{X}$ along its vertical bars, and $y_{\mathrm{max},i} \doteq \max_{j \in \{1,\ldots,64\}} \tilde{X}_{ij}$ be the corresponding maximum over horizontal bars. Then, one expects the norms of $x_{\mathrm{max}}$ and $y_{\mathrm{max}}$ to be roughly proportional to the length of the ellipses projection onto the horizontal and vertical axes of the image. Hence, taking

$$h_{\mathrm{new}} \doteq a \cdot (\|x_{\mathrm{max}}\|_{\ell_2}/\|y_{\mathrm{max}}\|_{\ell_2} - b),$$

one should obtain a function which depends on the ellipse's orientation and is insensitive to its scale or position. The scalars $a, b$ are chosen in order to match the first two moments of the original structural function $h_{\mathrm{old}}$. This is done in order to not artificially change the downstream problem's difficulty by making the norm of the signal relative to noise different.

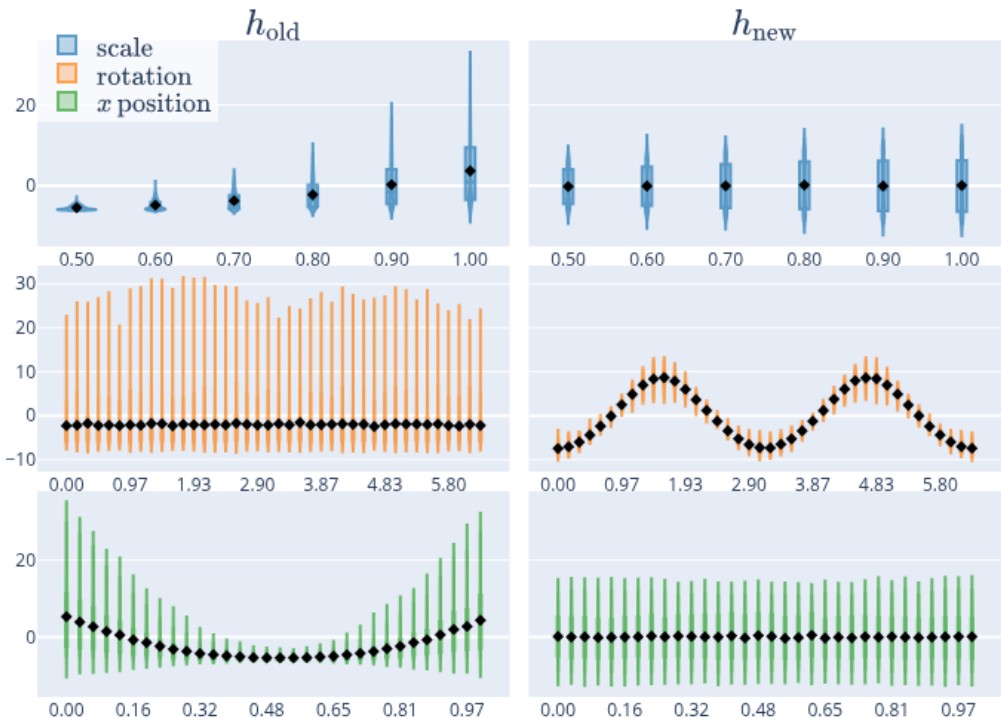

*Figure 6.* Comparison of how the distributions of $h_{\mathrm{old}}$ and $h_{\mathrm{new}}$ vary with each component of the instrument $Z$ (scale, orientation or $x$ position). The values of each component of $Z$ in dSprites are quantized. The $x$-axis positions in the figure correspond to those values. For each value of a component of $Z$, we display the distribution of the values of $h_0 = h_{\mathrm{old}}, h_{\mathrm{new}}$ evaluated on images where the component takes that value. The marked values are the means of each bin.

As seen in Figure 6, while $h_{\mathrm{old}}$ has a clear dependence on the sprite position and orientation, $h_{\mathrm{new}}$ is only visibly sinusoidal in its orientation and insensitive to the latter two. Figure 5 confirms that data generated using $h_{\mathrm{new}}$ should be considerably more challenging for spectral-feature-based IV regression, since its projection onto the leading singular functions is close to zero for all models with $\delta = 0$ that we have fitted. It is supported further in the spectrum, and in fact $\|\Pi_{\hat{v}(d)} h_{\mathrm{new}}\|^2$ is far from $\|h_{\mathrm{new}}\|^2$. This means that not even all of the learned features span $h_{\mathrm{new}}$.

### E.3. Synthetic data models

We train our models using the same architecture as Meunier et al. (2026), which is described in Table 1. Notably, the first layer of this network utilizes the $\sin(x)^2 + x$ activation proposed by Ziyin et al. (2020), which enables it to learn the oscillatory basis functions more reliably. To make the improvements in the alignment of leading features with $\delta$ clearer, we learn 10 $Z-$ and $X-$ features on synthetic datasets with $d = 11$. The models are trained on 50000 $(Z, X, Y)$ samples using the Adam algorithm.

Evaluation of the benefits of using a positive $\delta$ on a bigger range of $c_\sigma$ values than in the main body of the work can be found in Figure 8

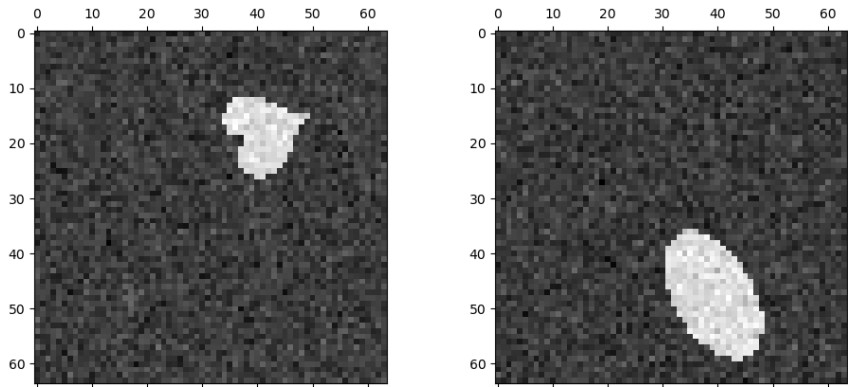

*Figure 7.* Examples of sprite images on which $h_{\text{old}}$ (left) and $h_{\text{new}}$ (right) are evaluated.

*Figure 8.* Distributions of relative IV regression MSEs for the synthetic example with $\delta \in \{0, 0.5, 1.0, 3.0, 5.0\}$.

| Layer | Configuration |
|-------|---------------|
| 1 | Input: 1 |
| 2 | FC(1,50), $x \mapsto x + \sin^2 x$ |
| 3 | FC(50, 50), GeLU |
| 4 | FC(50, 10) |

*Table 1.* $Z, X$ feature networks for synthetic data

### E.4. Spectral learning models for dSprites

For learning spectral features we utilize nearly the same models as Sun et al. (2025). The only difference is replacing ReLU activations with GeLU which we found to lead to slightly easier training. "BN" in Table 2 refers to Batch Normalization (Ioffe & Szegedy, 2015). The models are trained on 25000 $(Z, X, Y)$ samples and optimized with the Adam algorithm.

We observed that feature models utilizing 32 features, as originally proposed in Sun et al. (2025), were very prone to overfitting, often before this became apparent in the observed loss, making early stopping difficult to implement. Hence, we trained the models with 16 features instead. This led to better performance across the board for $h_{old}$. In $h_{new}$, vanilla SpecIV, trained with 32 features attained a smaller loss than it did for 16. However, the performance was still significantly below that of DFIV or spectral learning with a non-zero $\delta$. For all the "optimal" values of $\delta$ (between 0.1 and 1.0) we observed benefits from using 16 features over 32.

| Layer | Configuration |
|-------|---------------|
| 1 | Input: 4096 |
| 2 | FC(4096,1024), BN, GeLU |
| 3 | FC(1024,512), BN, GeLU |
| 4 | FC(512,128), BN, GeLU |
| 5 | FC(128,16) |

$X$ - features

| Layer | Configuration |
|-------|---------------|
| 1 | Input: 3 |
| 2 | FC(3,256), BN, GeLU |
| 3 | FC(256,128), BN, GeLU |
| 4 | FC(128,128), BN, GeLU |
| 5 | FC(128,16) |

$Z$ - features

*Table 2.* Architectures of spectral learning networks for dSprites datasets

### E.5. DFIV models

For the comparison to DFIV, which is the only viable competitor to SpecIV, we utilize the same architecture as proposed in the original work (Xu et al., 2021). "SN" in Table 3 refers to spectral normalization proposed by Miyato et al. (2018) and "BN" is Batch Normalization. The models are trained on 25000 $(Z, X, Y)$ samples and optimized with the Adam algorithm.

Since we decided to decrease the number of features employed by spectral learning, relative to the standard architecture used in the literature, we also investigated whether doing so leads to better performance in DFIV. The opposite was observed and hence we retain the original models.

| Layer | Configuration |
|-------|---------------|
| 1 | Input: 4096 |
| 2 | FC(4096,1024), SN, ReLU |
| 3 | FC(1024,512), SN, ReLU, BN |
| 4 | FC(512,128), SN, ReLU |
| 5 | FC(128,32), SN, BN, Tanh |

$X$ - features

| Layer | Configuration |
|-------|---------------|
| 1 | Input: 3 |
| 2 | FC(3,256), SpectralNorm, ReLU |
| 3 | FC(256,128), SN, ReLU, BN |
| 4 | FC(128,128), SN, ReLU, BN |
| 5 | FC(128,32), BN, ReLU |

$Z$ - features

*Table 3.* Architectures of DFIV networks for dSprites datasets

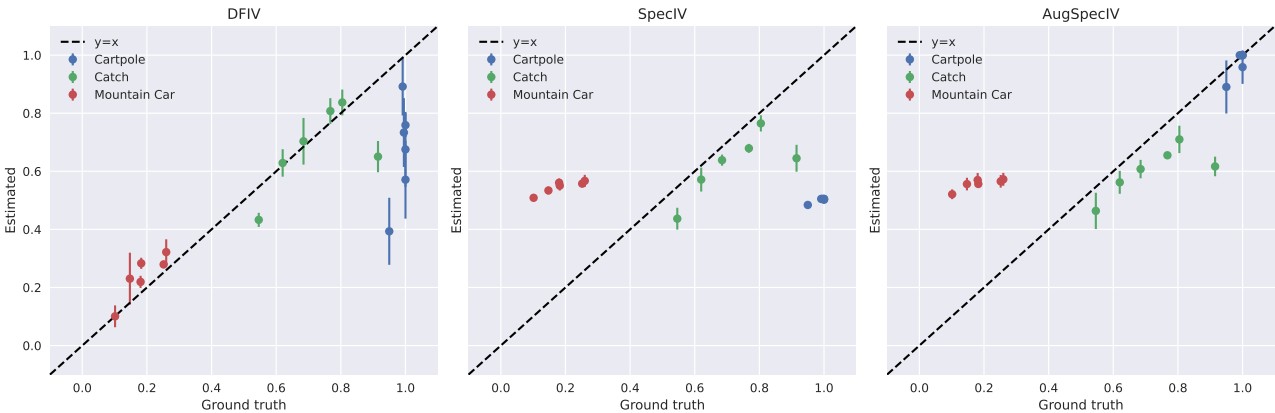

*Figure 9.* Scatter plot of estimated policy values vs. ground truth.

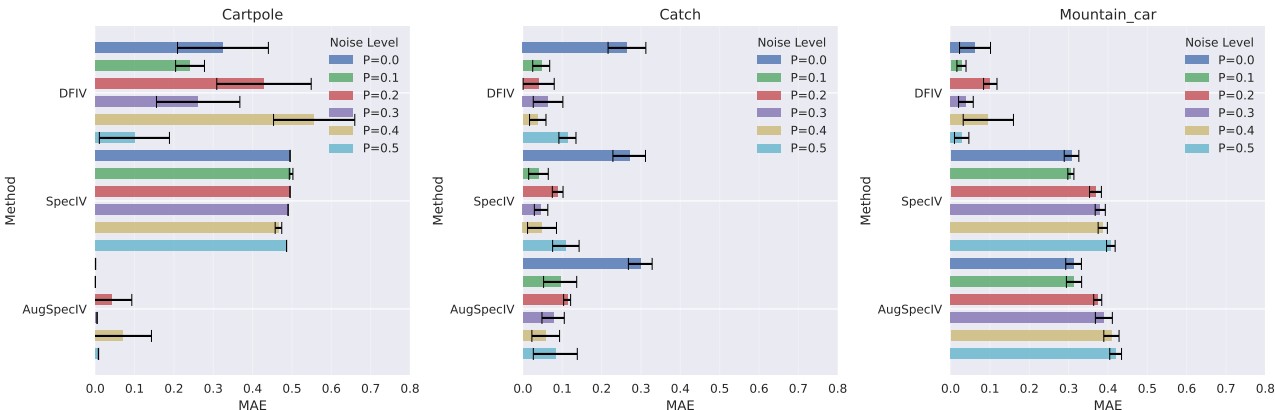

*Figure 10.* Box plot of mean absolute error (MAE) between policy value and ground-truth.

### E.6. Comparison to KIV

Since Kernel IV (KIV) proposed by Singh et al. (2019) is a very popular and easily applicable method, we also include it in our benchmarks. We evaluated it using the Gaussian kernel with a bandwidth proportional to $m\sqrt{d}$ where $m$ is the median distance between pairs samples on which the kernel is evaluated, and $d$ is the dimension of the samples (i.e. 3 for $Z$ and 4096 for $X$).

### E.7. OPE Experiment Protocol and Results

**Hyperparameter optimization.** For each method and task, we randomly sample 100 hyperparameter combinations from the grids in Tables 4 and 5, and select the configuration achieving the lowest stage-2 mean squared error on the corresponding task with $p = 0.2$. Orthonormal regularization is applied to both spectral methods, while the $\delta$ parameter is specific to AugSpecIV, i.e., $\delta = 0$ for SpecIV and is tuned as any other hyperparameter for AugSpecIV. With the chosen hyperparameters, we run each method five times across all tasks and noise levels and report the mean and standard deviation of the estimated policy values.

#### E.7.1. OPE CONTEXT: POLICIES AND DATA

In OPE, the "offline" constraint is critical:

- We cannot interact with the environment. We cannot take a state $s$ and an action $a \sim \pi$ to observe a new transition $(s, a, r, s')$.

- We can compute with $\pi$. Since $\pi$ is a known policy (e.g., a function in our code), we can sample actions $a' \sim \pi(a'|s')$

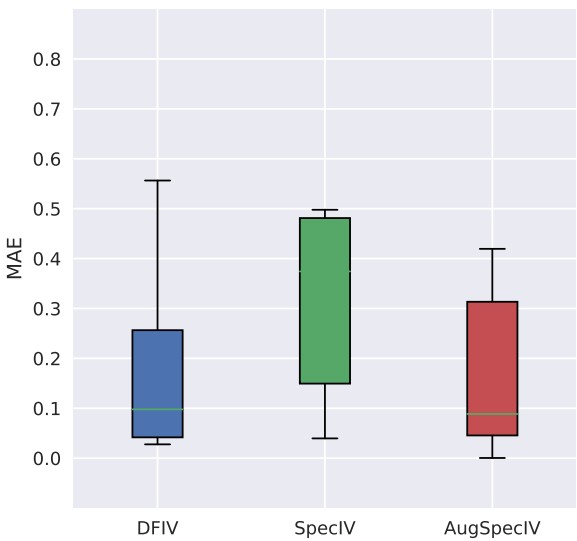

*Figure 11.* Distribution of MAE across NPIV methods.

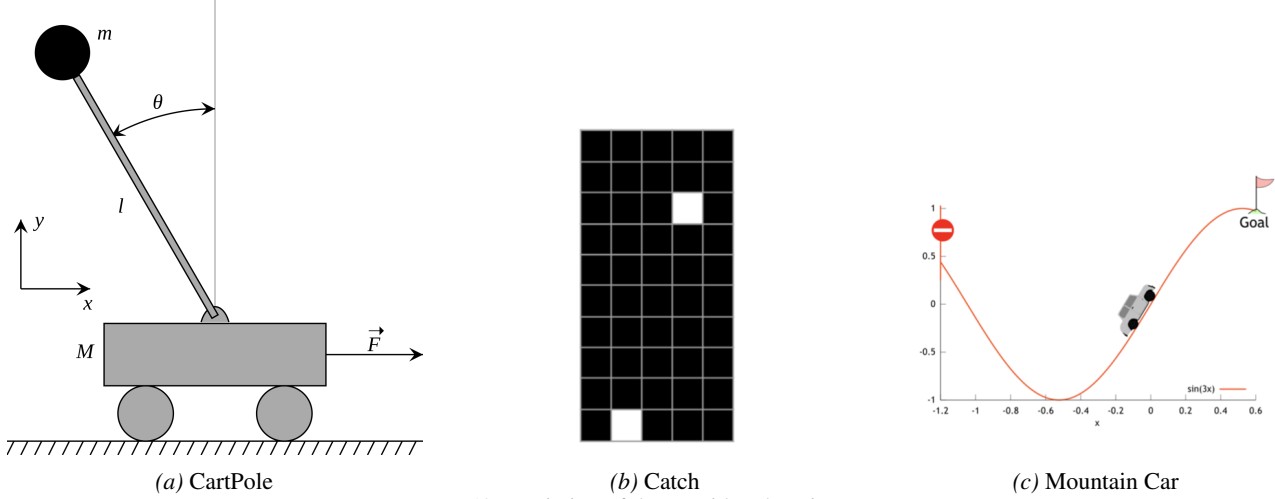

*(a)* CartPole       *(b)* Catch       *(c)* Mountain Car

*Figure 12.* Depiction of the considered environments.

| Hyperparameter | Values |
|---|---|
| Training Steps | $10^5$ |
| Batch Size | 2048 |
| Stage-1 reg. | $\{10^{-8}, 10^{-6}, 10^{-4}, 10^{-2}\}$ |
| Stage-2 reg. | $\{10^{-8}, 10^{-6}, 10^{-4}, 10^{-2}\}$ |
| Value reg. | $\{10^{-8}, 10^{-6}, 10^{-4}, 10^{-2}\}$ |
| Instrumental reg. | $\{10^{-8}, 10^{-6}, 10^{-4}, 10^{-2}\}$ |
| Value learning rate | $\{10^{-5}, 3{\cdot}10^{-5}, 10^{-4}, 3{\cdot}10^{-4}, 10^{-3}\}$ |
| Instrumental learning rate | $\{10^{-5}, 3{\cdot}10^{-5}, 10^{-4}, 3{\cdot}10^{-4}, 10^{-3}\}$ |
| $X$ net layer sizes | $(50, 50)$ |
| $Z$ net layer sizes | $\{(50, 50), (100, 100), (150, 150)\}$ |

*Table 4.* DFIV hyperparameters.

| Hyperparameter | Values |
|---|---|
| Training Steps | $10^4$ |
| Batch Size | 2048 |
| Stage-1 reg. | $\{10^{-8}, 10^{-6}, 10^{-4}, 10^{-2}\}$ |
| Stage-2 reg. | $\{10^{-8}, 10^{-6}, 10^{-4}, 10^{-2}\}$ |
| Orthonormal reg. | $\{10^{-8}, 10^{-6}, 10^{-4}, 10^{-2}\}$ |
| Learning rate | $\{10^{-5}, 3{\cdot}10^{-5}, 10^{-4}, 3{\cdot}10^{-4}, 10^{-3}\}$ |
| $X$ net layer sizes | $(50, 50)$ |
| $Z$ net layer sizes | $\{(50, 50), (100, 100), (150, 150)\}$ |
| $\delta$ | $\{10^{-3}, 10^{-2}, 10^{-1}, 1\}$ |

*Table 5.* SpecIV and AugSpecIV hyperparameters. The neural architectures are the same as those of DFIV described in (Chen et al., 2022), except for the usage of GELU activations.

for any state $s'$ that we observe in our dataset.

### E.7.2. NPIV FORMULATION OF OPE AND THE COMPACTNESS PROBLEM

In OPE, to estimate the expected return of a policy $\pi$, a common and effective approach consists of first estimating the Q-function $Q_\pi$, and then averaging over the initial state distribution. While early methods like Least Squares Temporal Difference (LSTD; Bradtke & Barto, 1996) pioneered this direction using linear function approximation, modern approaches increasingly leverage NPIV regression to handle general function approximation.

**Related work on OPE.** We briefly discuss existing methods for OPE and refer to Jiang & Xie (2025) and references therein for additional details. Early approaches such as LSTD were restricted to linear function approximation. For general function approximation, the dominant approach has historically been Fitted-Q Evaluation (FQE), which iteratively solves a regression problem to minimize the Bellman residual. Recent advances have highlighted that such an approach requires strong assumptions to succeed, namely: having access to a sufficiently expressive function class (e.g., that is realizable and Bellman complete) and using data with good coverage. NPIV formulations offer an alternative perspective to these regression-based methods. Rather than minimizing a squared error directly, NPIV approaches (Hu et al., 2025) frame the Bellman equation as a conditional moment restriction as shown below.

$Q$ **function and NPIV.** Let $(\mathcal{S}, \mathcal{A}, P, R, \mu_0)$ be a Markov Decision Process (MDP) with state space $\mathcal{S}$, action space $\mathcal{A}$, transition kernel $P(s' \mid s, a)$, reward distribution $P_{rew}(r \mid s, a)$, and initial distribution $\mu_0$. We denote by $R$ the random reward variable such that its distribution given any state-action pair $(s, a)$ is $P_{rew}(\cdot \mid s, a)$.

Given a target policy $\pi$ and discount factor $\gamma \in (0, 1)$, its value is defined as:

$$\rho(\pi) = \mathbb{E}\left[\sum_{t=0}^{\infty} \gamma^t R_t\right] = \mathbb{E}_{S_0 \sim \mu_0, A_0 \sim \pi(S_0)}[Q_\pi(S_0, A_0)], \tag{19}$$

where $R_t \sim P_{rew}(\cdot \mid S_t, A_t)$, $(S_t, A_t)$ follows $\pi$ and $P$, and $Q_\pi$ is the state-action value function (i.e. Q-function), given by

$$Q_\pi(s, a) = \mathbb{E}\left[\sum_{t=0}^{\infty} \gamma^t R_t \mid S_0 = s, A_0 = a\right].$$

The $Q$-function, $Q_\pi$, is the unique fixed point of the Bellman equation:

$$Q_\pi(s, a) = \mathbb{E}_{R \sim P_{rew}(\cdot|s,a)}[R] + \gamma \mathbb{E}_{S' \sim P(\cdot|s,a), A' \sim \pi(\cdot|S')}[Q_\pi(S', A')] \tag{20}$$

We first show that we can rewrite this Bellman Equation as an NPIV problem of the form Eq. (2). A given policy $\pi$ induces an occupancy measure $\mu_\pi$ on the state-action space $\mathcal{S} \times \mathcal{A}$. We then take $X = (S, A, S', A')$ equipped with the measure such that $(S, A) \sim \mu_\pi, S' \sim P(\cdot \mid S, A), A' \sim \pi(\cdot \mid S')$, $Z = (S, A)$ equipped with the measure such that $(S, A) \sim \mu_\pi$ and finally $Y = R$. We take $\mathcal{T}$ as defined in the main body so that Eq. (20) can be re-written as

$$\mathcal{T}h_\pi(S, A) = \mathbb{E}[R|S, A] \qquad \text{i.e.} \qquad \mathcal{T}h_\pi(Z) = \mathbb{E}[Y|Z],$$

where $h_\pi(X) = h_\pi(S, A, S', A') = Q_\pi(S, A) - \gamma Q_\pi(S', A')$. Note that solving NPIV with respect to $\mathcal{T}$ allows us to retrieve $Q_\pi$ on the support of $\mu_\pi$. However instead of using $\mu_\pi$ for which we don't have samples, we can use the logging policy. Recall the logging policy $\pi_b$ and denote by $\mu_b$ the induced state-action distribution. We introduce $\mathcal{T}_b$ as the standard conditional expectation operator but we now equip $X$ with $(S, A) \sim \mu_b, S' \sim P(\cdot \mid S, A), A' \sim \pi(\cdot \mid S')$ and $Z$ with $(S, A) \sim \mu_b$. The following equation

$$\mathcal{T}_b h_\pi(S, A) = \mathbb{E}[R|S, A] \qquad \text{i.e.} \qquad \mathcal{T}_b h_\pi(Z) = \mathbb{E}[Y|Z],$$

allows us to identify $h_\pi(X) = h_\pi(S, A, S', A') = Q_\pi(S, A) - \gamma Q_\pi(S', A')$ on the support $\mu_b$. Crucially, this identification relies on a coverage assumption (Jiang & Xie, 2025) that states that the support of the distribution $\mu_b$ covers the support of the target policy. Consider $\mathcal{Q}$ a function class to approximate $Q_\pi$. We can retrieve $Q_\pi$ by solving for an empirical version of the following loss

$$\arg\min_{q \in \mathcal{Q}} \mathbb{E}_{(S,A) \sim \mu_b, R \sim P_{rew}(\cdot|S,A)}[(Y - \mathcal{T}_b(h_q)(S, A))^2] \qquad h_q(s, a, s', a') = q(s, a) - \gamma q(s', a').$$

We can then plug our estimate into Eq. (19) to get an estimate of $\rho(\pi)$. Different methods have been employed with this loss where methods differ by their choice of $\mathcal{Q}$ and their approximation of $\mathcal{T}_b$, see Chen et al. (2022).

However, this formulation is not amenable to spectral feature methods since $\mathcal{T}_b$ is non-compact as $X$ contains $Z$, and thus cannot be learned using the usual contrastive loss. To overcome this, we instead recover $Q_\pi$ through a modified NPIV problem. This time consider $X = (S', A')$ equipped with the measure such that given $(s, a), S' \sim P(\cdot \mid s, a), A' \sim \pi(\cdot \mid S')$, $Z = (S, A)$ equipped with the measure such that $(S, A) \sim \mu_b$ and finally $Y(Q_\pi) = -\gamma^{-1}(R - Q_\pi(S, A))$. Define $\tilde{\mathcal{T}}q(Z) \doteq \mathbb{E}[q(S', A') \mid Z]$ so that

$$\mathbb{E}[Y(Q_\pi)|Z] = \tilde{\mathcal{T}}(Q_\pi)(Z)$$

This removes the direct shared-variable source of non-compactness and yields the operator we target with spectral feature learning, under the compactness assumption used throughout the paper. However, this modified formulation introduces a new challenge: the target outcome $Y(Q_\pi)$ depends on $Q_\pi$, which is the very function we are trying to estimate.

This necessitates an iterative procedure for $k = 0, 1, \ldots, K - 1$, similar to Fitted Q-Evaluation (FQE):

1. Start with an initial guess, $Q_0$ (e.g., $Q_0 = 0$).

2. At iteration $k + 1$, we solve the NPIV problem defined by $Q_k$:
   - We construct the target outcome $Y_k$ using our previous estimate $Q_k$ and the observed reward $r(S, A)$ from the data:
   $$Y_k(S, A) = -\frac{1}{\gamma}(r(S, A) - Q_k(S, A))$$
   - We solve the spectral NPIV problem $\mathbb{E}[Y_k \mid Z] = \tilde{\mathcal{T}}Q_{k+1}$ to find the new estimate $Q_{k+1}$.

3. Repeat until convergence.

This iterative process introduces a potential for dynamic spectral misalignment. The target $Y_k$ changes at every iteration $k$. If the spectral features required to estimate $Y_k$ (the "outcome-aware" direction) are not the same as the dominant spectral features of $\tilde{\mathcal{T}}$, or if this direction shifts as $Q_k$ converges, an outcome-agnostic method will fail. This is precisely the scenario where our **Augmented Spectral Feature Learning** could help.

### E.8. Validity of hyperparameter tuning by cross-validation on the 2SLS loss

The population level optimal estimator of $\mathcal{T}$ based on a fixed set of $\psi^{(d)}, \varphi^{(d)}$ features is $\Pi_{\psi^{(d)}}\mathcal{T}\Pi_{\varphi^{(d)}}$ where $\Pi_{\varphi^{(d)}}, \Pi_{\psi^{(d)}}$ denote the orthogonal projections onto the spans of the learned $X, Z$ features. By minimizing the contrastive loss we can ensure that our estimator is indeed close to having this structure, with whatever features are obtained in the process.

Given $\widehat{\mathcal{T}}$ an estimation of $\mathcal{T}$, consider the so-called stage-2 loss, $\ell(h) \doteq \mathbb{E}[(\widehat{\mathcal{T}}h(Z) - Y)^2]$ over $h$ spanned by the learned $X$-features. For simplicity assume it is performed with this population-level optimal estimate of the conditional mean $\widehat{\mathcal{T}} = \Pi_{\psi^{(d)}}\mathcal{T}\Pi_{\varphi^{(d)}}$, and let the resulting estimate of the structural function be $\widehat{h}$. We have,

$$\ell(\widehat{h}) = \mathbb{E}\left[(\mathcal{T}h_0(Z) - \Pi_{\psi^{(d)}}\mathcal{T}\Pi_{\varphi^{(d)}}\widehat{h}(Z))^2\right] + \mathbb{E}[(\mathcal{T}h_0(Z) - Y)^2].$$

The second term is a model-independent constant so we can disregard it. If we wanted to evaluate the quality of our model based on this loss, then we should be able to decompose it into a monotone function of $\|h_0 - \widehat{h}\|$ or (to make the task easier) of $\|\mathcal{T}(h_0 - \widehat{h})\|$ and a term that will not vary across different learned feature sets (or is sufficiently small to be negligible). However, there seems to be no clear way of achieving this. Consider the most natural approach below.

Note $\Pi_{\psi^{(d)}}\mathcal{T}\Pi_{\varphi^{(d)}}\widehat{h} = \Pi_{\psi^{(d)}}\mathcal{T}\widehat{h}$ since $\widehat{h}$ is spanned by the $X-$features. Let $\Pi_{\psi^{(d)}}^{\perp}$ be the projection onto the orthogonal complement of the span of the $\psi$ features. Then

$$\mathbb{E}\left[(\mathcal{T}h_0 - \Pi_{\psi^{(d)}}\mathcal{T}\widehat{h})^2\right] = \mathbb{E}\left[(\mathcal{T}(h_0 - \widehat{h}))^2\right] + \mathbb{E}\left[(\Pi_{\psi^{(d)}}^{\perp}\mathcal{T}\widehat{h})^2\right] + 2\mathbb{E}\left[\mathcal{T}(h_0 - \widehat{h}) \cdot \Pi_{\psi^{(d)}}^{\perp}\mathcal{T}\widehat{h}\right].$$

If we could argue that the latter two terms are negligible, then stage 2 error should reflect $\mathbb{E}[(\mathcal{T}(h_0 - \widehat{h}))^2]$ and we would indeed be done. This condition would be satisfied if one could for instance argue that $\left\|\Pi_{\psi^{(d)}}^{\perp}\mathcal{T}\Pi_{\varphi^{(d)}}\right\|_{\text{op}}$ is always small. But there is no clear reason why this should hold. We note that this indicates that minimization of the 2SLS loss is a generally unprincipled methodology, not only for tuning $\delta$ in our setting, but for IV model selection more broadly.

**Practical performance.**    Regardless of the aforementioned obstacles, we find that selecting the model, and, in particular, tuning $\delta$ based on $\ell(h)$, the stage-2 loss, yields good results. On the dSprites benchmarks these are consistent with the values selected by the procedure based on maximizing the estimated projection length of $h_0$ onto the learned $X-$features, as shown in Figure 13.

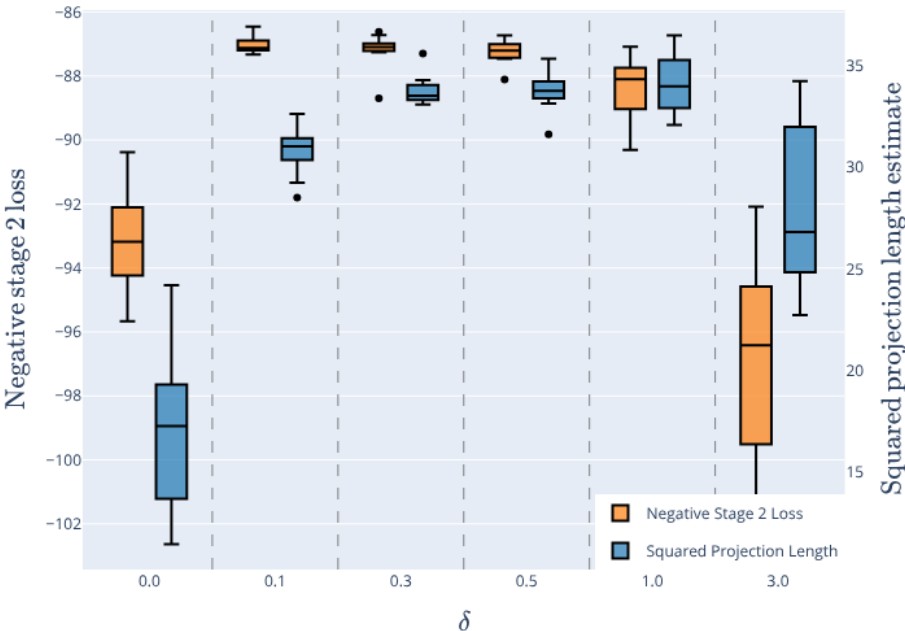

*Figure 13.* Comparison of the estimators of $\|\Pi_{\varphi_*^{(d)}}h_0\|^2$ proposed in Section 6.4 and negative stage-2 errors, for a range of $\delta$ values on the dSprites benchmark with $h_0 = h_{\text{new}}$. Despite the lack of theoretical backing for the latter, both methods attain their maxima near the same values of $\delta$.

### E.9. Hyperparameter tuning by maximizing estimated spectral alignment

Estimators of the alignment quantity in Proposition 6.1 provide a natural way to choose $\delta$. For a fitted augmented operator, let

$$\widehat{\mathcal{T}}_{\delta}^{(d)} = \sum_{i=1}^{d} \widehat{\sigma}_{\delta,i}\, \widehat{\psi}_{\delta,i} \otimes (\widehat{\varphi}_{\delta,i}, \widehat{\omega}_{\delta,i})$$

be its rank-$d$ empirical SVD, with the singular systems orthonormal in $L_2(\widehat{\mu}_Z)$ and $L_2(\widehat{\mu}_X) \times \mathbb{R}$, respectively. Define

$$\widehat{\alpha}_{\delta,i} = \widehat{\sigma}_{\delta,i}^{-1}\widehat{\mathbb{E}}\left[Y\widehat{\psi}_{\delta,i}(Z)\right], \qquad \widehat{\alpha}_{\delta} = (\widehat{\alpha}_{\delta,1}, \ldots, \widehat{\alpha}_{\delta,d})^{\top}, \qquad \widehat{\omega}_{\delta} = (\widehat{\omega}_{\delta,1}, \ldots, \widehat{\omega}_{\delta,d})^{\top}.$$

The plug-in version of Proposition 6.1 is then

$$\widehat{A}(\delta) = \widehat{\alpha}_\delta^\top \left(I_d - \widehat{\omega}_\delta \widehat{\omega}_\delta^\top\right)^{-1} \widehat{\alpha}_\delta \approx \|\Pi_{\varphi_\star^{(d)}} h_0\|_{L_2(X)}^2.$$

Thus one may tune $\delta$ by maximizing $\widehat{A}(\delta)$ over a candidate grid. The caveat is numerical: the inverse above becomes ill-conditioned when $\|\widehat{\omega}_\delta\|_{\ell_2}$ is close to one, since $\lambda_{\min}(I_d - \widehat{\omega}_\delta \widehat{\omega}_\delta^\top) = 1 - \|\widehat{\omega}_\delta\|_{\ell_2}^2$.

If $P_\delta$ denotes the rank-$d$ left singular projection of $\mathcal{T}_\delta$, then, since $\mathcal{T}_\delta \mathcal{T}_\delta^\star = \mathcal{T}\mathcal{T}^\star + \delta^2 r_0 \otimes r_0$,

$$P_\delta \in \arg\max_{\substack{P = P^\star = P^2 \\ \mathrm{rank}(P) = d}} \left\{ \mathrm{Tr}(P\mathcal{T}\mathcal{T}^\star) + \delta^2 \|Pr_0\|_{L_2(Z)}^2 \right\}.$$

Consequently, if $0 \le \delta_1 < \delta_2$ and the above maximizers are chosen exactly, then $\|P_{\delta_2} r_0\|_{L_2(Z)}^2 \ge \|P_{\delta_1} r_0\|_{L_2(Z)}^2$, by adding the two optimality inequalities. This formalizes the intuition that increasing $\delta$ encourages the learned $Z$-features to contain $r_0$. In fact, there is an explicit lower bound. Let $S_d := \sum_{j=1}^d \lambda_j^2$, where $(\lambda_j)_j$ are the singular values of $\mathcal{T}$. Comparing $P_\delta$ with any rank-$d$ projection whose range contains $r_0$ gives

$$\mathrm{Tr}(P_\delta \mathcal{T}\mathcal{T}^\star) + \delta^2 \|P_\delta r_0\|_{L_2(Z)}^2 \ge \delta^2 \|r_0\|_{L_2(Z)}^2.$$

$\mathrm{Tr}(P_\delta \mathcal{T}\mathcal{T}^\star) \le \sup_{\text{projection } P, \mathrm{rank}(P) = d} \mathrm{Tr}(P\mathcal{T}\mathcal{T}^\star) = S_d$. Hence

$$\|P_\delta r_0\|_{L_2(Z)}^2 \ge \left( \|r_0\|_{L_2(Z)}^2 - \frac{S_d}{\delta^2} \right)_+,$$

where $t_+ := \max\{t, 0\}$. Thus, at the population optimum, the $Z$-side projection not only increases with $\delta$ but approaches $\|r_0\|_{L_2(Z)}^2$.

The same lower bound explains why the plug-in inverse can become unstable. Fix a $d$-dimensional $Z$-feature subspace with projection $\Pi_\Psi$, and consider

$$A_{\delta,\Psi} := \Pi_\Psi[\mathcal{T} \mid \delta r_0] : L_2(X) \times \mathbb{R} \to L_2(Z).$$

Assume that $A_{\delta,\Psi}$ has rank $d$ and write its right singular vectors as $(\varphi_i, \omega_i)$, $i = 1, \ldots, d$. With $e = (0, 1) \in L_2(X) \times \mathbb{R}$ and $\omega = (\omega_1, \ldots, \omega_d)^\top$,

$$\|\omega\|_{\ell_2}^2 = \|\Pi_{\mathrm{Span}(A_{\delta,\Psi}^\star)} e\|^2 = \|\Pi_{\mathrm{Ker}(A_{\delta,\Psi})^\perp} e\|^2.$$

Moreover,

$$\mathrm{Ker}(A_{\delta,\Psi}) = \{(f, a) : \Pi_\Psi \mathcal{T} f + a\delta \Pi_\Psi r_0 = 0\}.$$

Since $\Pi_\Psi r_0 = \Pi_\Psi \mathcal{T} h_0$, the minimum-norm $f$ satisfying this constraint for a fixed $a$ is $-a\delta (\Pi_\Psi \mathcal{T})^\dagger \Pi_\Psi r_0$. Hence, with

$$\eta_\Psi^2 := \|(\Pi_\Psi \mathcal{T})^\dagger \Pi_\Psi r_0\|_{L_2(X)}^2,$$

we obtain

$$\|\omega\|_{\ell_2}^2 = \min_{a \in \mathbb{R}} \left\{ a^2 \delta^2 \eta_\Psi^2 + (a-1)^2 \right\} = \frac{\delta^2 \eta_\Psi^2}{1 + \delta^2 \eta_\Psi^2}.$$

It remains only to note that $\eta_\Psi^2$ is itself bounded below by the $Z$-side projection length. If $\Pi_\Psi r_0 = 0$ this is immediate; otherwise,

$$\eta_\Psi = \min\{\|f\|_{L_2(X)} : \Pi_\Psi \mathcal{T} f = \Pi_\Psi r_0\} \ge \frac{\|\Pi_\Psi r_0\|_{L_2(Z)}}{\|\Pi_\Psi \mathcal{T}\|_{\mathrm{op}}} \ge \|\Pi_\Psi r_0\|_{L_2(Z)},$$

where the last inequality uses $\|\Pi_\Psi \mathcal{T}\|_{\mathrm{op}} \le \|\mathcal{T}\|_{\mathrm{op}} \le 1$. Therefore, for the population subspace $\Pi_\Psi = P_\delta$,

$$\delta^2 \eta_{P_\delta}^2 \ge \delta^2 \left( \|r_0\|_{L_2(Z)}^2 - \frac{S_d}{\delta^2} \right)_+ \longrightarrow \infty$$

whenever $r_0 \ne 0$. Consequently $\|\omega\|_{\ell_2}^2 \to 1$ along the population optima. For empirical learned features the same conclusion is approximate when their $Z$-span is close to $P_\delta$. Thus increasing $\delta$ encourages $Z$-side alignment with $r_0$, but the resulting inverse in $\widehat{A}(\delta)$ becomes fragile when that alignment pushes $\|\omega\|_{\ell_2}$ close to one.

In practice we therefore use $\widehat{A}(\delta)$ conservatively. We scan an increasing grid of candidate values and accept a larger $\delta$ only when it produces a substantial relative increase in the estimated alignment. In Figures 14 and 15, the threshold is $5\%$: a larger value of $\delta$ is accepted only if $\widehat{A}(\delta)$ improves by at least $5\%$ relative to the previous accepted value. This rule selects models that outperform the non-outcome-aware SpecIV baseline, and in the dSprites experiments also outperform DFIV.

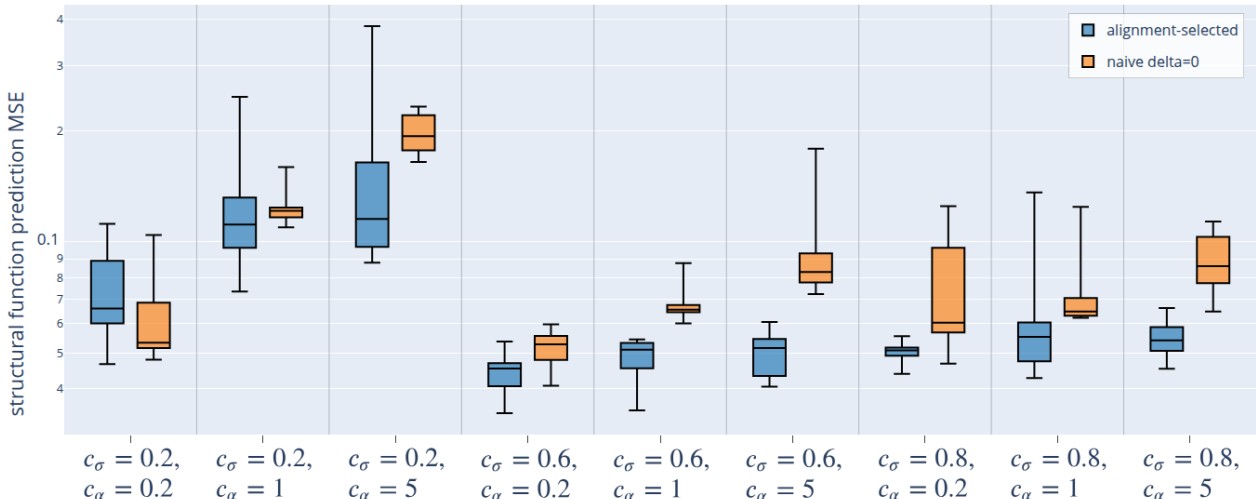

*Figure 14.* $h_0$ prediction MSE on synthetic one-dimensional experiments when $\delta$ is selected by estimated spectral alignment, compared to the non-outcome-aware $\delta = 0$ baseline.

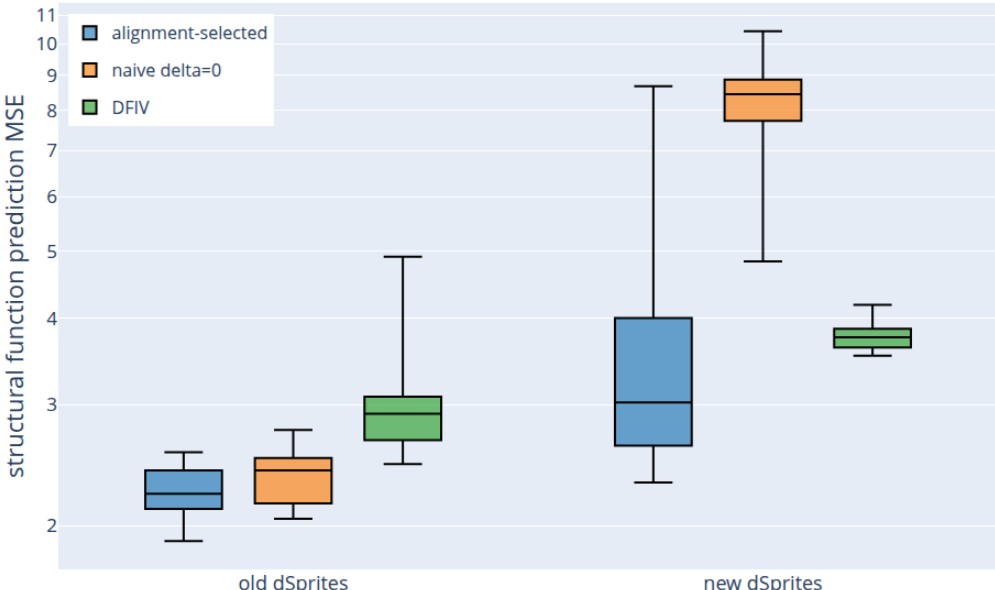

*Figure 15.* Structural-function prediction MSE on the dSprites benchmarks when $\delta$ is selected by estimated spectral alignment, compared to the non-outcome-aware $\delta = 0$ baseline and DFIV.

## F. Higher rank perturbations

The rank one augmentation we propose extends directly to higher rank perturbations. This setting includes vector–valued outcomes and also the case where one wishes to encourage the learned $Z$ features to retain predictive information about multiple functions of $Y$.

Let $f_1, \ldots, f_K$ be functions of $Y$ that we would like to be well approximated from $Z$. We define $r_k = \mathbb{E}[f_k(Y) \mid Z]$, $k = 1, \ldots, K$. For $\underline{\delta} \in \mathbb{R}^K$ we define:

$$\mathcal{L}_{\underline{\delta}}^{(d)}(\theta) \doteq \mathcal{L}_0^{(d)}(\theta) - \sum_{k=1}^{K} \delta_k^2 \left\| \Pi_{\psi_\theta^{(d)}} r_k(Z) \right\|^2.$$

As in the rank one case, to avoid differentiating through matrix inverses, this can be rewritten with auxiliary variables $\omega_k \in \mathbb{R}^d$ as:

$$\mathcal{L}_{\underline{\delta}}^{(d)}(\theta, \omega_1, \ldots, \omega_K) \doteq \mathcal{L}_0^{(d)}(\theta) + \sum_{k=1}^{K} \left( \omega_k^\top C_{\psi_\theta^{(d)}} \omega_k - 2 \underline{\delta}_k \mathbb{E}[f_k(Y)\psi_\theta^{(d)}(Z)]^\top \omega_k \right).$$

The optimal features correspond to the truncated SVD of the following rank $K$ perturbed operator:

$$\mathcal{T}_{\underline{\delta}} : L_2(X) \times \mathbb{R}^K \to L_2(Z), \qquad (h, a) \mapsto \mathcal{T}h + \sum_{k=1}^{K} a_k \, \underline{\delta}_k \, r_k.$$

A complete theoretical analysis of the rank $K$ setting is beyond the present scope and we regard it as an important direction for future work. We report preliminary results for $K = 2$ using $f_k(y) = y^k$ in Figure 16. In the dSprites setting with $h_0 = h_{\text{new}}$, including $f_2$ alone yields similar improvements as $f_1$ alone. Using both functions gives slightly lower error, but the differences are small and do not allow strong conclusions at this stage.

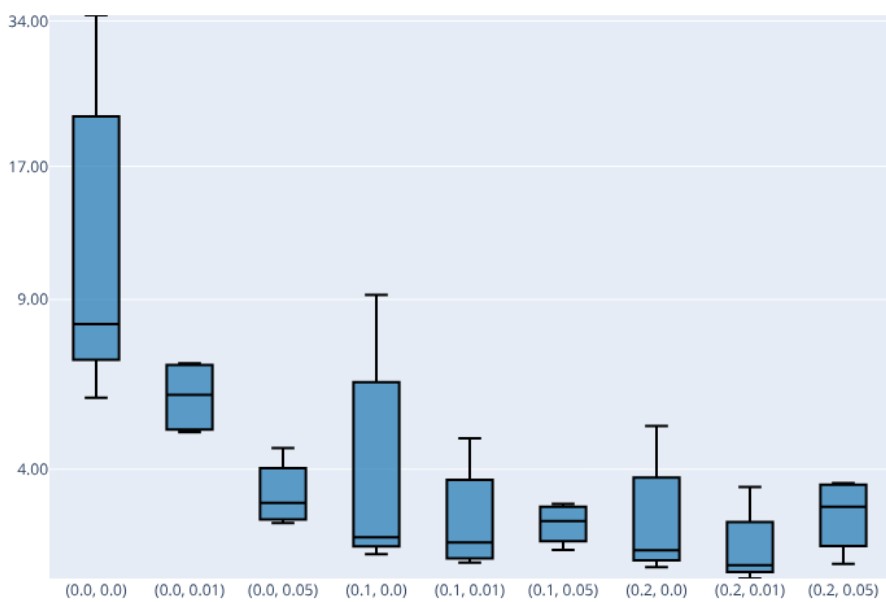

Figure 16. IV MSE for rank two perturbations with $f_k(y) = y^k$, $k = 1, 2$, on dSprites with $h_0 = h_{\text{new}}$. For example, $\delta = (0.2, 0.0)$ reduces to the rank one case.

## G. Naive outcome-aware baselines

One can conceive of far simpler outcome-aware IV methods than ours. For instance, a naive idea is to train the features in two stages.

1. Train $Z$-features $\psi_\theta^{(d)}$ by maximizing $\|\Pi_{\psi_\theta^{(d)}} r_0\|_{L_2(Z)}^2$, perhaps with an additional regularization term preventing features from blowing up or becoming collinear.

2. Keeping the $Z$-features fixed, train $\varphi_\theta^{(d)}$ by minimizing a loss that measures how well $\varphi_\theta^{(d)}$ predicts $\psi_\theta^{(d)}$. Natural candidates include $\mathcal{L}_0(\theta)$ or simply $\mathbb{E}[\|\psi_\theta(Z) - \varphi_\theta(X)\|^2]$.

This approach is not guaranteed to yield consistent results even in population. For instance, when learning $d = 1$ features, if the first stage learns $\psi(Z) \propto r_0$, then the minimizer of $\mathbb{E}[\|\varphi(X) - \psi(Z)\|^2]$ is $\varphi(X) \propto \mathcal{T}^* r_0$. Unless $r_0$ happens to be a singular function of $\mathcal{T}$, this means that the method is not learning the correct features. Indeed, the nature of the first stage of the representation learning objective makes it impossible to discern the properties of the learned $Z$ features whenever $d > 1$, other than asserting that jointly they should span $r_0$. This is precisely why our method balances the projection length of $r_0$ onto the $Z$ features, with capturing the geometry of $\mathcal{T}$. Nonetheless, we report on the performance of such baselines, in part because the one based on minimizing $\mathbb{E}[\|\psi_\theta^{(d)} - \varphi_\theta^{(d)}\|^2]$ in the second stage sometimes performed surprisingly well. We found minimizing $\mathcal{L}_0(\theta)$ for a fixed $\psi_\theta$ in the second step to yield very poor results, and we do not include them below.

| Layer | Configuration |
|-------|---------------|
| 1 | Input: 1 |
| 2 | FC(1,50), $x \mapsto x + \sin^2 x$ |
| 3 | FC(50, 50), GeLU |
| 4 | FC(50, 5) |

*Table 6.* $Z, X$ feature networks for the outcome-aware baseline.

**Practical performance.** We evaluate the "naive outcome-aware baseline" on a range of fully synthetic one-dimensional experiments identical to the ones in Section 6, all with the additive noise variance at $\sigma^2 = 0.2$, 15000 training samples and 5000 evaluation samples. The models used are specified in Table 6. We compare the attained IV strong-norm losses to those of AugSpecIV. Further, in contrast to our experiments reported in the main body of the paper, we report on a fully-adaptive $\delta$-tuning rule. For each $\delta \in \{0, 0.5, 1.0, 3.0, 5.0\}$, we fit a model by minimizing $\mathcal{L}_\delta(\theta)$ jointly over the $X, Z$ features. We choose to evaluate the model with the largest $\delta$ such that the estimate of $\|\Pi_{\varphi^{(d)}} h_0\|^2$ based on Proposition 6.1 increases by at least $5\%$ relative to the previous, smaller $\delta$. In Figure 17 we report the median losses attained by so-trained models, and the newly proposed baseline, across a range of values of $c_\alpha$ and $c_\sigma$. The ridge regularization parameters used in both stages of 2SLS were chosen by the minimization of 2SLS weak-error estimate. Note that $\mathrm{MSE} = 0.5$ corresponds to the loss attained by a constant function estimator of $h_0$.

Since the newly proposed baseline's performance was very unstable, we introduced an additional post-hoc method for rejecting poorly-fitted models. Whenever the model generated forecasts for $h_0$ whose standard deviation was over 5 times larger than that of the observed $Y$, we deemed the predictions implausible and rejected them. Figures 18 and 19 report the fraction of fitted models passing this rejection rule. Across our testing parameter grid, this happened in 9 out of 1200 AugSpecIV models, but in around half of the instances of the new baseline. This is, in spirit, similar to the approach of Bruns-Smith (2025), who also proposes to learn outcome-aware representations based on a regression-only objective, but then performs a hypothesis test to verify whether the learned representations can be used for approximating the conditional-mean operator. Indeed, when the learned method produces "sensible-looking" predictions, it is competitive with ours, attaining comparable median losses, though with a noticeably heavier tail of large MSEs.

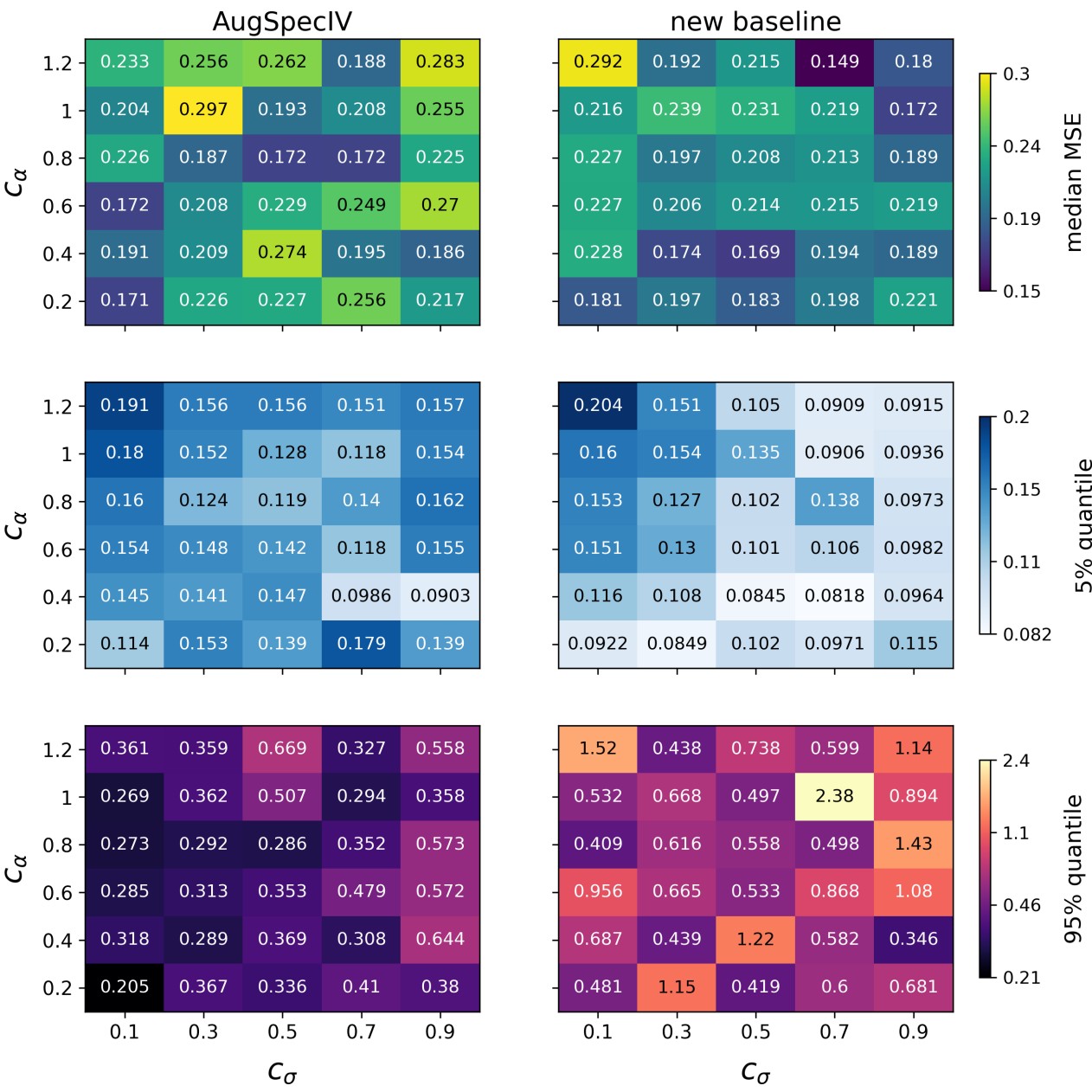

*Figure 17.* Comparison of AugSpecIV and the naive outcome-aware baseline on synthetic one-dimensional experiments across values of $c_\alpha$ and $c_\sigma$. The top row reports median IV MSE. The middle and bottom rows report the $5\%$ and $95\%$ quantiles, respectively.

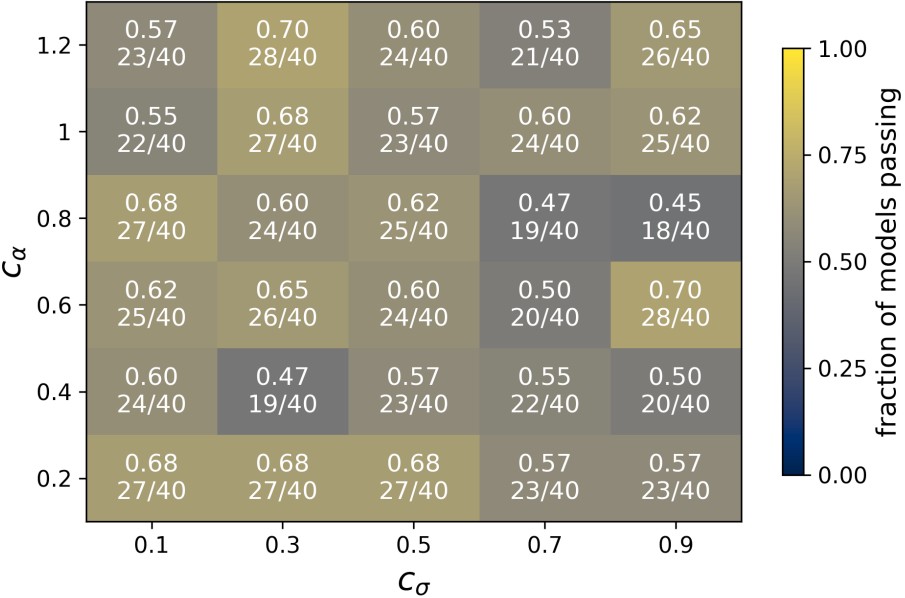

*Figure 18.* Fraction of naive outcome-aware baseline models passing the post-hoc rejection rule across values of $c_\alpha$ and $c_\sigma$. Each cell also reports the corresponding number of passing models.

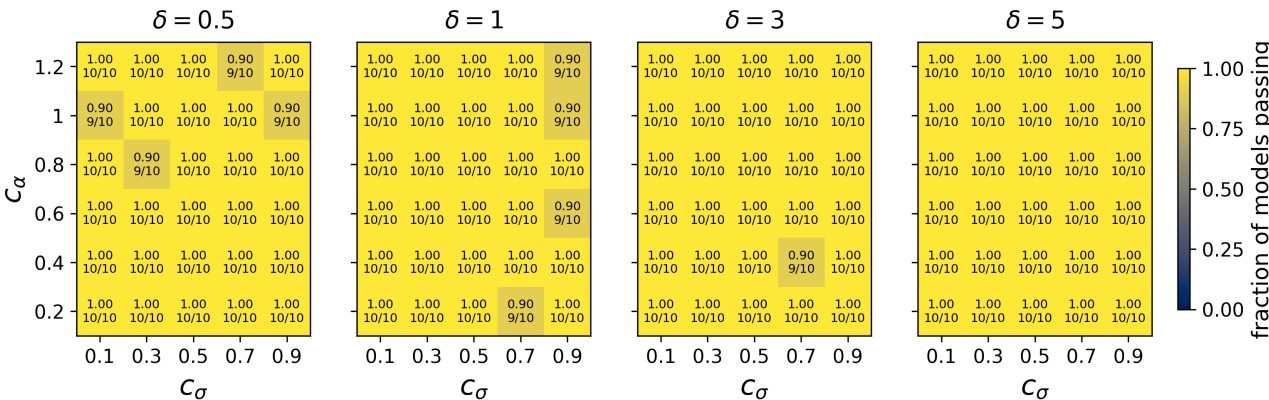

*Figure 19.* Fraction of AugSpecIV models passing the post-hoc rejection rule for each value of $\delta$, across values of $c_\alpha$ and $c_\sigma$. Each cell also reports the corresponding number of passing models.

