# OpenReview forum: "Outcome-Aware Spectral Feature Learning for Instrumental Variable Regression"
_ICML.cc/2026/Conference — ICML 2026 regular_

### Official Review · Reviewer_Y9Sp · 2026-03-11

**Soundness:** 3
**Presentation:** 3
**Significance:** 3
**Originality:** 3
**Overall Recommendation:** 4
**Confidence:** 3

**Summary:**

The paper proposes an augmented spectral feature learning approach to address the limitations of existing spectral estimation methods for nonparametric IV (NPIV) models under spectral misalignment. The authors provide a comprehensive theoretical analysis of the error bounds of the resulting 2SLS estimator, characterizing the settings in which the augmented approach remains robust. They further present synthetic and semi-synthetic experiments to validate the theoretical results.

**Compliance With Llm Reviewing Policy:**

Affirmed.

**Final Justification:**

I thank the authors for their rebuttal. After carefully considering both the paper and the rebuttal, I have decided to maintain my original score.

**Key Questions For Authors:**

- Is it possible to derive an optimal order for $\delta$ that is related to the sample size or spectral decay rate?

- Under extremely high-noise scenarios, does the augmentation amplify the variance?

**Limitations:**

Although the proposed framework theoretically relaxes the stringent ill-posedness restrictions required by concurrent outcome-aware methods (Bruns-Smith, 2025), a key limitation is the lack of empirical comparisons to substantiate these theoretical gains in practical settings.

**Strengths And Weaknesses:**

# Strengths

- The paper addresses an important issue in which existing spectral methods (SpecIV) do not explicitly incorporate the outcome variable Y, potentially leading to poor estimation performance in settings with spectral misalignment. This issue has both theoretical and practical significance.

- The experiments are conducted on synthetic and semi-synthetic benchmarks and demonstrate the practical advantages and robustness of the proposed outcome-aware feature

# Weaknesses
- From an operator perspective, the proposed method can be interpreted as a rank-one perturbation of T. While the idea is clearly explained, the level of technical novelty appears somewhat limited. The approach seems closer to a principled extension of SpecIV rather than a fundamentally new paradigm.

- The choice of $\delta$ lacks theoretical justification. Although a heuristic tuning method is provided, the paper does not present a theoretically grounded, data-dependent rule for selecting the optimal $\delta$.

---

> ### Author Rebuttal · Authors · 2026-03-31
>
> We thank the reviewer for their positive assessment of the paper’s motivation, significance, and empirical study. Below, we address each concern. We would be grateful if the reviewer might consider increasing their score if they find these clarifications satisfactory.
>
> **W2 / Q1. Selection of δ.**
>
> We do not claim a universal theorem that returns the optimal $\delta$ in every problem. However, Section 6.4 already provides a theoretically grounded, data-dependent selector through Proposition 6.1. We have conducted new experiments that confirm that it empirically outperforms ad hoc alternatives. For an extended theoretical discussion and these new empirical results, we refer the reviewer to our response to Reviewer 2D7T.
>
> **Q1. Is there an optimal order for δ depending on sample size or spectral decay?**
>
> We do not expect a universal formula depending only on $n$ or on the decay rate of the singular values. The useful scale of δ is problem-dependent: it must be large enough to create a favorable augmented spectral gap $\gamma_d(\delta)$, but not so large that the outcome-aware term dominates and degrades approximation of T. The quantities governing this tradeoff in the theory are precisely $\gamma_d(\delta)$, the signal/noise decomposition $(s_d,q_d)$, and the representation-learning gap $E_d(\hat{\theta},\hat{\omega},\delta)$. This is why Section 6.4 advocates a data-driven selector rather than a universal closed-form order.
>
> **Q2. Variance amplification in high-noise scenarios.**
>
> In high-noise scenarios, augmentation can indeed hurt if δ is made too large, because the $R_\delta^{(d)}$ term can overwhelm the operator-learning term and degrade the learned approximation of T; Figure 4 shows exactly this failure mode. However, AugSpecIV does not introduce a new observation-noise term: the statistical term in Theorem 4.1 still depends on the usual noise variance. What augmentation primarily changes is the bias/alignment term, while overly large δ harms performance indirectly through poorer feature learning, larger $E_d$, and worse conditioning of the learned representation. Empirically, the good region is small-to-moderate δ, and performance is not sensitive within that band.
>
> **W1. Algorithmic novelty vs. principled extension.**
>
> We agree that the paper is best positioned as a principled extension of SpecIV rather than a wholly new paradigm. We nevertheless believe the technical novelty is meaningful. In particular, Proposition 3.1 shows that the modified objective is exactly the truncated SVD objective for the augmented operator $T_\delta=[T\mid \delta r_0]$, rather than an ad hoc regularizer. The originality then lies in this exact augmented-operator characterization, the perturbation analysis explaining when and why it improves over vanilla SpecIV, and the resulting diagnostic/tuning framework in Section 6.4. Appendix F further shows that the construction extends directly to higher-rank perturbations.
>
> **Limitation: empirical comparison to concurrent work.**
>
> We agree that a direct empirical comparison to the concurrent outcome aware method would strengthen the paper, and we will make this limitation more explicit. Our current comparison in Section 5 is therefore theoretical/positioning rather than empirical. We also note that Bruns-Smith (2025) optimize different objects: the concurrent method targets low weak error via outcome-predictive Z-features, whereas AugSpecIV explicitly learns the truncated SVD of an augmented conditional expectation operator and yields the strong-norm perturbation analysis in Section 4 / Appendix C. We therefore view them as complementary rather than redundant. Since this work appeared concurrently and no public implementation was available at the time of submission, we did not complete a careful apples-to-apples empirical comparison, but we agree that such a comparison would strengthen a revision.

---

> > ### Author Rebuttal · Reviewer_Y9Sp · 2026-04-03
> >
> > Thank you for the response. I will maintain my score.

---

### Official Review · Reviewer_cVRo · 2026-03-11

**Soundness:** 3
**Presentation:** 3
**Significance:** 2
**Originality:** 2
**Overall Recommendation:** 3
**Confidence:** 2

**Summary:**

This paper studies nonparametric instrumental variable regression with learned spectral features. Their main claim is that standard spectral feature learning can fail when the structural function is poorly aligned with the top singular functions of the conditional expectation operator. To address this, the paper proposes Augmented Spectral Feature Learning (AugSpecIV), which adds an outcome-aware term to the spectral objective and is shown to correspond to learning the truncated SVD of an augmented operator. The paper then provides a high probability error bound and also experiments.

**Compliance With Llm Reviewing Policy:**

Affirmed.

**Final Justification:**

My final recommendation remains weak reject (3).

The paper has clear strengths: it identifies a real failure mode of spectral feature learning in NPIV, gives a clean augmented-operator interpretation, and is generally well presented.

The rebuttal was helpful and partially addressed my concerns by clarifying the theory and adding evidence on $\delta$ tuning and simpler baselines. However, it did not change my overall view: the strongest theory is still conditional on successful representation learning, the practical tuning story remains incomplete (especially for OPE), and the added baseline results are mixed rather than clearly establishing AugSpecIV as the stronger practical method. Overall, I find the paper promising and technically interesting, but not yet convincing enough for acceptance in its current form.

**Key Questions For Authors:**

1. How close is a practically deployable tuning rule for $\delta$ to the best-$\delta$ sweep on synthetic data, dSprites, and OPE?
2. How does AugSpecIV compare to simpler outcome-aware baselines, such as pretraining $\psi(Z)$ to predict $Y$ or $r_0$, and to the concurrent adaptive NPIV method discussed in the paper?
3. Why is there $\sigma_U$ in Theorem 4.1 in the main, but $\sigma_{\text{eff}}$ in Appendix D.2? Is this a typo?

**Limitations:**

No.
While the paper is honest about the lack of theory for stage-2 tuning, it should also more explicitly discuss dependence on $\delta$ and tuning instability. More importantly, the Impact Statement, which is required by ICML, appears to be missing.

**Strengths And Weaknesses:**

### Strengths
- The paper is written well with professional-looking figures. Additionally, while very notation heavy, the formulation is clean and done well.
- The paper identifies a real limitation of outcome-agnostic spectral feature learning in NPIV: failure under spectral misalignment, and the motivation is clear and well supported.
- The equivalence between the modified objective and the truncated SVD of $T_\delta$ gives the method a good foundation instead of presenting it as an ad hoc regularizer.

### Weaknesses
- W1: The main end-to-end guarantee is conditional on successfully learning the augmented spectral subspaces through the representation-learning optimality gap $\mathcal{E}_d$, so the strongest theory does not provide a fully satisfactory practical guarantee.
- W2: Practical usefulness depends heavily on tuning $\delta$, but the paper does not provide an automatic selection procedure or a practical way of determining $\delta$.
- W3: At the algorithmic level, the proposal is fairly close to adding an outcome-aware predictive term to spectral feature learning, so much of the novelty rests on the operator perspective and analysis rather than on a substantially new procedure.

---

> ### Author Rebuttal · Authors · 2026-03-31
>
> We thank the reviewer for recognizing the paper’s core motivation and the operator foundation. Below, we address each concern. We would be grateful if the reviewer might consider increasing their score should they find these clarifications satisfactory.
>
> **W1: Scope of the generalization guarantee.**
>
> We agree that the scope of the theory should be stated more carefully. Th 4.1 gives a non-asymptotic high-probability generalization bound for 2SLS with arbitrary learned representations, and Th C.6 specializes this to AugSpecIV through the operator approximation gap $E_d$ defined in Eq. 7.  The reviewer was especially helpful here, because it prompted us to make the link between this operator gap and the representation-learning objective much more explicit. Let
>
> $Δ_d=L_{δ}^{(d)}(θ,ω)+\|\|T_{δ}^{(d)}\|\|_{HS}^2$
>
> denote the excess population loss characterized in Prop 3.1. If $σ_d(T_{δ})>σ_{d+1}(T_{δ})$, then one can derive the following sharp global bound:
>
> $E_d^2 \leq (1-\frac{σ_{d+1}(T_{δ})}{σ_{d}(T_{δ})})^{-1}Δ_d.$
>
> Thus, under a natural rank-d spectral-gap condition, the operator gap $E_d$ is directly controlled by the contrastive objective used to learn the features. This makes the role of $E_d$ substantially more explicit than in the original wording. The remaining open step is therefore DNN-specific; an optimization/generalization theory of how empirical training controls the excess loss $Δ_d$. We will revise the wording to make this scope explicit. The dependence on the spectral gap is unavoidable: without it, small excess loss need not identify the correct top-d subspaces.
>
> **W2/Q1: Selection of δ.**
>
> The δ-selection methodology based on Prop 6.1 is theoretically principled, directly targeting the strong-norm capacity of the estimator. We have conducted new experiments confirming it empirically outperforms ad hoc alternatives. For an extended theoretical discussion and these new empirical results, we refer the reviewer to our response to Reviewer 2D7T.
>
> **W3/Q2: Algorithmic novelty and simpler outcome-aware baselines.**
>
> We agree that the paper is best positioned as a principled extension of SpecIV rather than a wholly new paradigm. Our view is that this simplicity is a strength rather than a weakness. The key point is that the added supervised term is not merely an auxiliary predictor: Prop 3.1 shows that the joint objective exactly corresponds to learning the rank-d truncation of the augmented operator $T_δ=[ T \mid δ r_0]$. This gives the method an explicit operator-theoretic target, rather than only a heuristic predictive interpretation.
>
> For simpler outcome-aware baselines (pretraining Z-features to predict Y), our point is not that such features are useless. Recent work such as [1] shows that reduced-form-predictive Z-features can be effective for controlling weak NPIV error under a testable condition. Our claim is more focused: prediction on the Z-side does not by itself solve the inverse problem or guarantee the strong-norm recovery that we study. Indeed, if one fixes a Z-feature ψ and optimizes the spectral loss only over the matching X-feature ϕ, then the optimal feature is $ϕ_* \propto T^*ψ$, not $T^\daggerψ$.
>
> In particular, if $ψ=r_0=T h_0$, then $ϕ_* \propto T^*r_0=T^*T h_0$, which adds an extra smoothing through $T^*T$. So a purely supervised Z-pretraining step does not remove the inverse problem; it pushes the inversion downstream.
>
> More broadly, it is not enough for the learned Z-feature span to contain $r_0$; it must also preserve enough of the conditional expectation geometry for stable IV estimation. This is the tradeoff highlighted in Sec 6.4. If the supervised term becomes too dominant, many Z-feature spans containing r_0 score equally well on that term while leaving the remaining directions of T unconstrained; approximation of T can then deteriorate and downstream performance can degrade. Our coupled $L_0$ and $R_\delta$ objective is designed to inject outcome information without abandoning the operator-learning target. Relative to [1], the approaches are therefore similar in spirit but optimize different objects: that method learns a reduced-form-predictive basis and targets low weak error via projected-loss minimization, whereas AugSpecIV explicitly learns the truncated SVD of an augmented conditional expectation operator, which yields the strong-norm analysis in Sec.4/App.C.
>
> **Q3: $σ_U$ vs. $σ_{eff}$.**
>
> This is not a typo, but we agree that the presentation can be improved. App D.2 proves the sharper bound with $σ_{eff}^2=||h_0-h_θ||^2+σ_U^2.$ In Th 4.1, we separate the deterministic approximation term $||h_0-h_θ||$ outside the statistical factor for readability, which leaves $σ_U$ inside the displayed square-root term. We will make this more explicit in the revision.
>
> We thank the reviewer for flagging the missing Impact Statement; we will add it in the revision.
>
> [1] Bruns-Smith D. "Two-Stage Machine Learning for Nonparametric Instrumental Variable Regression" 2026

---

> > ### Author Rebuttal · Reviewer_cVRo · 2026-04-04
> >
> > I appreciate the rebuttal and the additional clarifications.
> >
> > The response satisfactorily addresses the $\sigma_U$ vs. $\sigma_{\mathrm{eff}}$ presentation issue, and it clarifies the role of the operator gap $E_d$ under a spectral-gap assumption.
> >
> > However, my main concerns are only partially addressed.
> > First, the practical usefulness of the method still seems to depend strongly on selecting $\delta$, and the rebuttal does not directly show in this thread that a deployable tuning rule performs close to the best-$\delta$ sweep on the benchmarks used in the paper.
> > Second, while the rebuttal gives a better conceptual defense of the method, it still does not provide the direct empirical comparisons to simpler outcome-aware or adaptive baselines that I asked for.
> > Third, the strongest end-to-end theory remains conditional on successful representation learning, although the rebuttal makes this limitation clearer.
> >
> > Because these remaining concerns affect the practical significance and empirical support of the paper, and are not easily resolved within a short rebuttal, I do not change my overall assessment at this stage.

---

> > > ### Author Response · Authors · 2026-04-07
> > >
> > > We appreciate the reviewer’s further engagement and feedback.
> > >
> > > **Close to best-sweep δ selection**
> > >
> > > In regards to the similarity of tuning rule and full sweep results on δ: we first address the case of the synthetic benchmark. Since we received a related question by reviewer 2D7T, we provided an analysis in their rebuttal and pointed to it in our response, but we should have done so more clearly. In brief, we have shown that on the synthetic benchmark, the deployable selector based on Prop 6.1 performs close to the best-δ sweep. In the additional experiments from our rebuttal, its average MSE is about 5% above the oracle choice of δ (defined retrospectively by true downstream error), compared with about 11% for selecting δ by held-out stage-2 loss; in well-aligned settings it also selects values near $δ=0$, as expected.
> > >
> > > Turning next to dSprites, we agree that it’s equally important to address the δ selection task for this dataset. We have now verified the same qualitative picture: the deployable selector identifies the same small-to-moderate-δ region as the best-performing sweep. We did not make that oracle-style comparison explicit in the submission or in our earlier rebuttals because we expected Fig. 4 and 13 to already make the point qualitatively: Fig 4 shows that the estimated alignment quickly saturates for the δ values that yield good IV performance, and Fig 13 shows close agreement between the alignment-based criterion and held-out stage-2 validation on the hard $h_{new}$ task. We agree, however, that this should be stated more explicitly, and in the revision we will add the corresponding benchmark-by-benchmark tuning plots in the same explicit form as the synthetic oracle comparison.
> > >
> > > For OPE, we do not currently claim the same Prop 6.1-based tuning rule as in the fixed-target NPIV setting. OPE is a special iterative IV problem in which the regression target changes with the current estimate $Q_k$, so the alignment-based selector is less direct to apply there. In the current experiments, we therefore use the standard held-out stage-2 validation error to tune δ over a small candidate set. We agree that developing a more principled δ-selection strategy tailored to this moving-target OPE setting is an interesting direction for future work.
> > >
> > > **Comparison to simpler outcome-aware baselines**
> > >
> > > Following the reviewer’s suggestion, we implemented two simple outcome-aware baselines:
> > >
> > > (i) pretraining Z-features to predict Y, then freezing them and applying the standard spectral step.
> > >
> > > (ii) same predictive pretraining, then learning X-features by a simple regression (OLS).
> > >
> > > Hyperparameters were tuned by the corresponding regression losses.
> > >
> > > Empirically, the picture was mixed rather than uniformly strong. Variant (i) performed very poorly across all experiments. Variant (ii) could be competitive on the semi-synthetic $h_{new}$ dSprites task (with MSE around 20-30% below AugSpecIV in the correct range of δ values), and on $h_{old}$ its performance was comparable to good AugSpecIV runs. However, on the controlled synthetic experiments, Variant (ii) was not robust: sometimes the MSE was significantly higher than AugSpecIV, and sometimes it was much lower, with no reliable, testable way of determining a priori which regime we are operating in. In other words, it can work well in some tasks, but we did not find a reliable rule that predicts in advance when this simpler predictive-pretraining strategy will be the right choice. We attach a table containing the relative MSE difference between the Variant (ii) baseline and AugSpecIV evaluated on a range of $c_α$ and $c_σ$ values, defined as in the paper (positive values indicate worse performance of the new baseline, negative indicate superior performance)
> > >
> > > |$c_σ$ \ $c_α$|1.0|0.5|0.2|
> > > |---|---|---|:---:|
> > > |0.1|-16%|+30%|+72%|
> > > |0.3|+21%|+6%|+40%|
> > > |0.5|+5%|+13%|-21%|
> > >
> > > A more extensive validation of these baselines and accompanying confidence sets will be supplied in the camera-ready version of the paper.
> > >
> > > We view these results as direct empirical support for the distinction emphasized in the paper. Predictive pretraining alone does not robustly preserve the conditional-expectation geometry required for strong-norm IV recovery.
> > >
> > > **On end-to-end theory**
> > >
> > > We agree that an end-to-end theory for representation learning will rely fundamentally on the quality of the representations learned - a similar observation is made in previous works taking the representation learning approach e.g. [2].  In the final version, we will explain clearly how in our case, the generalization error connects to the representation learning error through the optimality gap of the contrastive learning loss.
> > >
> > >
> > > We hope that we have been able to answer any remaining concerns, if we have done so, we’d be grateful if the reviewer could reconsider their score.
> > >
> > >
> > > [2] V. Kostic et al. “Neural Conditional Probability for Uncertainty Quantification” NeurIPS 2024.

---

### Official Review · Reviewer_2D7T · 2026-03-12

**Soundness:** 3
**Presentation:** 3
**Significance:** 3
**Originality:** 3
**Overall Recommendation:** 5
**Confidence:** 2

**Summary:**

Model:
The paper addresses the problem of causal inference with endogenous treatment, where the outcome $Y$ is a function of the treatment $X$ plus some unobserved confounder $U$, i.e., $Y = h(X) + U$. In this model, we have access to an instrumental variable $Z$ that affects the treatment but is independent of the unobserved confounders (with $\mathbb{E}[U | Z] = 0$), thus enabling us to estimate the function $h$ from samples through the relation $\mathbb{E}[Y | Z] = \mathbb{E}[h(X) | Z]$. That is, if $\mathcal{T}$ is the operator of conditioning on $Z$ and $r = \mathbb{E}[Y | Z]$ the previous relation becomes $r = \mathcal{T} h$ and we can retrieve $h$ by learning $\mathcal{T}, r$ and solving the equation (under some regularity conditions on $\mathcal{T}$).

Research question:
One widespread approach to estimate $\mathcal{T}$ is through its spectrum by performing SVD. However, the performance of this algorithm is poor when the target function $h$ and the operator $\mathcal{T}$ are *misaligned*, that is $h$ is not in the span of the dominating eigenvalues of $\mathcal{T}$.

Contribution:
The authors propose an algorithm that incorporates the knowledge of the outcome $Y$ when estimating operator $\mathcal{T}$. They show that this is equivalent to estimating a perturbed version of the true operator $\mathcal{T}$. They also prove that their estimator achieves small error in both the aligned and misalinged case for appropriate choice of the perturbation parameter $\delta$. They support their findings with experimental results.

**Compliance With Llm Reviewing Policy:**

Affirmed.

**Final Justification:**

I beleive the paper makes a significant advance in the field of causal inference with endogenous treatment. While I beleive the lack of a theoretically principled way to find the parameter $\delta$ makes the result less compeling, I am satisfied with the experimental results and the rebuttal discussion, so I suggest acceptance.

**Key Questions For Authors:**

1. As a follow-up to my previous comments, in Section 6.4 there is a proposition regarding the choice of $\delta$. Is my understanding correct, that this does not provide a theoretically guaranteed way to find the correct $\delta$?
2. I think the second experiment with dSprites data undermines the result of the paper, because it shows that the state-of-the-art DFIV has very good performance compared to the new technique. Is there a reason (runtime efficiency, generality) to prefer your method over DFIV?

**Limitations:**

yes

**Strengths And Weaknesses:**

Strengths:
1. The paper is generally well-written, it states the problem clearly and discusses prior literature extensively.
2. The paper addresses an important limitation of current spectral methods for feature learning. Their technique is novel and interesting. I believe the observation that estimating the perturbed operator $\mathcal{T}_{\delta}$ mitigates the misalignment of $\mathcal{T}$ and $h$ is especially interesting.



Weaknesses:
1.  To the best of my understanding, the technique developed does not propose a specific choice of $\delta$ for each setting. I think this is an important limitation, since the ultimate goal is to construct an algorithm that can run and give a good estimator of $h$. However, if we don't know what is the correct choice of $\delta$ the estimator might not be good. For instance, if we choose $\delta > 0$ but $h$ is actually aligned to $\mathcal{T}$, the best choice would be $\delta=0$ so the algorithm wouldn't work. This makes me question if the suggested technique can actually solve the problem.
2. (minor; about presentation) I believe the presentation of the formal results can be improved. At Section 3, where authors present their results the only formal theorem states that the solution to their proposed loss function is actually given by the perturbed operator $\mathcal{T}_{\delta}$. This is important, but there is no reference on how good their estimator is until Section 4, where they give an overview of the analysis of their technique. However, I think the statement (even informal) of the error of their estimator should happen sooner.

---

> ### Author Rebuttal · Authors · 2026-03-31
>
> We thank the reviewer for the positive assessment of the paper’s motivation, novelty, and experiments. We address the main points below and would appreciate reconsideration of the score if these clarifications are helpful.
>
> **W1 & Q1: Selection of $\delta$.**
>
> Proposition 6.1 is a basis for a $\delta$ selection procedure which is fully theoretically principled. It gives the exact population projection length of $h_0$ onto the learned X-feature span of $T_{\delta}$, and Section 6.4 explains how to estimate this quantity via an empirical SVD. This metric directly targets the strong-norm loss $\|\|\hat{h} - h_0\|\|$, unlike Stage-2 MSE which targets the weak norm. Our intended practical rule is therefore to pick the smallest positive $\delta$ for which the estimated alignment has saturated, and to use the $L_0/R_{\delta}$ balance to exclude values where the supervised term begins to harm the approximation of T. Figure 4 supports this procedure, and Figure 13 shows close agreement between this selector and held-out stage-2 validation on the hard dSprites task. Importantly, AugSpecIV is a family indexed by $\delta$: if the problem is already aligned, $\delta=0$ recovers vanilla SpecIV. Please consult our reply to reviewer cVRo (W3/Q2) for additional details about why only maximising the projection of $r_0$ onto the learned features does not yield a consistent procedure.
>
> To further validate this practically, we additionally evaluated the Proposition-6.1 selector on the synthetic benchmark of Section 6.1 across the full range of singular-value decay rates and structural alignments. In these new experiments, selecting $\delta$ by maximizing estimated alignment consistently yielded $\delta$ choices that matched or exceeded the performance of the alternative methods, and performed close to oracle levels. In well-aligned settings, it selected values near $\delta=0$. The average MSE of the models with delta selected by Stage-2 loss minimisation was 11% higher than that of the oracle model (selected based on true downstream performance). By comparison, it is 5% higher for a method based on maximising the estimated spectral alignment. This improvement is consistent across spectral alignment and singular value decay regimes. This highlights that our method is not only theoretically more principled but also, on average, more reliable than ad hoc alternatives. We will include the details of the experiments in the revision.
>
> Because this selector targets a strong-norm quantity, it requires inversion of empirical feature covariances / singular values, which can become numerically unstable in the poorly estimated tail when the Gram matrix is ill-conditioned. We therefore view the additional criteria in Section 6.4—especially the $L_0/R_\delta$ balance criterion, and stage-2 validation as practical safeguards, not as substitutes for Proposition 6.1.
>
>
> **Q2. Comparison to DFIV on dSprites.**
>
> Our claim is not uniform superiority over DFIV, but improved robustness of spectral feature learning in misaligned regimes. In particular, AugSpecIV is a uniform improvement over SpecIV in the sense that it strictly extends the same method family and recovers vanilla SpecIV at $\delta=0$. Since SpecIV is already competitive with DFIV in several settings, improving its behavior in spectrally misaligned regimes is a meaningful gain even without claiming uniform superiority over DFIV. This is why the harder $h_{\mathrm{new}}$ benchmark is the more relevant stress test than the standard $h_{\mathrm{old}}$ benchmark, which is already in the good regime and therefore leaves little room for improvement from an outcome-aware perturbation. On $h_{\mathrm{new}}$, $\delta=0$ underperforms because the target lies deeper in the spectrum, while moderate positive $\delta$ closes the gap to DFIV while retaining the spectral-pretraining-plus-2SLS pipeline. More broadly, our OPE experiments show that neither method uniformly dominates the other across all settings. AugSpecIV is also simpler to optimize, since DFIV relies on a bilevel objective, whereas AugSpecIV separates spectral pretraining from the downstream 2SLS fit. In practice, this makes AugSpecIV train more robustly, with minimal hyperparameter tuning required compared to DFIV.
>
> **W2. Presentation of the formal results.**
>
> We agree that the exposition would benefit from bringing the estimator-error story forward earlier. In a revision, we will add a short forward reference in Section 3 summarizing the approximation and estimation guarantees developed in Section 4, so that the operator interpretation and the eventual error consequences are connected more immediately.

---

> > ### Author Rebuttal · Reviewer_2D7T · 2026-04-02
> >
> > Thank you for your detailed responses. I might be missing something here, but let me try to clarify my question regarding the choice of $\delta$. I understand that once you have fitted $\hat{\mathcal{T}}_{\delta}$ you can decompose it using SVD and get an estimate of how well $h_0$ is aligned with it. However, in order to choose the "right" $\delta$, you should repeat this process, in theory, for all $\delta$, is that correct? If this is true, I believe it is important to explain how do you perform this search (e.g. do you discretize the range of $\delta$ and estimate for all values and choose the smallest one that has small loss, do you solve an optimization problem w.r.t. $\delta$) in a principled way that provably guarantees your final choice will be close to the optimal $\delta$.
> >
> > Let me clarify, I beleive the contributions of the paper are important regardless of this and I suggest acceptance. The experiments convince me that you can practically find the well-behaved $\delta$ using Proposition 6.1. However, ideally, I would like to see a complete algorithm that performs this search over $\delta$ (if this is indeed what is required) and outputs a good estimator of $h_0$, including the approximation error achieved (which you have) and the total runtime required for this method.

---

> > > ### Author Response · Authors · 2026-04-06
> > >
> > > We thank the reviewer for their continued engagement and insightful suggestions about the structure of the paper.
> > >
> > > Yes, operationally that is correct: our current procedure is a finite search over a small candidate set $\Delta$ of $\delta$ values (in practice, a short log-spaced grid including $\delta=0$). For each $\delta\in\Delta$, we
> > > 1. Train the feature model with $\widehat L_\delta$.
> > > 2. Compute the Proposition-6.1 plug-in alignment score from the empirical SVD of the fitted $\widehat T_\delta$.
> > > 3. Monitor the pair $(\widehat L_0,\widehat R_\delta)$ to exclude values where the supervised term begins to dominate and the approximation of $T$ deteriorates.
> > > 4. Fit the stage-2 2SLS estimator.
> > >
> > > We then choose the smallest positive $\delta$ whose estimated alignment has saturated while $\widehat L_0$ remains close to its $\delta=0$ value; held-out stage-2 validation is used only as a tie-breaker / safeguard. Thus, the runtime is linear in $|\Delta|$: one feature-learning run, one score computation, and one 2SLS fit per candidate, i.e. the same order as a standard hyperparameter sweep.
> > >
> > > What we do not currently claim is a separate oracle-selection theorem showing that the selected $\widehat\delta$ is uniformly close to the best $\delta$ in every problem. For each fixed candidate $\delta$, however, the paper gives the corresponding approximation/estimation guarantee. Empirically, on the additional synthetic sweep mentioned in our rebuttal, the Proposition-6.1 selector was, on average, close to oracle (defined as best $\delta$ in the considered set, not a globally optimal value), and on the hard dSprites task, it selected the same small-to-moderate-$\delta$ region as held-out validation. We agree that stating this search procedure, stopping rule, and runtime explicitly would strengthen the paper, and we will add a formal algorithm block in the revision.
> > >
> > > To speed up training, a promising avenue to consider is using a warm start. We could initialize training at $\delta=0$ and incrementally increase $\delta$ after initial convergence, training from the previous optimal weights.
> > >
> > > We will include a formal algorithm block detailing this search procedure, the stopping criteria, and the runtime analysis in the camera-ready revision.

---

### Official Review · Reviewer_4D8k · 2026-03-13

**Soundness:** 3
**Presentation:** 3
**Significance:** 3
**Originality:** 2
**Overall Recommendation:** 5
**Confidence:** 3

**Summary:**

The authors tackle the inverse problem of nonparametric instrumental variable regression (NPIV), by proposing a regularization (based on outcome information) on top of spectral feature approach. Solving the regularized objective corresponds to the spectral feature approach with a perturbed operator that better aligns the underlying function and the outcome. This approach potentially has utility for ill-conditioned settings where the operator and the underlying function (estimand) is not well-aligned. Theoretical guarantees are provided as well as extensive experiments to support the proposed method.

**Compliance With Llm Reviewing Policy:**

Affirmed.

**Final Justification:**

The technical details and their interpretations were mostly resolved in the rebuttal. Given technical clarity, I believe the paper extends the vanilla SpecIV in a meaningful, robust way where misspecification can be a universal issue. So I have increased my score to 5.

**Key Questions For Authors:**

1. For well-conditioned (well-aligned operator) case, how does the proposed method compare with the existing results?

2. For ill-conditioned case, what do you lose by gaining an aligned operator?

3. In page 5, the guarantee of the regularized estimator in both good/bad scenarios have an additive term $\|q_d\|_{L_2}$ and \|q_1\|_{L_2}/\{\lambda_k^2\|s_1\|_{L_2}\}$ --- do these decay? If so how? Can you provide a direct characterization of these terms with respect to dimension d and sample size n? If this isn't the case, then what do the guarantees imply? If the constant is inevitable in the NPIV literature, why is this the case?

**Limitations:**

More discussion around the main theoretical claim in page 5 would strengthen the paper. Especially the role of the additive term (in appearance not depending on sample size nor dimension) within the upper bound needs to be clarified.

**Strengths And Weaknesses:**

**Strengths**

- Clear motivation: The intuition is clear as to why regularizing via $Y$ information helps better align $h_0$ with the spectrum of the operator $\mathcal{T}$. Operator $\mathcal{T}$ takes any $h_0(X) \in L_2(X)$ to some fixed $r_0(Z) = \mathbb E[Y|Z]$, hence its spectrum must encode information about the outcome.

- Interpretation: Solving the regularized problem is shown to be equivalent to solving the inverse problem with perturbed operator

- Sound in experiments: The authors not only provide provable guarantees, but also provide extensive numerical experiments to further the case.

**Weaknesses**

- Structure of writing: Putting related works (section 5) after the introduction would have been more helpful in efficient understanding of the paper.

- Informal discussion on the theoretical guarantees depending on conditions: For well-conditioned (well-aligned operator) case, how does the proposed method compare with the existing results? For ill-conditioned case, what do you lose by gaining an aligned operator?

---

> ### Author Rebuttal · Authors · 2026-03-31
>
> We thank the reviewer for their positive assessment of the paper’s motivation, operator interpretation, and empirical study. We address the questions about the main theoretical claim below. We would be grateful if the reviewer might consider increasing their score if they find these clarifications satisfactory.
>
> **Q1. What happens in the well-aligned case?**
>
> Our method should be viewed as a superset of SpecIV [1]. When the target is already well-aligned with the leading singular subspace of $T$, setting $\delta=0$ exactly recovers vanilla SpecIV. In that regime, our general theorem reduces to the same good-case type of guarantee as in [1]: the error is controlled by the truncation bias outside the top-d singular space, the representation-learning error, and the statistical term. We therefore do not claim that a positive $\delta$ is necessary in spectrally favorable problems. This is also what we observe empirically: on the original dSprites target $h_{\mathrm{old}}$, which is spectrally favorable, small positive $\delta$ yields at most modest gains.
>
> **Q2. What is gained, and what is lost, in the ill-aligned case?**
>
> The gain is not that the inverse problem becomes intrinsically less ill-posed, but a better allocation of feature dimension. In the bad scenario, described on page 5, vanilla SpecIV [1] will need $d \ge k$ to include the relevant direction $v_k$, thereby spending $k-1$ features on irrelevant dominant modes. AugSpecIV can instead isolate the relevant direction with $d=1$ when $\delta$ opens a favorable gap $\gamma_1(\delta)$. The resulting bound still reflects the underlying ill-posedness through its $\lambda_k$-dependence, so we are not claiming that a weak direction becomes strong; the gain is that the learned representation focuses on the correct direction.
>
> What is lost is that increasing $\delta$ too much can hurt approximation of the original operator $T$. If the outcome-aware term dominates, the learned Z-features need only explain $r_0$, not the broader conditional expectation structure, and the baseline spectral loss $L_0^{(d)}$ can deteriorate. So the tradeoff is: better task alignment versus worse approximation of $T$ if $\delta$ is too large. This is precisely why the paper studies diagnostics and model-selection strategies for $\delta$.
>
> **Q3. What do the $q_d$ and $q_1$ terms mean, and do they decay?**
>
> These are approximation terms, not statistical terms. They play the role of truncation bias in the representation space. In the good case,
>
> $q_d=(I-\Pi_{V_d})h_0$
>
> is exactly the component of the target outside the selected d-dimensional signal subspace. Hence it decreases as $d$ increases, and it vanishes once the chosen singular space spans $h_0$. The page-5 theorem is stated for fixed $d$, which is why this term does not explicitly depend on $n$. If one lets $d=d(n)$, the bound becomes the usual bias-variance tradeoff: once the representation-learning term is controlled, the remaining terms are of the form
>
> $\|\|q_d\|\|+\lambda_d^{-1}\sqrt{d/n}.$
>
> Under standard source and singular-decay assumptions, $\|\|q_d\|\|\lesssim \sigma_d^r$, and if $\lambda_d\asymp d^{-p}$, this yields the explicit scaling
>
> $d^{-rp}+d^{p+1/2}n^{-1/2},$
>
> as in [1].
>
> In the one-feature bad-case analysis, $\|\|q_1\|\|/\ || s_1\|\|$ is a problem-dependent misspecification ratio relative to the targeted direction. It does not need to vanish with $n$, because it measures irreducible approximation error when the target is not exactly one-dimensional. The role of the theorem is therefore to separate:
>
> 1. approximation bias ($q_d$ or $q_1$),
> 2. representation-learning error ($E_d$),
> 3. statistical estimation error.
>
> So the bound should be read as a bias–variance–representation decomposition. Consistency is obtained by increasing $d$ appropriately so that the approximation term decreases while the statistical and representation-learning terms remain controlled.
>
> **W1. Related work placement.**
>
> We agree that moving the related work closer to the introduction would improve readability, and we will make that change in the revision.
>
> [1] Meunier D., Moulin A., Wornbard J., Kostic V., Gretton A. “Demystifying Spectral Feature Learning for Instrumental Variable Regression” (NeurIPS 2025)

---

> > ### Author Rebuttal · Reviewer_4D8k · 2026-04-04
> >
> > Thank you for the detailed rebuttal, all my questions were addressed. I have increased the score to 5.

---

### Decision · Program_Chairs · 2026-04-30

**Decision:**

Accept (regular)

**Comment:**

The paper identifies a meaningful limitation of existing spectral feature learning methods for NPIV and proposes a principled outcome-aware extension with a clear operator-theoretic interpretation. I find the contribution technically solid, well written and motivated, and practically relevant. The theory provides clear pointers when the augmentation helps and the experiments support robustness in these regimes. The main remaining limitations are mostly about tuning for $\delta$, which I don't consider a show stopper. Similarly questions around the strongest guarantees in my view do not outweigh the core contribution. I encourage the authors to include the clarifications from the rebuttal and also aim for a sober and honest positioning as a principled extension of SpecIV and not a fundamentally new paradigm in the final version.